# Local Hyper-Flow Diffusion

**Kimon Fountoulakis**
School of Computer Science
University of Waterloo
Waterloo, ON, Canada
kimon.fountoulakis@uwaterloo.ca

**Pan Li**
Department of Computer Science
Purdue University
West Lafayette, IN, United States
panli@purdue.edu

**Shenghao Yang**
School of Computer Science
University of Waterloo
Waterloo, ON, Canada
shenghao.yang@uwaterloo.ca

## Abstract

Recently, hypergraphs have attracted a lot of attention due to their ability to capture complex relations among entities. The insurgence of hypergraphs has resulted in data of increasing size and complexity that exhibit interesting small-scale and local structure, e.g., small-scale communities and localized node-ranking around a given set of seed nodes. Popular and principled ways to capture the local structure are the local hypergraph clustering problem and the related seed set expansion problem. In this work, we propose the first local diffusion method that achieves edge-size-independent Cheeger-type guarantee for the problem of local hypergraph clustering while applying to a rich class of higher-order relations that covers a number of previously studied special cases. Our method is based on a primal-dual optimization formulation where the primal problem has a natural network flow interpretation, and the dual problem has a cut-based interpretation using the $\ell_2$-norm penalty on associated cut-costs. We demonstrate the new technique is significantly better than state-of-the-art methods on both synthetic and real-world data.

## 1   Introduction

Hypergraphs [1] generalize graphs by allowing a hyperedge to consist of multiple nodes that capture higher-order relationships in complex systems and datasets [2]. Hypergraphs have been used for music recommendation on Last.fm data [3], news recommendation [4], sets of product reviews on Amazon [5], and sets of co-purchased products at Walmart [6]. Beyond the internet, hypergraphs are used for analyzing higher-order structure in neuronal, air-traffic and food networks [7, 8].

In order to explore and understand higher-order relationships in hypergraphs, recent work has made use of cut-cost functions that are defined by associating each hyperedge with a specific set function. These functions assign specific penalties of separating the nodes within individual hyperedges. They generalize the notion of hypergraph cuts and are crucial for determining small-scale community structure [8, 9]. The most popular cut-cost functions with increasing capability to model complex multiway relationships are the unit cut-cost [10–12], cardinality-based cut-cost [13, 14] and general submodular cut-cost [15, 16]. An illustration of a hyperedge and the associated cut-cost function is given in Figure 1. In the simplest setting, all cut-cost functions take value either 0 or 1 (e.g., the case when $\gamma_1 = \gamma_2 = 1$ in Figure 1b), we obtain a unit cut-cost hypergraph. In a slightly more general setting, the cut-costs are determined solely by the number of nodes in either side of the hyperedge cut (e.g., the case when $\gamma_1 = 1/2$ and $\gamma_2 = 1$ in Figure 1b), we obtain a cardinality-based hypergraph.

35th Conference on Neural Information Processing Systems (NeurIPS 2021).

We refer to hypergraphs associated with arbitrary submodular cut-cost functions (e.g., the case when $\gamma_1 = 1/2$ and $0 \le \gamma_2 \le 1$ in Figure 1b) as general submodular hypergraphs.

Hypergraphs that arise from data science applications consist of interesting small-scale local structure such as local communities [9, 17]. Exploiting this structure is central to the above mentioned applications on hypergraphs and related applications in machine learning and applied mathematics [18]. We consider local hypergraph clustering as the task of finding a community-like cluster around a given set of seed nodes, where nodes in the cluster are densely connected to each other while relatively isolated to the rest of the graph. One of the most powerful primitives for the local hypergraph clustering task is the graph diffusion. Diffusion on a graph is the process of spreading a given initial mass from some seed node(s) to neighbor nodes using the edges of the graph. Graph diffusions have been successfully employed in the industry, for example, both Pinterest and Twitter use diffusion methods for their recommendation systems [19–21]. Google uses diffusion methods to perform clustering query refinements [22]. Let us not forget PageRank [23, 24], Google's model for their search engine.

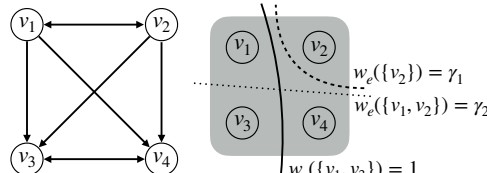

Empirical and theoretical performance of local diffusion methods is often measured on the problem of local hypergraph clustering [9, 25, 26]. Existing local diffusion methods only directly apply to hypergraphs with the unit cut-cost [17,27]. For the slightly more general cardinality-based cut-cost, they rely on graph reduction techniques which result in a rather pessimistic edge-size-dependent approximation error [9, 13, 27, 28]. Moreover, none of the existing methods is capable of processing general submodular cut-costs. In this work, we are interested in designing a diffusion framework that (i) achieves stronger theoretical guarantees for the problem of local hypergraph clustering, (ii) is flexible enough to work with general submodular hypergraphs, and (iii) permits computationally efficient algorithms. We propose the first local diffusion method that simultaneously accomplishes these goals.

In what follows we describe our main contributions and previous work. In Section 2 we provide preliminaries and notations. In Section 3 we introduce our diffusion model from a combinatorial flow perspective. In Section 4 we discuss the local hypergraph clustering problem and Cheeger-type quadratic approximation error. In Section 6 we perform experiments using both synthetic and real datasets.

(a) Network motif (b) Hyperedge and cut-cost $w_e$

Figure 1: A food network can be mapped into a hypergraph by taking each network pattern in (a) as a hyperedge [8]. This network pattern captures carbon flow from two preys $(v_1, v_2)$ to two predators $(v_3, v_4)$. (b) is a hyperedge associated with cut-cost $w_e$ that models their relations: $w_e$ is a set function defined over the node set $e$ s.t. $w_e(\{v_i\}) = \gamma_1$ for $i = 1, 2, 3, 4$, $w_e(\{v_1, v_2\}) = \gamma_2$, $w_e(\{v_1, v_3\}) = w_e(\{v_1, v_4\}) = 1$ and $w_e(S) = w_e(e \backslash S)$ for $S \subseteq e$. $w_e$ becomes the unit cut-cost when $\gamma_1 = \gamma_2 = 1$; $w_e$ is cardinality-based if $\gamma_1 = 1/2$ and $\gamma_2 = 1$; more generally, $w_e$ is submodular if $\gamma_1 = 1/2$ and $0 \le \gamma_2 \le 1$. The specific choices depend on the application.

## 1.1 Our main contributions

In this work we propose a generic local diffusion model that applies to hypergraphs characterized by a rich class of cut-cost functions that covers many previously studied special cases, e.g., unit, cardinality-based and submodular cut-costs. We provide the first edge-size-independent Cheeger-type approximation error for the problem of local hypergraph clustering using any of these cut-costs. In particular, assume that there exists a cluster $C$ with conductance $\Phi(C)$, and assume that we are given a set of seed nodes that reasonably overlaps with $C$, then the proposed diffusion model can be used to find a cluster $\hat{C}$ with conductance at most $O(\sqrt{\Phi(C)})$ (in the appendix we show that an $\ell_p$-norm version of the proposed model can achieve $O(\Phi(C))$ asymptotically). Our hypergraph diffusion model is formulated as a convex optimization problem. It has a natural combinatorial flow interpretation that generalizes the notion of network flows over hyperedges. We show that the optimization problem can be solved efficiently by an alternating minimization method. In addition, we prove that the number of nonzero nodes in the optimal solution is independent of the size of the hypergraph, and it only depends on the size of the initial mass. This key property ensures that our algorithm scales well in practice for large datasets. We evaluate our method using both synthetic and real-world data. We show that our method improves accuracy significantly for hypergraphs with unit, cardinarlity-based and general submodular cut-costs for local clustering.

## 1.2 Previous work

Recently, clustering methods on hypergraphs received renewed interest. Different methods require different assumptions about the hyperedge cut-cost, which can be roughly categorized into unit cut-cost, cardinality-based (and submodular) cut-cost and general submodular cut-cost. Moreover, existing methods can be either global, where the output is not localized around a given set of seed nodes, or local, where the output is a tiny cluster around a set of seed nodes. Local algorithms are the only scalable ones for large hypergraphs, which is our main focus. Many works propose global methods and thus they are not scalable to large hypergraphs [7,8,15–17,29–38]. Local diffusion-based methods are more relevant to our work [9,13,27]. In particular, iterative hypergraph min-cut methods for the local hypergraph clustering problem can be adopted [13]. However, these methods require in theory and in practice a large seed set, i.e., they are not expansive and thus cannot work with one seed node. The expansive combinatorial diffusion [25] is generalized for hypergraphs [27], which can detect a target cluster using only one seed node. However, combinatorial methods have a large bias towards low conductance clusters as opposed to finding the target cluster [39]. The most relevant paper to our work is [9]. However, the proposed methods in [9] depend on a reduction from hypergraphs to directed graphs. This results in an approximation error for clustering that is proportional to the size of hyperedges and induces performance degeneration when the hyperedges are large. In fact, none of the above approaches (including global and local ones) has an edge-size-independent approximation error bound for even simple cardinality-based hypergraphs. Moreover, existing local approaches do not work for general submodular hypergraphs.

## 2 Preliminaries and Notations

**Submodular function.** Given a set $S$, we denote $2^S$ the power set of $S$ and $|S|$ the cardinality of $S$. A submodular function $F : 2^S \to \mathbb{R}$ is a set function such that $F(A) + F(B) \geq F(A \cup B) + F(A \cap B)$ for any $A, B \subseteq S$.

**Submodular hypergaph.** A hypergraph $H = (V, E)$ is defined by a set of nodes $V$ and a set of hyperedges $E \subseteq 2^V$, i.e., each hyperedge $e \in E$ is a subset of $V$. A hypergraph is termed *submodular* if every $e \in E$ is associated with a submodular function $w_e : 2^e \to \mathbb{R}_{\geq 0}$ [15]. The weight $w_e(S)$ indicates the cut-cost of splitting the hyperedge $e$ into two subsets, $S$ and $e \setminus S$. This general form allows us to describe the potentially complex higher-order relation among multiple nodes (Figure 1). A proper hyperedge weight $w_e$ should satisfy that $w_e(\emptyset) = w_e(e) = 0$. To ease notation we extend the domain of $w_e$ to $2^V$ by setting $w_e(S) := w_e(S \cap e)$ for any $S \subseteq V$. We assume without loss of generality that $w_e$ is normalized by $\vartheta_e := \max_{S \subseteq e} w_e(S)$, so that $w_e(S) \in [0, 1]$ for any $S \subseteq V$. For the sake of simplicity in presentation, we assume that $\vartheta_e = 1$ for all $e$.[1] A submodular hypergraph is written as $H = (V, E, \mathcal{W})$ where $\mathcal{W} := \{w_e, \vartheta_e\}_{e \in E}$. Note that when $w_e(S) = 1$ for any $S \in 2^e \setminus \{\emptyset, e\}$, the definition reduces to *unit cut-cost hypergraphs*. When $w_e(S)$ only depends on $|S|$, it reduces to *cardinality-based cut-cost hypergraphs*.

**Vector/Function on $V$ or $E$.** For a set of nodes $S \subseteq V$, we denote $1_S$ the indicator vector of $S$, i.e., $[1_S]_v = 1$ if $v \in S$ and 0 otherwise. For a vector $x \in \mathbb{R}^{|V|}$, we write $x(S) := \sum_{v \in S} x_v$, where $x_v$ in the entry in $x$ that corresponds to $v \in V$. We define the support of $x$ as $\mathrm{supp}(x) := \{v \in V | x_v \neq 0\}$. The support of a vector in $\mathbb{R}^{|E|}$ is defined analogously. We refer to a function over nodes $x : V \to \mathbb{R}$ and its explicit representation as a $|V|$-dimensional vector interchangeably.

**Volume, cut, conductance.** Given a submodular hypergraph $H = (V, E, \mathcal{W})$, the *degree* of a node $v$ is defined as $d_v := |\{e \in E : v \in e\}|$. We reserve $d$ for the vector of node degrees and $D = \mathrm{diag}(d)$. We refer to $\mathrm{vol}(S) := d(S)$ as the *volume* of $S \subseteq V$. A *cut* is treated as a proper subset $C \subset V$, or a partition $(C, \bar{C})$ where $\bar{C} := V \setminus C$. The *cut-set* of $C$ is defined as $\partial C := \{e \in E | e \cap C \neq \emptyset, e \cap \bar{C} \neq \emptyset\}$; the *cut-size* of $C$ is defined as $\mathrm{vol}(\partial C) := \sum_{e \in \partial C} \vartheta_e w_e(C) = \sum_{e \in E} \vartheta_e w_e(C)$. The *conductance* of a cut $C$ in $H$ is $\Phi(C) := \frac{\mathrm{vol}(\partial C)}{\min\{\mathrm{vol}(C), \mathrm{vol}(V \setminus C)\}}$.

**Flow.** A flow *routing* over a hyperedge $e$ is a function $r_e : e \to \mathbb{R}$ where $r_e(v)$ specifies the amount of mass that flows from $\{v\}$ to $e \setminus \{v\}$ over $e$. To ease notation we extend the domain of $r_e$ to $V$ by identifying $r_e(v) = 0$ for $v \notin e$, so $r_e$ is treated as a function $r_e : V \to \mathbb{R}$ or equivalently a $|V|$-dimensional vector. The net (out)flow at a node $v$ is given by $\sum_{e \in E} r_e(v)$. Given a routing

---

[1] This is without loss of generality. In the appendix we show that our method works with arbitrary $\vartheta_e > 0$.

function $r_e$ and a set of nodes $S \subseteq V$, a *directional routing* on $e$ with direction $S \to e \setminus S$ is represented by $r_e(S)$, which specifies the net amount of mass that flows from $S$ to $e \setminus S$. A routing $r_e$ is called *proper* if it obeys flow conservation, i.e., $r_e^T 1_e = 0$. Our flow definition generalizes the notion of network flows to hypergraphs. We provide concrete illustrations in Figure 2.

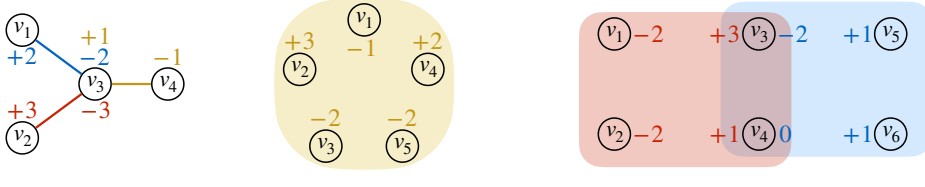

(a) Flows on graph          (b) Hyperedge routing          (c) Flows on hypergraph

Figure 2: Illustration of proper flow routings. The numbers next to each node correspond to entries in the flow routing $r_e$ over a (hyper)edge $e$. We assign the same color to a (hyper)edge and its associated flow values. Our flow definition is a natural generalization of graph edge flow where $r_e(v) = \pm f$ if and only if $v \in e$, i.e., $v$ is incident to $e$, where $f$ and the sign determine the amplitude and direction of the flow over $e$. In Figure 2a, the net (out)flow at node $v_3$ is given by $\sum_{e \in E} r_e(v_3) = 1 - 2 - 3 = -4$. In Figure 2b, the directional flow from $\{v_1\}$ to $\{v_2, v_3, v_4, v_5\}$ over this hyperedge equals $-1$; similarly, the directional flow from $\{v_1, v_2, v_4\}$ to $\{v_3, v_5\}$ equals $3 + 2 - 1 = 4$. In Figure 2c, the net (out)flow at node $v_3$ is given by $\sum_{e \in E} r_e(v_3) = 3 - 2 = 1$.

## 3   Diffusion as an Optimization Problem

In this section we provide details of the proposed local diffusion method. We consider diffusion process as the task of spreading mass from a small set of seed nodes to a larger set of nodes. More precisely, given a hypergraph $H = (V, E, \mathcal{W})$, we assign each node a sink capacity specified by a sink function $T$, i.e., node $v$ is allowed to hold at most $T(v)$ amount of mass. In this work we focus on the setting where $T(v) = d_v$, so that a high-degree node that is part of many hyperedges can hold more mass than a low-degree node that is part of few hyperedges. Moreover, we assign each node some initial mass specified by a source function $\Delta$, i.e., node $v$ holds $\Delta(v)$ amount of mass at the start of the diffusion. In order to encourage the spread of mass in the hypergraph, the initial mass on the seed nodes is larger than their capacity. This forces the seed nodes to diffuse mass to neighbor nodes to remove their excess mass. In Section 4 we will discuss the choice of $\Delta$ to obtain good theoretical guarantees for the problem of local hypergraph clustering. Details about the local hypergraph clustering problem are provided in Section 4.

Given a set of proper flow routings $r_e$ for $e \in E$, recall that $\sum_{e \in E} r_e(v)$ specifies the amount of net (out)flow at node $v$. Therefore, the vector $m = \Delta - \sum_{e \in E} r_e$ gives the amount of net mass at each node after routing. The *excess mass* at a node $v$ is $\text{ex}(v) := \max\{m_v - d_v, 0\}$. In order to force the diffusion of initial mass we could simply require that $\text{ex}(v) = 0$ for all $v \in V$, or equivalently, $\Delta - \sum_{e \in E} r_e \leq d$. But to provide additional flexibility in the diffusion dynamics, we introduce a hyper-parameter $\sigma \geq 0$ and we impose a softer constraint $\Delta - \sum_{e \in E} r_e \leq d + \sigma D z$, where $z \in \mathbb{R}^{|V|}$ is an optimization variable that controls how much excess mass is allowed on each node. In the context of numerical optimization, we show in Section 5 that $\sigma$ allows a reformulation which makes the optimization problem amenable to efficient alternating minimization schemes.

Note that so far we have not yet talked about how specific higher-order relations among nodes within a hyperedge would affect the flow routings over it. Apparently, simply requiring that the $r_e$'s obey flow conservation (i.e., $r_e^T 1_e = 0$) similar to the standard graph setting is not enough for hypergraphs. An important difference between hyperedge flows and graph edge flows is that additional constraints on $r_e$ are in need. To this end, we consider $r_e = \phi_e \rho_e$ for some $\phi_e \in \mathbb{R}_+$ and $\rho_e \in B_e$, where

$$B_e := \{\rho_e \in \mathbb{R}^{|V|} \mid \rho_e(S) \leq w_e(S), \forall S \subseteq V, \text{ and } \rho_e(V) = w_e(V)\}$$

is the *base polytope* [40] for the submodular cut-cost $w_e$ associated with hyperedge $e$. It is straightforward to see that $r_e(v) = 0$ for every $v \notin e$ and $r_e^T 1_e = 0$, so $r_e$ defines a proper flow routing over $e$. Moreover, for any $e \subseteq V$, recall that $r_e(S)$ represents the net amount of mass that moves from $S$ to $e \setminus S$ over hyperedge $e$. Therefore, the constraints $\rho_e(S) \leq w_e(S)$ for $S \subseteq e$ mean that the directional flows $r_e(S)$ are upper bounded by a submodular function $\phi_e w_e(S)$. Intuitively, one may think of $\phi_e$ and $\rho_e$ as the *scale* and the *shape* of $r_e$, respectively.

The goal of our diffusion problem is to find low cost routings $r_e \in \phi_e B_e$ for $e \in E$ such that the capacity constraint $\Delta - \sum_{e \in E} r_e \leq d + \sigma D z$ is satisfied. We consider the (weighted) $\ell_2$-norm of $\phi$ and $z$ as the cost of diffusion. In the appendix we show that one readily extends the $\ell_2$-norm to $\ell_p$-norm for any $p \geq 2$. Formally, we arrive at the following convex optimization formulation (input: the source function $\Delta$, the hypergraph $H = (V, E, \mathcal{W})$, and a hyper-parameter $\sigma$):

$$\min_{\phi \in \mathbb{R}_+^{|E|}, z \in \mathbb{R}_+^{|V|}} \frac{1}{2} \sum_{e \in E} \phi_e^2 + \frac{\sigma}{2} \sum_{v \in V} d_v z_v^2, \text{ s.t. } \Delta - \sum_{e \in E} r_e \leq d + \sigma D z, \ r_e \in \phi_e B_e, \forall e \in E. \quad (1)$$

We name problem (1) Hyper-Flow Diffusion (HFD) for its combinatorial flow interpretation we discussed above. The dual problem of (1) is:

$$\max_{x \in \mathbb{R}_+^{|V|}} (\Delta - d)^T x - \frac{1}{2} \sum_{e \in E} f_e(x)^2 - \frac{\sigma}{2} \sum_{v \in V} d_v x_v^2, \quad (2)$$

where $f_e$ in (2) is the support function of the base polytope $B_e$ given by $f_e(x) := \max_{\rho_e \in B_e} \rho_e^T x$. $f_e$ is also known as the *Lovász extension* of the submodular function $w_e$.

We provide a combinatorial interpretation for (2) and leave algebraic derivations to the appendix. For the dual problem, one can view the solution $x$ as assigning heights to nodes, and the goal is to separate/cut the nodes with source mass from the rest of the hypergraph. Observe that the linear term in the dual objective function encourages raising $x$ higher on the seed nodes and setting it lower on others. The cost $f_e(x)$ captures the discrepancy in node heights over a hyperedge $e$ and encourages smooth height transition over adjacent nodes. The dual solution embeds nodes into the nonnegative real line, and this embedding is what we actually use for local clustering and node ranking.

## 4   Local Hypergraph Clustering

In this section we discuss the performance of the primal-dual pair (1)-(2), respectively, in the context of local hypergraph clustering. We consider a generic hypergraph $H = (V, E, \mathcal{W})$ with submodular hyperedge weights $\mathcal{W} = \{w_e, \vartheta_e\}_{e \in E}$. Given a set of seed nodes $S \subset V$, the goal of local hypergraph clustering is to identify a target cluster $C \subset V$ that contains or overlaps well with $S$. This generalizes the definition of local clustering over graphs [41]. To the best of our knowledge, we are the first one to consider this problem for general submodular hypergraphs. We consider a subset of nodes having low conductance as a good cluster, i.e., these nodes are well-connected internally and well-separated from the rest of the hypergraph. Following prior work on local hypergraph clustering, we assume the existence of an unknown target cluster $C$ with conductance $\Phi(C)$. We prove that applying sweep-cut to an optimal solution $\hat{x}$ of (2) returns a cluster $\hat{C}$ whose conductance is at most quadratically worse than $\Phi(C)$. Note that this result resembles Cheeger-type approximation guarantees of spectral clustering in the graph setting [42], and it is the first result that is independent of hyperedge size for general hypergraphs. We keep the discussion at high level and defer details to the appendix, where we prove a more general, and stronger, i.e., constant approximation error result when the primal problem (1) is penalized by the $\ell_p$-norm for any $p \geq 2$.

In order to start a diffusion process we need to provide the source mass $\Delta$. Similar to the $p$-norm flow diffusion in the graph setting [26], we let

$$\Delta(v) = \begin{cases} \delta d_v & \text{if } v \in S, \\ 0 & \text{otherwise,} \end{cases} \quad (3)$$

where $S$ is a set of seed nodes and $\delta \geq 1$. Below, we make the assumptions that the seed set $S$ and the target cluster $C$ have some overlap, there is a constant factor of $\text{vol}(C)$ amount of mass trapped in $C$ initially, and the hyper-parameter $\sigma$ is not too large. Note that Assumption 2 is without loss of generality: if the right value of $\delta$ is not known apriori, we can always employ binary search to find a good choice. Assumption 3 is very weak as it allows $\sigma$ to reside in an interval containing 0.

**Assumption 1.** $\text{vol}(S \cap C) \geq \alpha \text{vol}(C)$ *and* $\text{vol}(S \cap C) \geq \beta \text{vol}(S)$ *for some* $\alpha, \beta \in (0, 1]$.

**Assumption 2.** *The source mass* $\Delta$ *as specified in* (3) *satisfies* $\delta = 3/\alpha$, *so* $\Delta(C) \geq 3 \text{vol}(C)$.

**Assumption 3.** $\sigma$ *satisfies* $0 \leq \sigma \leq \beta \Phi(C)/3$.

Let $\hat{x}$ be an optimal solution for the dual problem (2). For $h > 0$ define the sweep sets $S_h := \{v \in V | \hat{x}_v \geq h\}$. We state the approximation property in Theorem 1.

**Theorem 1.** *Under Assumptions 1, 2, 3, there exists $h > 0$ such that $\Phi(S_h) \leq O(\sqrt{\Phi(C)}/\alpha\beta)$.*

Some previous works on local hypergraph clustering provide an approximation guarantee by focusing on a specific setting when the seed set $S$ contains precisely one seed node, e.g., [9, 17]. However, they rely on different assumptions about the seed node which may not hold in general. For example, [17] assumes that the seed node is not a neighbor of any node not belonging to the target cluster. Even in the classical graph setting, establishing a local clustering guarantee based on diffusing mass from a single node requires assuming that a good seed node has been selected from a proper subset of the target cluster [42]. On the other hand, our theoretical framework is more natural in the sense that Theorem 1 applies to any set of seed node(s) as long as it has nonempty intersection with the target. We view the $\alpha$ in the denominator of the bound as the price we pay for being certain that the approximation guarantee holds for an arbitrary seed set. The empirical experiments in Section 6 show that our method has superior performance in practice even when only one seed node is used.

One of the challenges we face in establishing the result in Theorem 1 is making sure that our diffusion model enjoys not only good clustering guarantees but also practical algorithmic advantages. This is achieved by introducing the hyper-parameter $\sigma$ to our diffusion problem. We will demonstrate how $\sigma$ helps with algorithmic development in Section 5, but from a clustering perspective, the additional flexibility given by $\sigma > 0$ complicates the underlying diffusion dynamics, making it more difficult to analyze. Another challenge is connecting the Lovász extension $f_e(x)$ in (2) with the conductance of a cluster. We resolve all these problems by combining a generalized Rayleigh quotient result for submodular hypergraphs [15], primal-dual convex conjugate relations between (1) and (2), and a classical property of the Choquet integral/Lovász extension.

Let $(\hat{\phi}, \hat{r}, \hat{z})$ be an optimal solution for the primal problem (1). We state the following lemma on the locality (i.e., sparsity) of the optimal solutions, which justifies why HFD is a local diffusion method.

**Lemma 2.** $|\text{supp}(\hat{\phi})| \leq \text{vol}(\text{supp}(\hat{x})) \leq \|\Delta\|_1$; *moreover,* $\text{vol}(\text{supp}(\hat{z})) = \text{vol}(\text{supp}(\hat{x}))$ *if* $\sigma > 0$.

## 5 Optimization algorithm for HFD

We use a simple Alternating Minimization (AM) [43] method that efficiently solves the primal diffusion problem (1). For $e \in E$, we define a diagonal matrix $A_e \in \mathbb{R}^{|V| \times |V|}$ such that $[A_e]_{v,v} = 1$ if $v \in e$ and 0 otherwise. Denote $\mathcal{C} := \{(\phi, r) : r_e \in \phi_e B_e, \forall e \in E\}$. The following Lemma 3 allows us to cast problem (1) to an equivalent separable formulation amenable to the AM method.

**Lemma 3.** *The following problem is equivalent to* (1) *for any $\sigma > 0$, in the sense that $(\hat{\phi}, \hat{r}, \hat{z})$ is optimal in* (1) *for some $\hat{z} \in \mathbb{R}^{|V|}$ if and only if $(\hat{\phi}, \hat{r}, \hat{s})$ is optimal in* (4) *for some $\hat{s} \in \bigotimes_{e \in E} \mathbb{R}^{|V|}$.*

$$\min_{\phi, r, s} \frac{1}{2} \sum_{e \in E} \left( \phi_e^2 + \frac{1}{\sigma} \|s_e - r_e\|_2^2 \right), \ s.t. \ (\phi, r) \in \mathcal{C}, \ \Delta - \sum_{e \in E} s_e \leq d, \ s_{e,v} = 0, \forall v \notin e. \quad (4)$$

The AM method for problem (4) is given in Algorithm 1. The first sub-problem corresponds to computing projections to a group of cones, where all the projections can be computed in parallel. The computation of each projection depends on the choice of base polytope $B_e$. If the hyperedge weight $w_e$ is unit cut-cost, $B_e$ holds special structures and projection can be computed with $O(|e| \log |e|)$ [36]. For general $B_e$, a conic Fujishige-Wolfe minimum norm algorithm can be adopted to obtain the projection [36]. The second sub-problem in Algorithm 1 can be easily computed in closed-form. We provide more information about Algorithm 1 and its convergence properties in the appendix.

---

**Algorithm 1** Alternating Minimization for (4)

**Initialization:**

$\phi^{(0)} := 0, r^{(0)} := 0, s_e^{(0)} := D^{-1} A_e \left[ \Delta - d \right]_+, \forall e \in E.$

**For** $k = 0, 1, 2, \dots$ **do:**

$(\phi^{(k+1)}, r^{(k+1)}) := \underset{(\phi, r) \in \mathcal{C}}{\text{argmin}} \sum_{e \in E} (\phi_e^2 + \frac{1}{\sigma} \|s_e^{(k)} - r_e\|_2^2)$

$s^{(k+1)} := \underset{s}{\text{argmin}} \sum_{e \in E} \|s_e - r_e^{(k+1)}\|_2^2$

$$\text{s.t. } \Delta - \sum_{e \in E} s_e \leq d, \ s_{e,v} = 0, \forall v \notin e.$$

---

We remark that the reformulation (4) for $\sigma > 0$ is crucial from an algorithmic point of view. If $\sigma = 0$, then the primal problem (1) has complicated coupling constraints that are hard to deal with. In this

case, one has to resort to the dual problem (2). However, problem (2) has a nonsmooth objective function, which prohibits applicability of optimization methods for smooth objective functions. Even though subgradient method may be applied, we have observed empirically that its convergence rate is extremely slow for our problem, and early stopping results in a bad quality output.

Lastly, as noted in Lemma 2, the number of nonzeros in the optimal solution is upper bounded by $\|\Delta\|_1$. In Figure 3 we plot the number of nodes having positive excess (which equals the number of nonzeros in the dual solution $\hat{x}$) at every iteration of Algorithm 1. Figure 3 indicates that Algorithm 1 is strongly local, meaning that it works only on a small fraction of nodes (and their incident hyperedges) as opposed to producing dense iterates. This key empirical observation has enabled our algorithm to scale to large datasets by simply keeping track of all active nodes and hyperedges. Proving that the worst-case running time of AM depends only on the number of nonzero nodes at optimality as opposed to size of the whole hypergraph is an open problem, which we leave for future work.

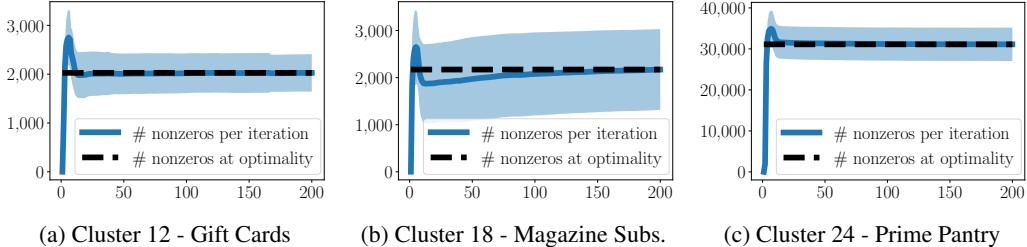

(a) Cluster 12 - Gift Cards    (b) Cluster 18 - Magazine Subs.    (c) Cluster 24 - Prime Pantry

Figure 3: The blue solid line plots the number of nonzeros in the dual solution $x$ over 200 iterations of Algorithm 1, when it is applied to solve HFD on the Amazon-reviews hypergraph for local clustering. See Section 6.2 for details about the dataset. The error bars show standard deviation over 10 trials. In each trial we pick a different seed node and set the same amount of initial mass. The black dashed line shows the average number of nonzeros at optimality. The algorithm touches only a small fraction of nodes out the total 2,268,264 nodes in the Amazon-reviews dataset.

## 6 Empirical Results

In this section we evaluate the performance of HFD for local clustering. First, we carry out experiments on synthetic hypergraphs with varying target cluster conductances and varying hyperedge sizes. For the unit cut-cost setting, we show that HFD is more robust and has better performance when the target cluster is noisy; for a cardinality-based cut-cost setting, we show that the edge-size-independent approximation guarantee is important for obtaining good recovery results. Second, we carry out experiments using real-world data. We show that HFD significantly outperforms existing state-of-the-art diffusion methods for both unit and cardinality-based cut-costs. Moreover, we provide a compelling example where specialized submodular cut-cost is necessary for obtaining good results. Code that reproduces all results is available at `https://github.com/s-h-yang/HFD`.

### 6.1 Synthetic experiments using hypergraph stochastic block model (HSBM)

**The generative model.** We generalize the standard $k$-uniform hypergraph stochastic block model ($k$HSBM) [44] to allow different types of inter-cluster hyperedges appear with possibly different probabilities according to the cardinality of hyperedge cut. Let $V = \{1, 2, \ldots, n\}$ be a set of nodes and let $k \geq 2$ be the required constant hyperedge size. We consider $k$HSBM with parameters $k$, $n$, $p$, $q_j$, $j = 1, 2, \ldots, \lfloor k/2 \rfloor$. The model samples a $k$-uniform hypergraph according to the following rules: (i) The community label $\sigma_i \in \{0, 1\}$ is chosen uniformly at random for $i \in V$;[2] (ii) Each size $k$ subset $e = \{v_1, v_2, \ldots, v_k\}$ of $V$ appears independently as a hyperedge with probability

$$\mathbb{P}(e \in E) = \begin{cases} p & \text{if } \sigma_{v_1} = \sigma_{v_2} = \cdots = \sigma_{v_k}, \\ q_j & \text{if } \min\{k - \sum_{i=1}^{k} \sigma_{v_i}, \sum_{i=1}^{k} \sigma_{v_i}\} = j. \end{cases}$$

If $k = 3$ or all $q_j$'s are the same, then we obtain the standard two-block $k$HSBM. We use this setting to evaluate HFD for unit cut-cost. If $q_j$'s are different, then we obtain a cardinality-based $k$HSBM.

---

[2]We consider two blocks for simplicity. In general the model applies to any number of blocks.

In particular, when $q_1 \geq q_2 \geq \cdots \geq q_{\lfloor k/2 \rfloor}$, it models the scenario where hyperedges containing similar numbers of nodes from each block are rare, while small noises (e.g., hyperedges that have one or two nodes in one block and all the rest in the other block) are more frequent. We use $q_1 \gg q_j$, $j \geq 2$, to evaluate HFD for cardinality-based cut-cost. There are other random hypergraph models, for example the Poisson degree-corrected HSBM [45] that deals with degree heterogeneity and edge size heterogeneity. In our experiments we focus on $k$HSBM because it allows stronger control over hyperedge sizes. We provide details on data generation in the appendix.

**Task and methods.** We consider the local hypergraph clustering problem. We assume that we are given a single labelled node and the goal is to recover all nodes having the same label. Using a single seed node the most common (and sought-after) practice for local graph clustering tasks. We test the performance of HFD with two other methods: (i) Localized Quadratic Hypergraph Diffusions (LH) [9], which can be seen as a hypergraph analogue of Approximate Personalized PageRank (APPR); (ii) ACL [42], which is used to compute APPR vectors on a standard graph obtained from reducing a hypergraph through star expansion [46].[3]

**Cut-costs and parameters.** We consider both unit cut-cost, i.e., $w_e(S) = 1$ if $S \cap e \neq \emptyset$ and $e \setminus S \neq \emptyset$, and *cardinality cut-cost* $w_e(S) = \min\{|S \cap e|, |e \setminus S|\}/\lfloor |e|/2 \rfloor$. HFD that uses unit and cardinality cut-costs are denoted by U-HFD and C-HFD, respectively. LH also works with both unit and cardinality cut-costs and we specify them by U-LH and C-LH, respectively. For HFD, we initialize the seed mass so that $\|\Delta\|_1$ is a constant factor times the volume of the target cluster. We set $\sigma = 0.01$. We highly tune LH by performing binary search over its parameters $\kappa$ and $\delta$ and pick the output cluster having the lowest conductance. For ACL we use the same parameter choices as in [9]. Details on parameter setting are provided in the appendix.

**Results.** For each hypergraph, we randomly pick a block as the target cluster. We run the methods 50 times. Each time we choose a different node from the target cluster as the single seed node.

*Unit cut-cost results.* Figure 4 shows local clustering results when we fix $k = 3$ but vary the conductance of the target cluster (i.e., constant $p$ but varying $q_1$). Observe that the performances of all methods become worse as the target cluster becomes more noisy, but U-HFD has significantly better performance than both U-LH and ACL when the conductance of the target cluster is between 0.2 and 0.4. The reason that U-HFD performs better is in part because it requires much weaker conditions for the theoretical conductance guarantee to hold. On the contrary, LH assumes an upper bound on the conductance of the target cluster [9]. This upper bound is dataset-dependent and could become very small in many cases, leading to poor practical performances. We provide more details in this perspective in the appendix. ACL with star expansion is a heuristic method that has no performance guarantee.

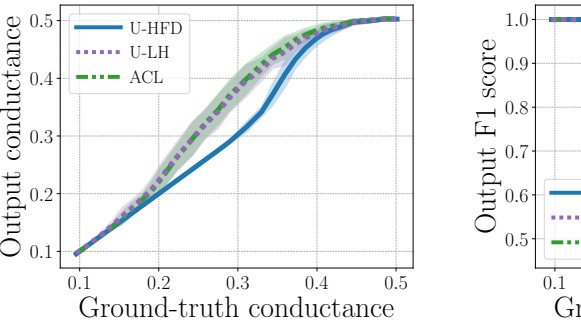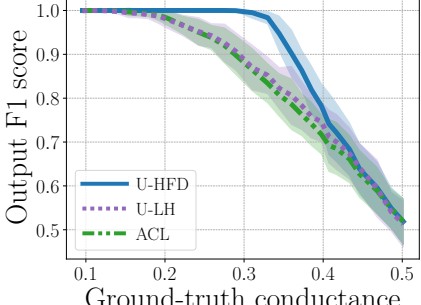

Figure 4: Output conductance and F1 against ground-truth conductance

*Cardinality cut-cost results.* Figure 5 shows the median (markers) and 25-75 percentiles (lower-upper bars) of conductance ratios (i.e., the ratio between output conductance and ground-truth conductance, lower is better) and F1 scores for different methods for $k \in \{3, 4, 5, 6\}$. The target cluster for each $k$

---

[3]There are other heuristic methods which first reduce a hypergraph to a graph by clique expansion [7] and then apply diffusion methods for standard graphs. We did not compare with this approach because clique expansion often results in a dense graph and consequently makes the computation slow. Moreover, it has been shown in [9] that clique expansion did not offer significant performance improvement over star expansion.

has conductance around 0.3.[4] For $k = 3$, unit and cardinality cut-costs are equivalent, therefore all methods have similar performances. As $k$ increases, cardinality cut-cost provides better performance than unit cut-cost in both conductance and F1. However, since the theoretical approximation guarantee of C-LH depends on hyperedge size [9], there is a noticeable performance degradation for C-LH when we increase $k = 3$ to $k = 4$. On the other hand, the performance of C-HFD appears to be independent from $k$, which aligns with our conductance bound in Theorem 1.

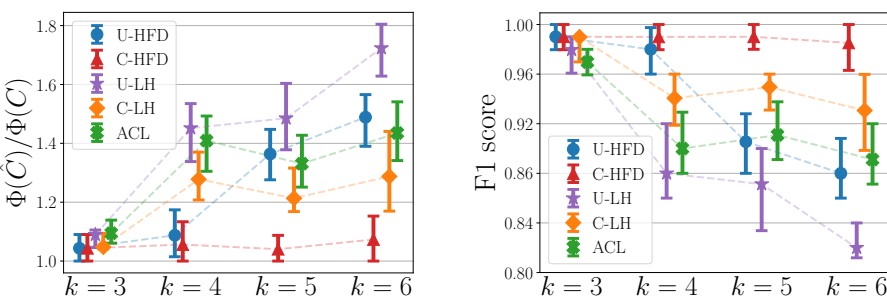

Figure 5: Conductance ratio and F1 on $k$-uniform hypergraphs

## 6.2 Experiments using real-world data

We conduct extensive experiments using real-world data. First, we show that HFD has superior local clustering performances than existing methods for both unit and cardinality-based cut-costs. Then, we show that general submodular cut-cost (recall that HFD is the only method that applies to this setting) can be necessary for capturing complex high-order relations in the data, improving F1 scores by up to 20% for local clustering and providing the only meaningful result for node ranking. Because of limited space, in the appendix we show additional local clustering experiments on two additional datasets, where our method improves F1 scores by 8% on average for 13 different target clusters.

**Datasets.** We provide basic information about the datasets used in our experiments. Complete descriptions are provided in the appendix. *Amazon-reviews* ($|V| = 2,268,264$, $|E| = 4,285,363$) [13,47]. In this hypergraph each node represents a product. A set of products are connected by a hyperedge if they are reviewed by the same person. We use product category labels as ground truth cluster identities. We consider all clusters of less than 10,000 nodes. *Trivago-clicks* ($|V| = 172,738$, $|E| = 233,202$) [45]. The nodes in this hypergraph are accommodations/hotels. A set of nodes are connected by a hyperedge if a user performed "click-out" action during the same browsing session. We use geographical locations as ground truth cluster identities. There are 160 such clusters, and we filter them using cluster size and conductance. *Florida Bay food network* ($|V| = 128$, $|E| = 141,233$) [8]. Nodes in this hypergraph correspond to different species or organisms that live in the Bay, and hyperedges correspond to transformed motifs (Figure 1) of the original dataset. Each species is labelled according its role in the food chain: producers, low-level consumers, high-level consumers.

**Methods and parameters.** We compare HFD with LH and ACL.[5] There is a heuristic nonlinear variant of LH which is shown to outperform linear LH in some cases [9]. Therefore we also compare with the same nonlinear variant considered in [9]. We denote the linear and nonlinear versions by LH-2.0 and LH-1.4, respectively. We set $\sigma = 0.0001$ for HFD and we set the parameters for LH-2.0, LH-1.4 and ACL as suggested by the authors [9]. More details on parameter choices appear in the appendix. We prefix methods that use unit and cardinality-based cut-costs by U- and C-, respectively.

**Experiments for unit and cardinality cut-costs.** For each target cluster in Amazon-reviews and Trivago-clicks, we run the methods multiple times, each time we use a different node as the singe seed node.[6] We report the median F1 scores of the output clusters in Table 1 and Table 2. For

---

[4]See the appendix for similar results when we fix the target cluster conductances around 0.2 and 0.25, respectively. These cover a reasonably wide range of scenarios in terms of the target conductance and illustrate the performance of algorithms for different levels of noise.

[5]We also tried a flow-improve method for hypergraphs [13], but the method was very slow in our experiments, so we only used it for small datasets. See appendix for results. The flow-improve method did not improve the performance of existing methods, therefore, we omitted it from comparisons on larger datasets.

[6]We show additional results using seed sets of more than one node in the appendix.

Amazon-reviews, we only compare the unit cut-cost because it is both shown in [9] and verified by our experiments that unit cut-cost is more suitable for this dataset. Observe that U-HFD obtains the highest F1 scores for nearly all clusters. In particular, U-HFD significantly outperforms other methods for clusters 12, 18, 24, where we see an increase in F1 score by up to 52%. For Trivago-clicks, C-HFD has the best performance for all but one clusters. Among the rest of all other methods, U-HFD has the second highest F1 scores for nearly all clusters. Moreover, observe that for each method (i.e., HFD, LH-2.0, LH-1.4), cardinality cut-cost leads to higher F1 than its unit cut-cost counterpart.

Table 1: F1 results for Amazon-reviews network

| Method | 1 | 2 | 3 | 12 | 15 | 17 | 18 | 24 | 25 |
|---|---|---|---|---|---|---|---|---|---|
| U-HFD | **0.45** | **0.09** | **0.65** | **0.92** | 0.04 | **0.10** | **0.80** | **0.81** | **0.09** |
| U-LH-2.0 | 0.23 | 0.07 | 0.23 | 0.29 | **0.05** | 0.06 | 0.21 | 0.28 | 0.05 |
| U-LH-1.4 | 0.23 | **0.09** | 0.35 | 0.40 | 0.00 | 0.07 | 0.31 | 0.35 | 0.06 |
| ACL | 0.23 | 0.07 | 0.22 | 0.25 | 0.04 | 0.05 | 0.17 | 0.20 | 0.04 |

Table 2: F1 results for Trivago-clicks network

| Method | KOR | ISL | PRI | UA-43 | VNM | HKG | MLT | GTM | UKR | EST |
|---|---|---|---|---|---|---|---|---|---|---|
| U-HFD | 0.75 | **0.99** | 0.89 | 0.85 | 0.28 | 0.82 | **0.98** | 0.94 | 0.60 | **0.94** |
| C-HFD | **0.76** | **0.99** | **0.95** | **0.94** | **0.32** | 0.80 | **0.98** | **0.97** | **0.68** | **0.94** |
| U-LH-2.0 | 0.70 | 0.86 | 0.79 | 0.70 | 0.24 | 0.92 | 0.88 | 0.82 | 0.50 | 0.90 |
| C-LH-2.0 | 0.73 | 0.90 | 0.84 | 0.78 | 0.27 | **0.94** | 0.96 | 0.88 | 0.51 | 0.83 |
| U-LH-1.4 | 0.69 | 0.84 | 0.80 | 0.75 | 0.28 | 0.87 | 0.92 | 0.83 | 0.47 | 0.90 |
| C-LH-1.4 | 0.71 | 0.88 | 0.84 | 0.78 | 0.27 | 0.88 | 0.93 | 0.85 | 0.50 | 0.85 |
| ACL | 0.65 | 0.84 | 0.75 | 0.68 | 0.23 | 0.90 | 0.83 | 0.69 | 0.50 | 0.88 |

**Experiments for general submodular cut-cost.** In order to understand the importance of specialized general submodular hypergraphs we study the node-ranking problem for the Florida Bay food network using hypergraph modelling shown in Figure 1. We compare HFD using unit (U-HFD, $\gamma_1 = \gamma_2 = 1$), cardinality-based (C-HFD, $\gamma_1 = 1/2$ and $\gamma_2 = 1$) and submodular (S-HFD, $\gamma_1 = 1/2$ and $\gamma_2 = 0$) cut-costs. Our goal is to search the most similar species of a queried species based on the food-network structure. Table 3 shows that S-HFD provides the only meaningful node ranking results. Intuitively, when $\gamma_2 = 0$, separating the preys $v_1, v_2$ from the predators $v_3, v_4$ incurs 0 cost. This encourages S-HFD to diffuse mass among preys or predators only and not to cross from a predator to a prey or vice versa. As a result, similar species receive similar amount of mass and thus are ranked similarly. In the local clustering setting, Table 3 compares HFD using different cut-costs. By exploiting specialized higher-order relations, S-HFD further improves F1 scores by up to 20% over U-HFD and C-HFD. This is not surprising, given the poor node-ranking results of other cut-costs. In the appendix we show another application of submodular cut-cost for node-ranking in an international oil trade network.

Table 3: Node-ranking and local clustering in Florida Bay food network using different cut-costs

| | Top-2 node-ranking results | | Clustering F1 | | |
|---|---|---|---|---|---|
| Method | Query: Raptors | Query: Gray Snapper | Prod. | Low | High |
| U-HFD | Epiphytic Gastropods, Detriti. Gastropods | Meiofauna, Epiphytic Gastropods | **0.69** | 0.47 | 0.64 |
| C-HFD | Epiphytic Gastropods, Detriti. Gastropods | Meiofauna, Epiphytic Gastropods | 0.67 | 0.47 | 0.64 |
| S-HFD | Gruiformes, Small Shorebirds | Snook, Mackerel | **0.69** | **0.62** | **0.84** |

## Acknowledgments and Disclosure of Funding

K.F. would like to acknowledge the support of the Natural Sciences and Engineering Research Council of Canada (NSERC). Cette recherche a été financée par le Conseil de recherches en sciences naturelles et en génie du Canada (CRSNG), [RGPIN-2019-04067, DGECR- 2019-00147]. P.L. is partially supported by the 2021 JP Morgan Faculty Award and the National Science Foundation (NSF) award HDR-2117997. S.Y. is partially supported by the Borealis AI 2021 Fellowship.

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
