# *Appendices for*: **Local Hyper-Flow Diffusion**

**Kimon Fountoulakis**
School of Computer Science
University of Waterloo
Waterloo, ON, Canada
kimon.fountoulakis@uwaterloo.ca

**Pan Li**
Department of Computer Science
Purdue University
West Lafayette, IN, United States
panli@purdue.edu

**Shenghao Yang**
School of Computer Science
University of Waterloo
Waterloo, ON, Canada
shenghao.yang@uwaterloo.ca

Outline of the Appendix:

- Appendix A contains supplementary material to Section 3 and Section 4 of the paper:
    - mathematical derivation of the dual diffusion problem;
    - proofs of Theorem 1 and Lemma 2.

- Appendix B contains supplementary material to Section 5 of the paper:
    - proof of Lemma 3;
    - convergence properties of Algorithm 1;
    - specialized algorithms for alternating minimization sub-problems of Algorithm 1.

- Appendix C contains supplementary material to Section 6 of the paper:
    - additional synthetic experiments using $k$-uniform hypergraph stochastic block model;
    - complete information about the real datasets considered in Section 6 of the paper;
    - experiments for local clustering using seed sets that contain more than one node;
    - experiments using 3 additional real datasets that are not discussed in the main paper;
    - parameter settings and implementation details.

## A    Approximation guarantee for local hypergraph clustering

In this section we prove a generalized and stronger version of Theorem 1 in the main paper, where the primal and dual diffusion problems are penalized by $\ell_p$-norm and $\ell_q$-norm, respectively, where $p \geq 2$ and $1/p + 1/q = 1$. Moreover, we consider a generic hypergraph $H = (V, E, \mathcal{W})$ with general submodular weights $\mathcal{W} = \{w_e, \vartheta_e\}_{e \in E}$ for any nonzero $\vartheta_e := \max_{S \subseteq e} w_e(S)$. *All claims in the main paper are therefore immediate special cases when $p = q = 2$ and $\vartheta_e = 1$ for all $e \in E$.*

Unless otherwise stated, we use the same notation as in the main paper. We generalize the definition of the degree of a node $v \in V$ as

$$d_v := \sum_{e \in E : v \in e} \vartheta_e.$$

Note that when $\vartheta_e = 1$ for all $e$, the above definition reduces to $d_v = |\{e \in E : v \in e\}|$, which is the number of hyperedges to which $v$ belongs to.

35th Conference on Neural Information Processing Systems (NeurIPS 2021).

Given $H = (V, E, \mathcal{W})$ where $\mathcal{W} = \{w_e, \vartheta_e\}_{e \in E}$, $p \geq 2$, and a hyperparameter $\sigma \geq 0$, our primal Hyper-Flow Diffusion (HFD) problem is written as

$$\min_{\phi \in \mathbb{R}_+^{|E|}, z \in \mathbb{R}_+^{|V|}} \quad \frac{1}{p} \sum_{e \in E} \vartheta_e \phi_e^p + \frac{\sigma}{p} \sum_{v \in V} d_v z_v^p$$

$$\text{s.t.} \quad \Delta - \sum_{e \in E} \vartheta_e r_e \leq d + \sigma D z \tag{A.1}$$

$$r_e \in \phi_e B_e, \ \forall e \in E$$

where

$$B_e := \{\rho_e \in \mathbb{R}^{|V|} \mid \rho_e(S) \leq w_e(S), \forall S \subseteq V, \text{ and } \rho_e(V) = w_e(V)\}$$

is the base polytope of $w_e$. The vector $m = \Delta - \sum_{e \in E} \vartheta_e r_e$ gives the net amount of mass after routing. Note that we multiply $r_e$ by $\vartheta_e$ because we have normalized $w_e$ by $\vartheta_e$ in its definition.

**Lemma A.1.** *The following optimization problem is dual to* (A.1):

$$\max_{x \in \mathbb{R}_+^{|V|}} \ (\Delta - d)^T x - \frac{1}{q} \sum_{e \in E} \vartheta_e f_e(x)^q - \frac{\sigma}{q} \sum_{v \in V} d_v x_v^q \tag{A.2}$$

*where $f_e(x) := \max_{\rho_e \in B_e} \rho_e^T x$ is the support function of base polytope $B_e$.*

*Proof.* Using convex conjugates, for $x \in \mathbb{R}_+^{|V|}$, we have

$$\frac{1}{q} f_e(x)^q = \max_{\phi_e \geq 0} \ \phi_e f_e(x) - \frac{1}{p} \phi_e^p, \ \forall e \in E, \tag{A.3a}$$

$$\frac{1}{q} x_v^q = \max_{z_v \geq 0} \ z_v x_v - \frac{1}{p} z_v^p, \ \forall v \in V. \tag{A.3b}$$

Apply the definition of $f_e(x)$, we can write (A.3a) as

$$\frac{1}{q} f_e(x)^q = \max_{\phi_e \geq 0} \ \phi_e f_e(x) - \frac{1}{p} \phi_e^p = \max_{\phi_e \geq 0, r_e \in \phi_e B_e} r_e^T x - \frac{1}{p} \phi_e^p.$$

Therefore,

$$\max_{x \in \mathbb{R}_+^{|V|}} \ (\Delta - d)^T x - \frac{1}{q} \sum_{e \in E} \vartheta_e f_e(x)^q - \frac{\sigma}{q} \sum_{v \in V} d_v x_v^q$$

$$= \max_{x \in \mathbb{R}_+^{|V|}} \ (\Delta - d)^T x - \sum_{e \in E} \vartheta_e \left( \max_{\phi_e \geq 0, r_e \in \phi_e B_e} r_e^T x - \frac{1}{p} \phi_e^p \right) - \sigma \sum_{v \in V} d_v \left( \max_{z_v \geq 0} z_v x_v - \frac{1}{p} z_v^p \right)$$

$$= \max_{x \in \mathbb{R}_+^{|V|}} \ (\Delta - d)^T x + \min_{\substack{\phi \in \mathbb{R}_+^{|E|} \\ r_e \in \phi_e B_e, \forall e \in E}} \sum_{e \in E} \left( \frac{1}{p} \vartheta_e \phi_e^p - \vartheta_e r_e^T x \right) + \min_{z \in \mathbb{R}_+^{|V|}} \sigma \sum_{v \in V} \left( \frac{1}{p} d_v z_v^p - d_v z_v x_v \right)$$

$$= \min_{\substack{\phi \in \mathbb{R}_+^{|E|}, z \in \mathbb{R}_+^{|V|} \\ r_e \in \phi_e B_e, \forall e \in E}} \frac{1}{p} \sum_{e \in E} \vartheta_e \phi_e^p + \frac{\sigma}{p} \sum_{v \in V} d_v z_v^p + \max_{x \in \mathbb{R}_+^{|V|}} \left( (\Delta - d)^T x - \sum_{e \in E} \vartheta_e r_e^T x - \sigma \sum_{v \in V} d_v z_v x_v \right)$$

$$= \min_{\substack{\phi \in \mathbb{R}_+^{|E|}, z \in \mathbb{R}_+^{|V|} \\ r_e \in \phi_e B_e, \forall e \in E}} \frac{1}{p} \sum_{e \in E} \vartheta_e \phi_e^p + \frac{\sigma}{p} \sum_{v \in V} d_v z_v^p \quad \text{s.t.} \quad \Delta - d - \sum_{e \in E} \vartheta_e r_e - \sigma D z \leq 0.$$

In the above derivations, we may exchange the order of minimization and maximization and arrive at the second last equality, due to Proposition 2.2, Chapter VI, in [1]. The last equality follows from

$$\max_{x \in \mathbb{R}_+^{|V|}} \left( (\Delta - d)^T x - \sum_{e \in E} \vartheta_e r_e^T x - \sigma \sum_{v \in V} d_v z_v x_v \right) = \begin{cases} 0, & \text{if } \Delta - d - \sum_{e \in E} \vartheta_e r_e - \sigma D z \leq 0, \\ +\infty, & \text{otherwise.} \end{cases}$$

$\square$

**Notation.** For the rest of this section, we reserve the notation $(\hat{\phi}, \hat{z})$ and $\hat{x}$ for optimal solutions of (A.1) and (A.2) respectively. If $\sigma = 0$, we simply treat $\hat{z} = 0$.

The next lemma relates primal and dual optimal solutions. We make frequent use of this relation throughout our discussion.

**Lemma A.2.** *We have that $\hat{\phi}_e^p = f_e(\hat{x})^q$ for all $e \in E$. Moreover, if $\sigma > 0$, then $\hat{z}_v^p = \hat{x}_v^q$ for all $v \in V$.*

*Proof.* Given $\hat{x}$ an optimal solution to (A.2), it follows directly from (A.3) and strong duality that $(\hat{\phi}, \hat{z})$ must satisfy, for each $e \in E$ and $v \in V$,

$$\hat{\phi}_e = f(\hat{x})^{q-1} = \operatorname*{argmax}_{\phi_e \geq 0} \; \phi_e f_e(\hat{x}) - \frac{1}{p}\phi_e^p \quad \text{and} \quad \hat{z}_v = \hat{x}_v^{q-1} = \operatorname*{argmax}_{z_v \geq 0} \; z_v \hat{x}_v - \frac{1}{p}z_v^p.$$

$\square$

**Diffusion setup.** Recall that we pick a scalar $\delta$ and set the source $\Delta$ as

$$\Delta_v = \begin{cases} \delta d_v, & \text{if } v \in S, \\ 0, & \text{otherwise.} \end{cases} \tag{A.4}$$

For convenience we restate the assumptions in the following.

**Assumption 1.** $\operatorname{vol}(S \cap C) \geq \alpha\operatorname{vol}(C)$ *and* $\operatorname{vol}(S \cap C) \geq \beta\operatorname{vol}(S)$ *for some* $\alpha, \beta \in (0, 1]$.

**Assumption 2.** *The source mass $\Delta$ as specified in (A.4) satisfies $\delta = 3/\alpha$, which gives $\Delta(C) \geq 3\operatorname{vol}(C)$.*

**Assumption 3.** $\sigma$ *satisfies* $0 \leq \sigma \leq \beta\Phi(C)/3$.

## A.1 Technical lemmas

In this subsection we state and prove some technical lemmas that will be used for the main proof in the next subsection.

The following lemma characterizes the maximizers of the support function for a base polytope.

**Lemma A.3** (Proposition 4.2 in [2]). *Let $w$ be a submodular function such that $w(\emptyset) = 0$. Let $x \in \mathbb{R}^{|V|}$, with unique values $a_1 > \cdots > a_m$, taken at sets $A_1, \ldots, A_m$ (i.e., $V = A_1 \cup \cdots \cup A_m$ and $\forall i \in \{1, \ldots, m\}, \forall v \in A_i, x_v = a_v$). Let $B$ be the associated base polytope. Then $\rho \in B$ is optimal for $\max_{\rho \in B} \rho^T x$ if and only if for all $i = 1, \ldots, m, \rho(A_1 \cup \cdots \cup A_i) = w(A_1 \cup \cdots \cup A_i)$.*

Recall that $(\hat{\phi}, \hat{z})$ and $\hat{x}$ denote the optimal solutions of (A.1) and (A.2) respectively. We start with a lemma on the locality of the optimal solutions.

**Lemma A.4** (Lemma 2 in the main paper). *We have*

$$\sum_{e \in \operatorname{supp}(\hat{\phi})} \vartheta_e \; = \; \operatorname{vol}(\operatorname{supp}(\hat{x})) \; \leq \; \|\Delta\|_1.$$

*Moreover, if $\sigma > 0$, then $\operatorname{vol}(\operatorname{supp}(\hat{z})) = \operatorname{vol}(\operatorname{supp}(\hat{x}))$.*

*Proof.* To see the first inequality, note that if $\hat{x}_v = 0$ for every $v \in e$ for some $e$, then $f_e(\hat{x}) = 0$. By Lemma A.2, this means $\hat{\phi}_e = 0$. Thus, $\hat{\phi}_e \neq 0$ only if there is some $v \in e$ such that $\hat{x}_v \neq 0$. Therefore, we have that

$$\sum_{e \in \operatorname{supp}(\hat{\phi})} \vartheta_e \leq \sum_{v \in \operatorname{supp}(\hat{x})} \sum_{e \in E: v \in e} \vartheta_e = \sum_{v \in \operatorname{supp}(\hat{x})} d_v = \operatorname{vol}(\operatorname{supp}(\hat{x})).$$

To see the last inequality, note that, by the first order optimality condition of (A.2), if $\hat{x}_v \neq 0$ then we must have

$$\Delta_v - d_v = \sum_{e \in E} \vartheta_e f_e(\hat{x})^{q-1}\hat{\rho}_{e,v} + \sigma d_v \hat{x}_v^{q-1}, \quad \text{for some } \hat{\rho}_e \in \partial f_e(\hat{x}) = \operatorname*{argmax}_{\rho_e \in B_e} \rho_e^T \hat{x}. \tag{A.5}$$

Denote $N := \mathrm{supp}(\hat{x})$ and $E[N] := \{e \in E \mid v \in N \text{ for all } v \in e\}$. Note that $E[N] \cap \partial N = \emptyset$, and $E[N] \cup \partial N = \{e \in E \mid v \in N \text{ for some } v \in e\}$, that is, $E[N] \cup \partial N$ contain all hyperedges that are incident to some node in $N$. Moreover, we have that for any $\hat{\rho}_e \in \mathrm{argmax}_{\rho_e \in B_e} \rho_e^T \hat{x}$,

$$\sum_{v \in N} \hat{\rho}_{e,v} = \hat{\rho}_e(N) = \begin{cases} w_e(N), & \text{if } e \in \partial N, \\ 0, & \text{if } e \in E[N], \end{cases}$$

where $\hat{\rho}_e(N) = w_e(N)$ for $e \in \partial N$ follows from Lemma A.3, since $\hat{x}_v > 0$ for $v \in N$ and $\hat{x}_v = 0$ for $v \notin N$. The equality $\hat{\rho}_e(N) = 0$ for $e \in E[N]$ follows from $\hat{\rho}_e(N) = \hat{\rho}_e(e) = 0$ because $e \subseteq N$ and $\hat{\rho}_{e,v} = 0$ for all $v \notin e$.

Taking sums over $v \in N$ on both sides of equation (A.5) we obtain

$$
\begin{aligned}
\Delta(N) - \mathrm{vol}(N) &= \sum_{v \in N} \sum_{e \in E} \vartheta_e f_e(\hat{x})^{q-1} \hat{\rho}_{e,v} + \sum_{v \in N} \sigma d_v \hat{x}_v^{q-1} \\
&= \sum_{v \in N} \sum_{e \in E[N]} \vartheta_e f_e(\hat{x})^{q-1} \hat{\rho}_{e,v} + \sum_{v \in N} \sum_{e \in \partial N} \vartheta_e f_e(\hat{x})^{q-1} \hat{\rho}_{e,v} + \sum_{v \in N} \sigma d_v \hat{x}_v^{q-1} \\
&= \sum_{e \in E[N]} \vartheta_e f_e(\hat{x})^{q-1} \sum_{v \in N} \hat{\rho}_{e,v} + \sum_{e \in \partial N} \vartheta_e f_e(\hat{x})^{q-1} \sum_{v \in N} \hat{\rho}_{e,v} + \sum_{v \in N} \sigma d_v \hat{x}_v^{q-1} \\
&= 0 + \sum_{e \in \partial N} \vartheta_e f_e(\hat{x})^{q-1} w_e(N) + \sum_{v \in N} \sigma d_v \hat{x}_v^{q-1} \\
&\geq 0.
\end{aligned}
$$

The second equality follows from $\hat{\rho}_{e,v} = 0$ for all $v \notin e$. This proves $\mathrm{vol}(\mathrm{supp}(\hat{x})) \leq \Delta(\mathrm{supp}(\hat{x})) \leq \|\Delta\|_1$.

Finally, if $\sigma > 0$, then $\mathrm{vol}(\mathrm{supp}(\hat{z})) = \mathrm{vol}(\mathrm{supp}(\hat{x}))$ follows from Lemma A.2 that $\hat{z}^p = \hat{x}^q$ for all $v \in V$. $\qquad\square$

The following inequality is a special case of Hölder's inequality for degree-weighted norms. It will become useful later.

**Lemma A.5.** *For $x \in \mathbb{R}^{|V|}$ and $p > 1$ we have that*

$$\left( \sum_{v \in V} d_v |x_v| \right)^p \leq \mathrm{vol}(\mathrm{supp}(x))^{p-1} \sum_{v \in V} d_v |x_v|^p.$$

*Proof.* Let $q = p/(p-1)$. Apply Hölder's inequality we have

$$
\sum_{v \in V} d_v |x_v| = \sum_{v \in \mathrm{supp}(x)} |d_v^{1/q}||d_v^{1/p} x_v| \leq \left( \sum_{v \in \mathrm{supp}(x)} d_v \right)^{1/q} \left( \sum_{v \in \mathrm{supp}(x)} d_v |x_v|^p \right)^{1/p}
$$
$$
= \mathrm{vol}(\mathrm{supp}(x))^{1/q} \left( \sum_{v \in V} d_v |x_v|^p \right)^{1/p}.
$$

$\qquad\square$

**Lemma A.6** (Lemma I.2 in [3]). *For any $x \in \mathbb{R}_+^{|V|} \setminus \{0\}$ and $q \geq 1$, one has*

$$\frac{\sum_{e \in E} \vartheta_e f_e(x)^q}{\sum_{v \in V} d_v x_v^q} \geq \frac{c(x)^q}{q^q},$$

*where*

$$c(x) := \min_{h \geq 0} \frac{\mathrm{vol}(\partial\{v \in V \mid x_v^q > h\})}{\mathrm{vol}(\{v \in V \mid x_v^q > h\})} = \min_{h \geq 0} \frac{\mathrm{vol}(\partial\{v \in V \mid x_v > h\})}{\mathrm{vol}(\{v \in V \mid x_v > h\})}.$$

Recall that the objective function of our primal diffusion problem (A.1) consists of two parts. The first part is $\sum_{e \in E} \vartheta_e \phi_e^p$ and it penalizes the cost of flow routing, the second part is $\sum_{v \in V} d_v z_v^p$ and

it penalizes the cost of excess mass. An immediate consequence of Lemma A.6 is the inequality in Lemma A.7 that relates the cost of optimal flow routing $\sum_{e \in E} \vartheta_e \hat{\phi}_e^p$ and the cost of excess mass $\sum_{v \in V} d_v \hat{z}_v^p$ at optimality.

For $h > 0$, recall that the sweep sets are defined as $S_h := \{v \in V | \hat{x}_v \geq h\}$.

Let $\hat{h} \in \mathrm{argmin}_{h>0} \Phi(S_h)$ and denote $\hat{S} = S_{\hat{h}}$. That is, $\hat{S} = S_h$ for some $h > 0$ and $\Phi(\hat{S}) \leq \Phi(S_h)$ for all $h > 0$.

**Lemma A.7.** *For $p > 1$ and $q = p/(p-1)$ we have that*

$$\sum_{e \in E} \vartheta_e \hat{\phi}_e^p \geq \left( \frac{\Phi(\hat{S})}{q} \right)^q \sum_{v \in V} d_v \hat{z}_v^p.$$

*Proof.* By Lemma A.2,

$$\sum_{e \in E} \vartheta_e \hat{\phi}_e^p = \sum_{e \in E} \vartheta_e f_e(\hat{x})^q \quad \text{and} \quad \sum_{v \in V} d_v \hat{z}_v^p = \sum_{v \in V} d_v \hat{x}_v^q,$$

and the result follows from applying Lemma A.6. $\qquad\square$

Given a vector $a \in \mathbb{R}^{|V|}$ and a set $S \subseteq V$, recall that we write $a(S) = \sum_{v \in S} a_v$. This actually defines a modular set-function $a$ taking input on subsets of $V$. The Lovász extension of modular function $a$ is simply $f(x) = a^T x$ [2]. Since all modular functions are also submodular, we arrive at the following lemma that follows from a classical property of the Choquet integral/Lovász extension.

**Lemma A.8.** *We have that*

$$\Delta^T \hat{x} = \int_{h=0}^{+\infty} \Delta(S_h) dh, \quad d^T \hat{x} = \int_{h=0}^{+\infty} \mathrm{vol}(S_h) dh, \quad f_e(\hat{x}) = \int_{h=0}^{+\infty} w_e(S_h) dh.$$

*Proof.* Recall that, by definition, $\mathrm{vol}(S) = d(S)$ where $d$ is the degree vector. $\Delta$ and $d$ are modular functions on $2^V$ and $w_e$ is a submodular function on $2^V$. The Lovász extension of $\Delta$ and $d$ are $\Delta^T x$ and $d^T x$, respectively. The Lovász extension of $w_e$ is $f_e(x)$. The results then follow immediately from representing the Lovász extensions using Choquet integrals. See, e.g., Proposition 3.1 in [2]. $\quad\square$

## A.2 Proof of Theorem 1 in the main paper

We restate the theorem below with respect to the general formulations (A.1) and (A.2) for any $p \geq 2$ and $q = p/(p-1)$.

Let us recall that the sweep sets are defined as $S_h := \{v \in V | \hat{x}_v \geq h\}$.

**Theorem A.9.** *Under Assumptions 1, 2, 3, for some $h > 0$ we have that*

$$\Phi(S_h) \leq O\left( \frac{\Phi(C)^{1/q}}{\alpha \beta} \right).$$

Recall that $\hat{S}$ is such that $\hat{S} = S_h$ for some $h > 0$ and $\Phi(\hat{S}) \leq \Phi(S_h)$ for all $h > 0$. We will assume without loss of generality that $\Phi(C) \leq (\Phi(\hat{S})/q)^q$, as otherwise $\Phi(\hat{S}) < q\Phi(C)^{1/q}$ and the statement in Theorem A.9 already holds.

Denote $\hat{\nu} := \sum_{e \in E} \vartheta_e \hat{\phi}_e^p$, the cost of optimal flow routing. The following claim states that $\hat{\nu}$ must be large.

**Claim A.1.** $\hat{\nu} \geq \mathrm{vol}(C)^p / \mathrm{vol}(\partial C)^{p-1}$.

*Proof.* The proof of this claim follows from a case analysis on the total amount of excess mass $\sigma \sum_{v \in V} d_v \hat{z}_v$ at optimality. Intuitively, if the excess is small, then naturally there must be a large amount of flow in order to satisfy the primal constraint; if the excess is large, then Lemma A.7 and Lemma A.5 guarantee that flow is also large. We give details below.

Suppose that $\sigma \sum_{v \in V} d_v \hat{z}_v < \mathrm{vol}(C)$. Note that this also includes the case where $\sigma = 0$. By Assumption 2 there is at least $\Delta(C) \geq 3\mathrm{vol}(C)$ amount of source mass trapped in $C$ at the beginning.

Moreover, the primal constraint enforces the nodes in $C$ can settle at most $\sum_{v \in C}(d_v + \sigma d_v \hat{z}_v) \leq$ $\text{vol}(C) + \sum_{v \in V} \sigma d_v \hat{z}_v < 2\text{vol}(C)$ amount of mass. Therefore, the remaining at least $\text{vol}(C)$ amount of mass needs to get out of $C$ using the hyperedges in $\partial C$. That is, the net amount of mass that moves from $C$ to $V \setminus C$ satisfies $\sum_{e \in \partial C} \vartheta_e \hat{r}_e(C) \geq \text{vol}(C)$. We focus on the cost of $\hat{\phi}$ restricted to these hyperedges along. It is easy to see that

$$
\sum_{e \in \partial C} \vartheta_e \hat{\phi}_e^p \geq \min_{\phi \in \mathbb{R}_+^{|\partial C|}} \sum_{e \in \partial C} \vartheta_e \phi_e^p \text{ subject to } \hat{r}_e \in \phi_e B_e, \ \forall e \in \partial C \tag{A.6a}
$$

$$
\geq \min_{\phi \in \mathbb{R}_+^{|\partial C|}} \sum_{e \in \partial C} \vartheta_e \phi_e^p \text{ subject to } \sum_{e \in \partial C} \vartheta_e \hat{r}_e(C) \leq \sum_{e \in \partial C} \vartheta_e \phi_e w_e(C) \tag{A.6b}
$$

$$
\geq \min_{\phi \in \mathbb{R}_+^{|\partial C|}} \sum_{e \in \partial C} \vartheta_e \phi_e^p \text{ subject to } \text{vol}(C) \leq \sum_{e \in \partial C} \vartheta_e \phi_e w_e(C). \tag{A.6c}
$$

The first inequality follows because $\hat{\phi}$ restricted to $\partial C$ is a feasible solution in problem (A.6a). The second inequality follows because $\hat{r}_e \in \phi_e B_e$ implies $\hat{r}_e(C) \leq \phi_e w_e(C)$, therefore every feasible solution for (A.6a) is also a feasible solution for (A.6b). The third inequality follows because $\text{vol}(C) \leq \sum_{e \in E} \vartheta_e \hat{r}_e(C)$. Let $\bar{\phi} \in \mathbb{R}_+^{|\partial C|}$ be an optimal solution of problem (A.6c). The optimality condition of (A.6c) is given by (we may assume the $p$ factor in the gradient of $\sum_{e \in \partial C} \vartheta_e \phi_e^p$ is absorbed into multipliers $\lambda$ and $\eta_e$)

$$
\begin{aligned}
&\vartheta_e \phi_e^{p-1} - \lambda \vartheta_e w_e(C) - \eta_e = 0, \ \forall e \in \partial C \\
&\phi_e \geq 0, \ \eta_e \geq 0, \ \phi_e \eta_e = 0, \ \forall e \in \partial C \\
&\text{vol}(C) \leq \sum_{e \in \partial C} \vartheta_e \phi_e w_e(C) \\
&\lambda \geq 0, \ \lambda \left( \text{vol}(C) - \sum_{e \in \partial C} \vartheta_e \phi_e w_e(C) \right) = 0.
\end{aligned} \tag{A.7}
$$

If $\lambda = 0$, then the conditions in (A.7) imply that $\vartheta_e \phi_e^{p-1} = \eta_e$, but then by complimentary slackness we would obtain $\phi_e = \eta_e = 0$ for all $e \in \partial C$ which will violate feasibility. Therefore we must have $\lambda > 0$, and consequently, we have that

$$
\sum_{e \in \partial C} \vartheta_e \bar{\phi}_e w_e(C) = \text{vol}(C). \tag{A.8}
$$

Moreover, the conditions in (A.7) imply that for $e \in \partial C$, $\bar{\phi}_e = 0$ if and only if $w_e(C) = 0$, and hence we have that

$$
\vartheta_e \bar{\phi}_e^{p-1} = \lambda \vartheta_e w_e(C), \ \forall e \in \partial C. \tag{A.9}
$$

Rearrange (A.9) we get

$$
\bar{\phi}_e w_e(C) = \lambda^{1/(p-1)} w_e(C)^{p/(p-1)}, \ \forall e \in \partial C.
$$

Substitute the above into (A.8),

$$
\text{vol}(C) = \sum_{e \in \partial C} \vartheta_e \bar{\phi}_e w_e(C) = \sum_{e \in \partial C} \vartheta_e \lambda^{1/(p-1)} w_e(C)^{p/(p-1)},
$$

this gives

$$
\lambda^{1/(p-1)} = \frac{\text{vol}(C)}{\sum_{e \in \partial C} \vartheta_e w_e(C)^{p/(p-1)}}.
$$

Therefore, the solution $\bar{\phi}$ for (A.6c) is give by

$$
\bar{\phi}_e = \lambda^{1/(p-1)} w_e(C)^{1/(p-1)} = \frac{\text{vol}(C) w_e(C)^{1/(p-1)}}{\sum_{e' \in \partial C} \vartheta_{e'} w_{e'}(C)^{p/(p-1)}}, \quad \forall e \in \partial C,
$$

and hence,

$$\hat{\nu} = \sum_{e \in E} \vartheta_e \hat{\phi}_e^p \geq \sum_{e \in \partial C} \vartheta_e \hat{\phi}_e^p \geq \sum_{e \in \partial C} \vartheta_e \bar{\phi}_e^p = \sum_{e \in \partial C} \vartheta_e \frac{\text{vol}(C)^p w_e(C)^{p/(p-1)}}{\left( \sum_{e' \in \partial C} \vartheta_{e'} w_{e'}(C)^{p/(p-1)} \right)^p}$$

$$= \frac{\text{vol}(C)^p \sum_{e \in \partial C} \vartheta_e w_e(C)^{p/(p-1)}}{\left( \sum_{e' \in \partial C} \vartheta_{e'} w_{e'}(C)^{p/(p-1)} \right)^p}$$

$$= \frac{\text{vol}(C)^p}{\left( \sum_{e' \in \partial C} \vartheta_{e'} w_{e'}(C)^{p/(p-1)} \right)^{p-1}}$$

$$\geq \frac{\text{vol}(C)^p}{\left( \sum_{e' \in \partial C} \vartheta_{e'} w_{e'}(C) \right)^{p-1}}$$

where the last inequality follows because $w_e(C) \in [0, 1]$ and $p \geq 1$.

Suppose now that $\sigma \sum_{v \in V} d_v \hat{z}_v \geq \text{vol}(C)$. Becase $\Phi(C) \leq (\Phi(\hat{S})/q)^q$ (recall that we assumed this without loss of generality), by Assumption 3, we know that $\sigma < (\phi(\hat{S})/q)^q$. Therefore,

$$\hat{\nu} = \sum_{e \in E} \vartheta_e \hat{\phi}_e^p \overset{(i)}{\geq} \sigma \sum_{v \in V} d_v \hat{z}_v^p$$

$$\overset{(ii)}{\geq} \frac{\sigma \left( \sum_{v \in V} d_v \hat{z}_v \right)^p}{\text{vol}(\text{supp}(\hat{z}))^{p-1}}$$

$$\overset{(iii)}{\geq} \frac{\sigma^p \left( \sum_{v \in V} d_v \hat{z}_v \right)^p}{\sigma^{p-1}(3\text{vol}(C)/\beta)^{p-1}}$$

$$\overset{(iv)}{\geq} \frac{\sigma^p \left( \sum_{v \in V} d_v \hat{z}_v \right)^p}{\text{vol}(\partial C)^{p-1}}$$

$$\overset{(v)}{\geq} \frac{\text{vol}(C)^p}{\text{vol}(\partial C)^{p-1}}.$$

$(i)$ is due to Lemma A.6. $(ii)$ is due to Lemma A.5. $(iii)$ is due to Lemma A.4 that $\text{vol}(\text{supp}(\hat{z})) \leq \|\Delta\|_1$ and Assumption 2 that $\|\Delta\|_1 \leq 3\text{vol}(c)/\beta$, so $\text{vol}(\text{supp}(\hat{z}))^{p-1} \leq (3\text{vol}(C)/\beta)^{p-1}$ for $p \geq 1$. $(iv)$ is due to Assumption 3 that $\sigma \leq \frac{\beta \text{vol}(\partial C)}{3\text{vol}(C)}$, so $(3\sigma\text{vol}(C)/\beta)^{p-1} \leq \text{vol}(\partial C)^{p-1}$ for $p \geq 1$. $(v)$ is due to the assumption that $\sigma \sum_{v \in V} d_v \hat{z}_v \geq \text{vol}(C)$. $\qquad \square$

To connect $\Phi(S_h)$ with $\Phi(C)$, we define the *length* of a hyperedge $e \in E$ as

$$\hat{l}(e) := \begin{cases} \max(1/\text{vol}(C)^{1/q}, f_e(\hat{x})/\hat{\nu}^{1/q}), & \text{if } f_e(\hat{x}) > 0, \\ 0, & \text{otherwise.} \end{cases}$$

The next claim follows from simple algebraic computations and the locality of solutions in Lemma A.4.

**Claim A.2.** $\sum_{e \in E} \vartheta_e f_e(\hat{x}) \hat{l}(e)^{q-1} \leq 4\hat{\nu}^{1/q}/\beta$.

*Proof.* For $e \in E$, define $l(e) := f_e(\hat{x})/\hat{\nu}^{1/q}$. Then $l(e) \leq \hat{l}(e)$. Moreover,

$$\sum_{e: l(e) < \hat{l}(e)} \vartheta_e \leq \sum_{e \in \text{supp}(\hat{\phi})} \vartheta_e \leq \text{vol}(\text{supp}(\hat{x})) \leq \|\Delta\|_1 = \frac{3}{\alpha}\text{vol}(S) \leq \frac{3}{\beta}\text{vol}(C).$$

The first inequality follows from that $l(e) < \hat{l}(e)$ only if $l(e) \neq 0$, and by Lemma A.2, $l(e) \neq 0$ if and only if $\hat{\phi}_e \neq 0$. The second and the third inequalities are due to Lemma A.4. The second to last equality follows from the diffusion setting (A.4) and Assumption 2 that $\delta = 3/\alpha$. The last inequality

follows from Assumption 1. Therefore,

$$\sum_{e\in E}\vartheta_e f_e(\hat{x})\hat{l}(e)^{q-1} = \sum_{e:l(e)=\hat{l}(e)}\vartheta_e f_e(\hat{x})\frac{f_e(\hat{x})^{q-1}}{\hat{\nu}^{(q-1)/q}} + \sum_{e:l(e)<\hat{l}(e)}\vartheta_e f_e(\hat{x})\frac{1}{\text{vol}(C)^{(q-1)/q}}$$

$$\le \sum_{e:l(e)=\hat{l}(e)}\vartheta_e f_e(\hat{x})\frac{f_e(\hat{x})^{q-1}}{\hat{\nu}^{(q-1)/q}} + \sum_{e:l(e)<\hat{l}(e)}\vartheta_e \frac{\hat{\nu}^{1/q}}{\text{vol}(C)^{1/q}}\frac{1}{\text{vol}(C)^{(q-1)/q}}$$

$$= \frac{1}{\hat{\nu}^{(q-1)/q}}\sum_{e:l(e)=\hat{l}(e)}\vartheta_e f_e(\hat{x})^q + \frac{\hat{\nu}^{1/q}}{\text{vol}(C)}\sum_{e:l(e)<\hat{l}(e)}\vartheta_e$$

$$\le \frac{1}{\hat{\nu}^{(q-1)/q}}\sum_{e\in E}\vartheta_e f_e(\hat{x})^q + \frac{\hat{\nu}^{1/q}}{\text{vol}(C)}\frac{3\text{vol}(C)}{\beta}$$

$$= \frac{\hat{\nu}}{\hat{\nu}^{(q-1)/q}} + \frac{3\hat{\nu}^{1/q}}{\beta}$$

$$\le \frac{4\hat{\nu}^{1/q}}{\beta}$$

where the last equality follows from Lemma A.2 that $\hat{\nu} = \sum_{e\in E}\vartheta_e\hat{\phi}_e^p = \sum_{e\in E}\vartheta_e f_e(\hat{x})^q$. $\qquad\square$

By the strong duality between (A.1) and (A.2), we know that

$$(\Delta - d)^T\hat{x} - \frac{1}{q}\sum_{e\in E}\vartheta_e f_e(\hat{x})^q - \frac{\sigma}{q}\sum_{v\in V}d_v\hat{x}_v^q = \frac{1}{p}\sum_{e\in E}\vartheta_e\hat{\phi}_e^p + \frac{\sigma}{p}\sum_{v\in V}d_v\hat{z}_v^p.$$

Hence, by Lemma A.2, we get

$$(\Delta - d)^T\hat{x} \ge \frac{1}{q}\sum_{e\in E}\vartheta_e f_e(\hat{x})^q + \frac{1}{p}\sum_{e\in E}\vartheta_e\hat{\phi}_e^p = \sum_{e\in E}\vartheta_e\hat{\phi}_e^p = \hat{\nu}.$$

It then follows that

$$\frac{\sum_{e\in E}\vartheta_e f_e(\hat{x})\hat{l}(e)^{q-1}}{(\Delta - d)^T\hat{x}} \le \frac{\sum_{e\in E}\vartheta_e f_e(\hat{x})\hat{l}(e)^{q-1}}{\hat{\nu}} \overset{(i)}{\le} \frac{4\hat{\nu}^{1/q}}{\beta\hat{\nu}} = \frac{4}{\beta\hat{\nu}^{1/p}} \overset{(ii)}{\le} \frac{4\text{vol}(\partial C)^{1/q}}{\beta\text{vol}(C)}, \quad \text{(A.10)}$$

where $(i)$ is follows from Claim A.2 and $(ii)$ follows from Claim A.1.

We can write the left-most ratio in (A.10) in its integral form, as follows. By Lemma A.8, we have

$$(\Delta - d)^T\hat{x} = \int_{h=0}^{\infty}(\Delta(S_h) - \text{vol}(S_h))dh,$$

and

$$\sum_{e\in E}\vartheta_e f_e(\hat{x})\hat{l}(e)^{q-1} = \sum_{e\in E}\vartheta_e\int_{h=0}^{\infty}w_e(S_h)dh\,\hat{l}(e)^{q-1}$$

$$= \int_{h=0}^{\infty}\sum_{e\in E}\vartheta_e w_e(S_h)\hat{l}(e)^{q-1}dh$$

$$= \int_{h=0}^{\infty}\sum_{e\in\partial S_h}\vartheta_e w_e(S_h)\hat{l}(e)^{q-1}dh,$$

where the last equality follows from the fact that $w_e(S_h) = 0$ for $e \notin \partial S_h$. Therefore, we get

$$\int_{h=0}^{\infty}\frac{\sum_{e\in\partial S_h}\vartheta_e w_e(S_h)\hat{l}(e)^{q-1}}{\Delta(S_h) - \text{vol}(S_h)}dh \le \frac{4\text{vol}(\partial C)^{1/q}}{\beta\text{vol}(C)},$$

which means that there exists $h > 0$ such that

$$\frac{\sum_{e\in\partial S_h}\vartheta_e w_e(S_h)\hat{l}(e)^{q-1}}{\Delta(S_h) - \text{vol}(S_h)} \le \frac{4\text{vol}(\partial C)^{1/q}}{\beta\text{vol}(C)}. \quad\text{(A.11)}$$

Finally, we connect the left hand side in inequality (A.11) to the conductance of $S_h$. For the denominator, by Assumption 2, we have

$$\Delta(S_h) - \text{vol}(S_h) \leq \frac{3}{\alpha}\text{vol}(S_h). \tag{A.12}$$

For the numerator, every hyperedge $e \in \partial S_h$ must contain some $u, v \in e$ such that $\hat{x}_u \neq \hat{x}_v$, thus $f_e(\hat{x}) > 0$, which means $\hat{l}(e) \geq 1/\text{vol}(C)^{1/q}$. This gives

$$\sum_{e \in \partial S_h} \vartheta_e w_e(S_h)\hat{l}(e)^{q-1} \geq \frac{\sum_{e \in \partial S_h} \vartheta_e w_e(S_h)}{\text{vol}(C)^{(q-1)/q}} = \frac{\text{vol}(\partial S_h)}{\text{vol}(C)^{(q-1)/q}}. \tag{A.13}$$

Put (A.11), (A.12) and (A.13) together, there exists $h > 0$ such that

$$\Phi(S_h) = \frac{\text{vol}(\partial S_h)}{\text{vol}(S_h)} \leq \frac{12\text{vol}(\partial C)^{1/q}}{\alpha\beta\text{vol}(C)^{1/q}} = \frac{12\Phi(C)^{1/q}}{\alpha\beta}.$$

# B Optimization algorithm for HFD

In this section we give details on an Alternating Minimization (AM) algorithm [4] that solves the primal problem (A.1). In Algorithm B.1 we write the basic AM steps in a slightly more general form than what is given by Algorithm 1 in the main paper. The key observation is that the AM method provides a unified framework to solve HFD, when the objective function of the primal problem (A.1) is penalized by any $\ell_p$-norm for $p \geq 2$.

Let us remind the reader the definitions and notation that we will use. We consider a generic hypergraph $H = (V, E, \mathcal{W})$ where $\mathcal{W} = \{w_e, \vartheta_e\}_{e \in E}$ are submodular hyperedge weights. For each $e \in E$, we define a diagonal matrix $A_e \in \mathbb{R}^{|V| \times |V|}$ such that $[A_e]_{v,v} = 1$ if $v \in e$ and 0 otherwise. We use the notation $r \in \bigotimes_{e \in E} \mathbb{R}^{|V|}$ to represent a vector in the space $\mathbb{R}^{|V||E|}$, where each $r_e \in \mathbb{R}^{|V|}$ corresponds to a block in $r$ indexed by $e \in E$. For a vector $r_e \in \mathbb{R}^{|V|}$, $r_{e,v}$ is the entry in $r_e$ that corresponds to $v \in V$. For a vector $x \in \mathbb{R}^{|V|}$, $[x]_+ := \max\{x, 0\}$ where the maximum is taken entry-wise.

We denote $\mathcal{C} := \{(\phi, r) \in \mathbb{R}_+^{|E|} \times (\bigotimes_{e \in E} \mathbb{R}^{|V|}) \mid r_e \in \phi_e B_e, \ \forall e \in E\}$.

---
**Algorithm B.1** Alternating Minimization for HFD

**Initialization:**

$$\phi^{(0)} := 0, r^{(0)} := 0, s_e^{(0)} := D^{-1}A_e\left[\Delta - d\right]_+, \forall e \in E.$$

**For** $k = 0, 1, 2, \ldots$ **do:**

$$(\phi^{(k+1)}, r^{(k+1)}) := \underset{(\phi, r) \in \mathcal{C}}{\text{argmin}} \sum_{e \in E} \vartheta_e\left(\phi_e^p + \frac{1}{\sigma^{p-1}}\|s_e^{(k)} - r_e\|_p^p\right)$$

$$s^{(k+1)} := \underset{s}{\text{argmin}} \sum_{e \in E} \vartheta_e\|s_e - r_e^{(k+1)}\|_p^p, \quad \text{s.t. } \Delta - \sum_{e \in E} \vartheta_e s_e \leq d, \ s_{e,v} = 0, \forall v \notin e.$$

---

We will prove the equivalence between the primal diffusion problem (A.1) and its separable reformulation shortly, but let us start with a simple lemma that gives closed-form solution for one of the AM sub-problems.

**Lemma B.1.** *The optimal solution to the following problem*

$$\min_{s \in \bigotimes_{e \in E} \mathbb{R}^{|V|}} \sum_{e \in E} \vartheta_e\|s_e - r_e\|_p^p, \ \text{s.t. } \Delta - \sum_{e \in E} \vartheta_e s_e \leq d, \ s_{e,v} = 0, \forall v \notin e. \tag{B.1}$$

*is given by*

$$s_e^* = r_e + A_e D^{-1}\left[\Delta - \sum_{e' \in E} \vartheta_{e'} r_{e'} - d\right]_+, \ \forall e \in E. \tag{B.2}$$

*Proof.* Rewrite (B.1) as

$$\min_{s\in\bigotimes_{e\in E}\mathbb{R}^{|V|}}\sum_{v\in V}\sum_{e\in E}\vartheta_e|s_{e,v}-r_{e,v}|^p$$

$$\text{s.t.}\quad \Delta_v-\sum_{e\in E}\vartheta_e s_{e,v}\le d_v,\ \forall v\in V$$

$$s_{e,v}=0,\ \forall v\notin e.$$

Then it is immediate to see that (B.1) decomposes into $|V|$ sub-problems indexed by $v\in V$,

$$\min_{\xi_v\in\mathbb{R}^{|E_v|}}\sum_{e\in E_v}\vartheta_e|\xi_{v,e}-r_{e,v}|^p,\ \text{s.t. }\Delta_v-\sum_{e\in E_v}\vartheta_e\xi_{v,e}\le d_v, \tag{B.3}$$

where $E_v:=\{e\in E\mid v\in e\}$ is the set of hyperedges incident to $v$, and we use $\xi_{v,e}$ for the entry in $\xi_v$ that corresponds to $e\in E_v$. Let $\xi_v^*$ denote the optimal solution for (B.3). We have that $s_{e,v}^*=\xi_{v,e}^*$ if $v\in e$ and $s_{e,v}^*=0$ otherwise. Therefore, it suffices to find $\xi_v^*$ for $v\in V$. The optimality condition of (B.3) is given by

$$p\vartheta_e|\xi_{v,e}-r_{e,v}|^{p-1}\operatorname{sign}(\xi_{v,e}-r_{e,v})-\vartheta_e\lambda\ni 0,\ \forall e\in E_v,$$

$$\lambda\ge 0,\ \Delta_v-\sum_{e\in E_v}\vartheta_e\xi_{v,e}\le d_v,\ \lambda\Big(\Delta_v-\sum_{e\in E_v}\vartheta_e\xi_{v,e}-d_v\Big)=0,$$

where

$$\operatorname{sign}(a):=\left\{\begin{array}{ll}\{-1\}, & \text{if } a<0,\\ \{1\}, & \text{if } a>0,\\ [-1,1] & \text{if } a=0.\end{array}\right.$$

There are two cases about $\lambda$. We show that in both cases the solution given by (B.2) is optimal.

*Case 1.* If $\lambda>0$, then we must have that $p\vartheta_e|\xi_{v,e}-r_{e,v}|^{p-1}>0$ for all $e\in E_v$ (otherwise, the stationarity condition would be violated). This means that $p|\xi_{v,e}-r_{e,v}|^{p-1}=\lambda$ for all $e\in E_v$, that is, $\xi_{v,e_1}-r_{e_1,v}=\xi_{v,e_2}-r_{e_2,v}>0$ for every $e_1,e_2\in E_v$. Denote $t_v:=\xi_{v,e}-r_{e,v}$. Because $\lambda>0$, by complementarity we have

$$\Delta_v-\sum_{e\in E_v}\vartheta_e(t_v+r_{e,v})=\Delta_v-\sum_{e\in E_v}\vartheta_e\xi_{v,e}=d_v,$$

which implies that $t_v=(\sum_{e\in E_v}\vartheta_e)^{-1}(\Delta_v-\sum_{e\in E_v}\vartheta_e r_{e,v}-d_v)$. Note that $\Delta_v-\sum_{e\in E_v}\vartheta_e r_{e,v}-d_v>0$ because $\Delta_v-\sum_{e\in E_v}\vartheta_e\xi_{v,e}-d_v=0$ and $\xi_{v,e}>r_{e,v}$ for all $e\in E_v$. Therefore we have that

$$s_{e,v}^*=\xi_{v,e}^*=r_{e,v}+d_v^{-1}\Big[\Delta_v-\sum_{e\in E_v}\vartheta_e r_{e,v}-d_v\Big]_+.$$

*Case 2.* If $\lambda=0$, then we have that $p\vartheta_e|\xi_{v,e}-r_{e,v}|^{p-1}\operatorname{sign}(\xi_{v,e}-r_{e,v})\ni 0$ for all $e\in E_v$, which implies $\xi_{v,e}-r_{e,v}=0$ for all $e\in E_v$. Then we must have

$$\Delta_v-\sum_{e\in E_v}\vartheta_e r_{e,v}=\Delta_v-\sum_{e\in E_v}\vartheta_e\xi_{v,e}\le d_v.$$

Therefore we still have that

$$s_{e,v}^*=\xi_{v,e}^*=r_{e,v}=r_{e,v}+d_v^{-1}\Big[\Delta_v-\sum_{e\in E_v}\vartheta_e r_{e,v}-d_v\Big]_+.$$

The required result then follows from the definition of $A_e$ and $D$. $\qquad\square$

We are now ready to show that the primal problem (A.1) can be cast into an equivalent separable formulation, which can then be solved by the AM method in Algorithm B.1. We give the reformulation under general $\ell_p$-norm penalty and arbitrary $\vartheta_e>0$.

**Lemma B.2** (Lemma 3 in the main paper). *The following problem is equivalent to* (A.1) *for any* $\sigma > 0$, *in the sense that* $(\hat{\phi}, \hat{r}, \hat{z})$ *is optimal in* (A.1) *for some* $\hat{z} \in \mathbb{R}^{|V|}$ *if and only if* $(\hat{\phi}, \hat{r}, \hat{s})$ *is optimal in* (B.4) *for some* $\hat{s} \in \bigotimes_{e \in E} \mathbb{R}^{|V|}$.

$$
\min_{\phi, r, s} \frac{1}{p} \sum_{e \in E} \vartheta_e \left( \phi_e^p + \frac{1}{\sigma^{p-1}} \|s_e - r_e\|_p^p \right)
$$

$$
\text{s.t. } (\phi, r) \in \mathcal{C}, \ \Delta - \sum_{e \in E} \vartheta_e s_e \leq d, \ s_{e,v} = 0, \forall v \notin e.
$$
(B.4)

*Proof.* We will show the forward direction and the converse follows from exactly the same reasoning. Let $\hat{\nu}_1$ and $\hat{\nu}_2$ denote the optimal objective value of problems (A.1) and (B.4), respectively. Let $(\hat{\phi}, \hat{r}, \hat{z})$ be an optimal solution for (A.1). Define $\hat{s}_e := \hat{r}_e + \sigma A_e \hat{z}$ for $e \in E$. We show that $(\hat{\phi}, \hat{r}, \hat{s})$ is an optimal solution for (B.4).

Because $\hat{r}_{e,v} = 0$ for all $v \notin e$, by the definition of $A_e$, we know that $\hat{s}_{e,v} = 0$ for all $v \notin e$. Moreover,

$$
\sigma D \hat{z} = \sigma \sum_{e \in E} \vartheta_e A_e \hat{z} = \sum_{e \in E} \vartheta_e (\hat{s}_e - \hat{r}_e),
$$

so

$$
\Delta - \sum_{e \in E} \vartheta_e \hat{s}_e = \Delta - \sum_{e \in E} \vartheta_e \hat{r}_e - \sigma D \hat{z} \leq d.
$$

Therefore, $(\hat{\phi}, \hat{r}, \hat{s})$ is a feasible solution for (B.4). Furthermore,

$$
\sigma \sum_{v \in V} d_v \hat{z}_v^p = \sigma \sum_{e \in E} \vartheta_e \sum_{v \in e} \hat{z}_v^p = \sigma \sum_{e \in E} \vartheta_e \|A_e \hat{z}\|_p^p
$$

$$
= \frac{1}{\sigma^{p-1}} \sum_{e \in E} \vartheta_e \|\sigma A_e \hat{z}\|_p^p = \frac{1}{\sigma^{p-1}} \sum_{e \in E} \vartheta_e \|\hat{s}_e - \hat{r}_e\|_p^p.
$$

This means that $(\hat{\phi}, \hat{r}, \hat{s})$ attains objective value $\hat{\nu}_1$ in (B.4). Hence $\hat{\nu}_1 \geq \hat{\nu}_2$.

In order to show that $(\hat{\phi}, \hat{r}, \hat{s})$ is indeed optimal for (B.4), it left to show that $\hat{\nu}_2 \geq \hat{\nu}_1$. Let $(\phi', r', s')$ be an optimal solution for (B.4). Then we know that

$$
s' = \operatorname*{argmin}_{s \in \bigotimes_{e \in E} \mathbb{R}^{|V|}} \sum_{e \in E} \vartheta_e \|s_e - r'_e\|_p^p, \text{ s.t. } \Delta - \sum_{e \in E} \vartheta_e s_e \leq d, \ s_{e,v} = 0 \ \forall v \notin e.
$$
(B.5)

According to Lemma B.1, we know that

$$
s'_e = r'_e + A_e D^{-1} \left[ \Delta - \sum_{e' \in E} \vartheta_{e'} r'_{e'} - d \right]_+, \ \forall e \in E.
$$
(B.6)

Define $z' := \frac{1}{\sigma} D^{-1} [\Delta - \sum_{e \in E} \vartheta_e r'_e - d]_+$. Then $z' \geq 0$. Moreover, we have that

$$
\sum_{e \in E} \vartheta_e s'_e - \sum_{e \in E} \vartheta_e r'_e = \sum_{e \in E} \vartheta_e A_e D^{-1} \left[ \Delta - \sum_{e' \in E} \vartheta_{e'} r'_{e'} - d \right]_+ = \left[ \Delta - \sum_{e' \in E} \vartheta_{e'} r'_{e'} - d \right]_+ = \sigma D z',
$$

so

$$
\Delta - \sum_{e \in E} \vartheta_e r'_e = \Delta - \sum_{e \in E} \vartheta_e s'_e + \sigma D z' \leq d + \sigma D z'.
$$

Therefore, $(\phi', r', z')$ is a feasible solution for (A.1). Furthermore,

$$
\frac{1}{\sigma^{p-1}} \sum_{e \in E} \vartheta_e \|s'_e - r'_e\|_p^p = \frac{1}{\sigma^{p-1}} \sum_{e \in E} \vartheta_e \|\sigma A_e z'\|_p^p = \sigma \sum_{e \in E} \vartheta_e \|A_e z'\|_p^p
$$

$$
= \sigma \sum_{e \in E} \vartheta_e \sum_{v \in e} z_v'^p = \sigma \sum_{v \in V} d_v z_v'^p.
$$

This means that $(\phi', r', z')$ attains objective value $\hat{\nu}_2$ in (A.1). Hence $\hat{\nu}_2 \geq \hat{\nu}_1$. $\quad\square$

**Remark.** The constructive proof of Lemma B.2 means that, given an optimal solution $(\hat{\phi}, \hat{r}, \hat{s})$ for problem (B.4), one can recover an optimal solution $(\hat{\phi}, \hat{r}, \hat{z})$ for our original primal formulation (A.1) via $\hat{z} := \frac{1}{\sigma} D^{-1}[\Delta - \sum_{e \in E} \vartheta_e \hat{r}_e - d]_+$. It then follows from Lemma A.2 that the dual optimal solution $\hat{x}$ is given by $\hat{x} = \hat{z}^{p-1}$. Therefore, a sweep cut rounding procedure readily applies to the solution $(\hat{\phi}, \hat{r}, \hat{s})$ of problem (B.4).

Let $g(\phi, r, s)$ denote the objective function of problem (B.4) and let $g^*$ denote its optimal objective value.

The following theorem gives the convergence rate of Algorithm B.1 applied to (B.4), when its objective function is penalized by $\ell_p$-norm for $p \geq 2$.

**Theorem B.3** ([4]). *Let $\{\phi^{(k)}, r^{(k)}, s^{(k)}\}_{k \geq 0}$ be the sequence generated by Algorithm B.1. Then for any $k \geq 1$,*

$$g(\phi^{(k)}, r^{(k)}, s^{(k)}) - g^* \leq \frac{3 \max\{g(\phi^{(0)}, r^{(0)}, s^{(0)}) - g^*, L_p R^2\}}{k},$$

*where*

$$R = \max_{(\phi, r, s) \in \mathcal{F}} \max_{(\hat{\phi}, \hat{r}, \hat{s}) \in \mathcal{O}} \left\{ \|\phi - \hat{\phi}\|_2^2 + \|r - \hat{r}\|_2^2 + \|s - \hat{s}\|_2^2 \mid g(\phi, r, s) \leq g(\phi^{(0)}, r^{(0)}, s^{(0)}) \right\},$$

$$L_p = (p-1) \frac{\vartheta_{\max}^{2/p} \|\Delta\|_p^{p-2}}{d_{\min}^{(p-1)(p-2)/p} \sigma^{p-1}},$$

*where $\mathcal{F}$ and $\mathcal{O}$ denote the feasible set and set of optimal solutions, respectively, $\vartheta_{\max} := \max_{e \in E} \vartheta_e$, and $d_{\min} := \min_{v \in \text{supp}(\Delta)} d_v$.*

**Remark.** When $p = 2$, as considered in the main paper, the objective function $g(\phi, r, s)$ has Lipschitz continuous gradient with constant $L_2 = \vartheta_{\max}/\sigma$. When $p > 2$, the gradient of $g(\phi, r, s)$ is not generally Lipschitz continuous. However, the sub-linear convergence rate in Theorem B.3 applies as long as $g(\phi, r, s)$ is block Lipschitz smooth in the sub-level sets containing the iterates generated by Algorithm B.1. We give more details in Subsection B.1.

## B.1 Block Lipschitz smoothness over sub-level set

Recall that $g(\phi, r, s)$ denotes the objective function of problem (B.4). Lemma B.4 concerns specifically the setting when problem B.4 is penalized by the $\ell_p$-norm for some $p > 2$.

**Lemma B.4** (Block Lipschitz smoothness). *The partial gradient $\nabla_{(\phi, r)} g(\phi, r, s)$ is Lipschitz continuous over the sub-level sets (given any fixed $s$)*

$$U_{\phi, r}(s) := \{(\phi, r) \in \mathbb{R}_+^{|V|} \times (\bigotimes_{e \in E} \mathbb{R}^{|V|}) \mid g(\phi, r, s) \leq g(\phi^{(0)}, r^{(0)}, s^{(0)})\}$$

*with constant $L_{\phi, r}$ such that*

$$L_{\phi, r} \leq (p-1) \frac{\vartheta_{\max}^{2/p} \|\Delta\|_p^{p-2}}{d_{\min}^{(p-1)(p-2)/p} \sigma^{p-1}},$$

*where $\vartheta_{\max} := \max_{e \in E} \vartheta_e$ and $d_{\min} := \min_{v \in \text{supp}(\Delta)} d_v$. The partial gradient $\nabla_s g(\phi, r, s)$ is Lipschitz continuous over the sub-level sets (given any fixed $(\phi, r)$)*

$$U_s(\phi, r) := \{s \in \bigotimes_{e \in E} \mathbb{R}^{|V|} \mid g(\phi, r, s) \leq g(\phi^{(0)}, r^{(0)}, s^{(0)})\}$$

*with constant $L_s \leq L_{\phi, r}$.*

*Proof.* Fix $s \in \bigotimes_{e \in E} \mathbb{R}^{|V|}$ and consider

$$g_1(\phi, r) := g(\phi, r, s) = \frac{1}{p} \sum_{e \in E} \vartheta_e \phi_e^p + \frac{1}{p \sigma^{p-1}} \sum_{e \in E} \sum_{v \in V} \vartheta_e |r_{e,v} - s_{e,v}|^p.$$

The function $g_1(\phi, r)$ is coordinate-wise separable and hence its second order derivative $\nabla^2 g_1(\phi, r)$ is a diagonal matrix. Therefore, the largest eigenvalue of $\nabla^2 g_1(\phi, r)$ is the largest coordinate-wise second order partial derivative, that is,

$$L_{\phi,r} = \max_{(\phi,r) \in U_{\phi,r}(s)} \lambda_{\max}(\nabla^2 g_1(\phi, r)) = \max_{(\phi,r) \in U_{\phi,r}(s)} \max_{e \in E, v \in V} \{\nabla^2_{\phi_e} g_1(\phi, r), \nabla^2_{r_{e,v}} g_1(\phi, r)\}.$$

So it suffices to upper bound $\nabla^2_{\phi_e} G(\phi, r)$ and $\nabla^2_{r_{e,v}} G(\phi, r)$ for all $(\phi, r) \in U_{\phi,r}(s)$. We have that

$$g(\phi^{(0)}, r^{(0)}, s^{(0)}) = \frac{1}{p\sigma^{p-1}} \sum_{e \in E} \vartheta_e \sum_{v \in e} \frac{[\Delta_v - d_v]^p_+}{d_v^p} = \frac{1}{p\sigma^{p-1}} \sum_{v \in V} \frac{[\Delta_v - d_v]^p_+}{d_v^{p-1}} \leq \frac{\|\Delta\|_p^p}{p\sigma^{p-1} d_{\min}^{p-1}}$$

where $d_{\min} = \min_{v \in \mathrm{supp}(\Delta)} d_v$. It follows that for all $(\phi, r) \in U_{\phi,r}(s)$,

$$\nabla^2_{\phi_e} g_1(\phi, r) = (p-1) \vartheta_e \phi_e^{p-2} \leq \frac{(p-1)\vartheta_e^{2/p} \|\Delta\|_p^{p-2}}{d_{\min}^{(p-1)(p-2)/p} \sigma^{(p-1)(p-2)/p}} \leq \frac{(p-1)\vartheta_e^{2/p} \|\Delta\|_p^{p-2}}{d_{\min}^{(p-1)(p-2)/p} \sigma^{p-1}}, \ \forall e \in E,$$

$$\nabla^2_{r_{e,v}} g_1(\phi, r) = (p-1) \frac{\vartheta_e}{\sigma^{p-1}} |s_{e,v} - r_{e,v}|^{p-2} \leq \frac{(p-1)\vartheta_e^{2/p} \|\Delta\|_p^{p-2}}{d_{\min}^{(p-1)(p-2)/p} \sigma^{p-1}}, \ \forall e \in E, \ \forall v \in V,$$

because otherwise we would have $g(\phi, r, s) > g(\phi^{(0)}, r^{(0)}, s^{(0)})$. Hence,

$$L_{\phi,r} \leq \max_{e \in E} \frac{(p-1)\vartheta_e^{2/p} \|\Delta\|_p^{p-2}}{d_{\min}^{(p-1)(p-2)/p} \sigma^{p-1}} = \frac{(p-1)\vartheta_{\max}^{2/p} \|\Delta\|_p^{p-2}}{d_{\min}^{(p-1)(p-2)/p} \sigma^{p-1}}.$$

Finally, by the symmetry between $r$ and $s$ in $F(\phi, r, s)$, we know that $L_s \leq L_{\phi,r}$. $\quad\square$

**Remark.** Because the iterates generated by Algorithm B.1 monotonically decrease the objective function value, in particular, we have that

$$g(\phi^{(0)}, r^{(0)}, s^{(0)}) \geq g(\phi^{(k+1)}, r^{(k+1)}, s^{(k)}) \geq g(\phi^{(k+1)}, r^{(k+1)}, s^{(k+1)})$$

for any $k \geq 0$. Therefore, the sequence of iterates live in the sub-level sets. As a result, for any $p > 2$, the block Lipschitz smoothness within sub-level sets suffices to obtain the sub-linear convergence rate for the AM method [4].

## B.2 Alternating minimization sub-problems

We now discuss how to solve the sub-problems in Algorithm B.1 efficiently. By Lemma B.1, we know that the sub-problem with respect to $s$,

$$s^{(k+1)} := \underset{s}{\operatorname{argmin}} \sum_{e \in E} \vartheta_e \|s_e - r_e^{(k+1)}\|_p^p, \ \text{s.t.} \ \Delta - \sum_{e \in E} \vartheta_e s_e \leq d, \ s_{e,v} = 0, \forall v \notin e,$$

has closed-form solution

$$s_e^{(k+1)} = r_e^{(k+1)} + A_e D^{-1} \left[\Delta - \sum_{e' \in E} \vartheta_{e'} r_{e'}^{(k+1)} - d\right]_+, \ \forall e \in E.$$

For the sub-problem with respect to $(\phi, r)$,

$$(\phi^{(k+1)}, r^{(k+1)}) := \underset{(\phi,r) \in \mathcal{C}}{\operatorname{argmin}} \sum_{e \in E} \vartheta_e \left(\phi_e^p + \frac{1}{\sigma^{p-1}} \|s_e^{(k)} - r_e\|_p^p\right),$$

note that it decomposes into $|E|$ independent problems that can be minimized separately. That is, for $e \in E$, we have

$$\begin{aligned}
(\phi_e^{(k+1)}, r_e^{(k+1)}) &= \underset{\phi_e \geq 0, r_e \in \phi_e B_e}{\operatorname{argmin}} \vartheta_e \phi_e^p + \frac{1}{\sigma^{p-1}} \vartheta_e \|s_e^{(k)} - r_e\|_p^p \\
&= \underset{\phi_e \geq 0, r_e \in \phi_e B_e}{\operatorname{argmin}} \frac{1}{p} \phi_e^p + \frac{1}{p\sigma^{p-1}} \|s_e^{(k)} - r_e\|_p^p.
\end{aligned} \tag{B.7}$$

The above problem (B.7) is strictly convex so it has a unique minimizer.

We focus on $p = 2$ first. In this case, problem (B.7) can be solved in sub-linear time using either the conic Frank-Wolfe algorithm or the conic Fujishige-Wolfe minimum norm algorithm studied in [5]. Notice that the dimension of problem (B.7) is the size of the corresponding hyperedge. Therefore, as long as the hyperedge is not extremely large, we can easily obtain a good update $(\phi_e^{(k+1)}, r_e^{(k+1)})$.

If $B_e$ has a special structure, for example, if the hyperedge weight $w_e$ models unit cut-cost, then an exact solution for (B.7) can be computed in time $O(|e| \log |e|)$ [5]. For completeness we transfer the algorithmic details in [5] to our setting and list them in Algorithm B.2. The basic idea is to find optimal dual variables achieving dual optimality, and then recover primal optimal solution from the dual. We refer the reader to [5] for detailed justifications. Given $e \in E$, $s_e \in \mathbb{R}^{|V|}$, and $a, b \in \mathbb{R}$, denote

$$e_{\geq}(a) := \{v \in e \mid s_{e,v} \geq \sigma a\} \text{ and } e_{\leq}(b) := \{v \in e \mid s_{e,v} \leq \sigma b\}.$$

Define

$$\gamma(a, b) := a - b + \sum_{v \in e_{\geq}(a)} \sigma \left(a - \frac{s_{e,v}}{\sigma}\right).$$

---

**Algorithm B.2** An Exact Projection Algorithm for (B.7) ($p = 2$, unit cut-cost) [5]

1: **Input:** $e$, $s_e$.
2: $a \leftarrow \max_{v \in e} s_{e,v}/\sigma$, $\quad b \leftarrow \min_{v \in e} s_{e,v}/\sigma$
3: **While** true:
4: $\quad\quad w_a \leftarrow \sigma |e_{\geq}(a)|$, $\; w_b \leftarrow \sigma |e_{\leq}(b)|$
5: $\quad\quad a_1 \leftarrow \max_{v \in e \setminus e_{\geq}(a)} s_{e,v}/\sigma$, $\; b_1 \leftarrow b + (a - a_1)w_a/w_b$
6: $\quad\quad b_2 \leftarrow \min_{v \in e \setminus e_{\leq}(b)} s_{e,v}/\sigma$, $\; a_2 \leftarrow a - (b_2 - b)w_b/w_a$
7: $\quad\quad i^* \leftarrow \arg\min_{i \in \{1,2\}} b_i$
8: $\quad\quad$ **If** $a_{i^*} \leq b_{i^*}$ **or** $\gamma(a_{i^*}, b_{i^*}) \leq 0$ **break**
9: $\quad\quad a \leftarrow a_{i^*}$, $\; b \leftarrow b_{i^*}$
10: $a \leftarrow a - \gamma(a, b)w_b/(w_a w_b + w_a + w_b)$, $\; b \leftarrow b + \gamma(a, b)w_a/(w_a w_b + w_a + w_b)$
11: **For** $v \in e$ **do**:
12: $\quad\quad$ **If** $v \in e_{\geq}(a)$ **then** $r_{e,v} \leftarrow s_{e,v} - \sigma a$
13: $\quad\quad$ **Else if** $v \in e_{\leq}(b)$ **then** $r_{e,v} \leftarrow s_{e,v} - \sigma b$
14: $\quad\quad$ **Else** $r_{e,v} \leftarrow 0$
15: **Return:** $r_e$

---

Now we discuss the case $p > 2$ in (B.7). The dual of (B.7) is written as

$$\min_{y_e} \frac{1}{q} f_e(y_e)^q + \frac{\sigma}{q} \|y_e\|_q^q - y_e^T s_e^{(k)}. \tag{B.8}$$

Let $(\phi_e^*, r_e^*)$ and $y_e^*$ be optimal solutions of (B.7) and (B.8), respectively. Then one has

$$r_e^* = s_e^{(k)} - \sigma (y_e^*)^{q-1} \text{ and } \phi_e^* = \left((r_e^*)^T y_e^*\right)^{1/q}.$$

Both the derivation of (B.8) and the above relations between $(\phi_e^*, r_e^*)$ and $y_e^*$ follow from similar reasoning and algebraic computations used in the proofs of Lemma A.1 and Lemma A.2. Therefore, we can use subgradient method to compute $y_e^*$ first and then recover $\phi_e^*$ and $r_e^*$. For special cases like the unit cut-cost, a similar approach to Algorithm B.2 can be adopted to obtain an almost (up to a binary search tolerance) exact solution, by modifying Steps 2-6 to work with general $\ell_p$-norm and replacing Step 10 with binary search. See Algorithm B.3 for details.

**Caution.** To simplify notation in Algorithm B.3, for $c \in \mathbb{R}$ and $p > 0$, $c^p$ is to be interpreted as $c^p := |c|^p \operatorname{sign}(c)$, where we treat $\operatorname{sign}(0) := 0$. For $q = p/(p-1)$, we define

$$\gamma_p(a, b) := (a - b)^{q-1} + \sum_{v \in e_{\geq}(a^{q-1})} \sigma \left(a^{q-1} - \frac{s_{e,v}}{\sigma}\right).$$

**Algorithm B.3** An $\ell_p$-Projection Algorithm for (B.7) ($p > 2$, unit cut-cost)

1: **Input:** $e$, $s_e$.
2: $a \leftarrow \max_{v \in e}(s_{e,v}/\sigma)^{p-1}$, $\quad b \leftarrow \min_{v \in e}(s_{e,v}/\sigma)^{p-1}$, $\quad q \leftarrow p/(p-1)$
3: **While** true:
4: $\quad w_a \leftarrow \sigma \left|e_{\geq}(a^{q-1})\right|$, $\quad w_b \leftarrow \sigma \left|e_{\leq}(b^{q-1})\right|$
5: $\quad a_1 \leftarrow \max_{v \in e \setminus e_{\geq}(a^{q-1})}(s_{e,v}/\sigma)^{p-1}$, $\quad b_1 \leftarrow (b^{q-1} + (a^{q-1} - a_1^{q-1})w_a/w_b)^{p-1}$
6: $\quad b_2 \leftarrow \min_{v \in e \setminus e_{\leq}(b^{q-1})}(s_{e,v}/\sigma)^{p-1}$, $\quad a_2 \leftarrow (a^{q-1} - (b_2^{q-1} - b^{q-1})w_b/w_a)^{p-1}$
7: $\quad i^* \leftarrow \operatorname{argmin}_{i \in \{1,2\}} b_i$
8: $\quad$ **If** $a_{i^*} \leq b_{i^*}$ **or** $\gamma_p(a_{i^*}, b_{i^*}) \leq 0$ **break**
9: $\quad a \leftarrow a_{i^*}$, $\quad b \leftarrow b_{i^*}$
10: Employ binary search for $\hat{a} \in [b, a]$ such that $\gamma_p(\hat{a}, \hat{b}) = 0$ while maintaining $\hat{b} = (b^{q-1} + (a^{q-1} - \hat{a}^{q-1})w_a/w_b)^{p-1}$ and $\hat{b} \leq \hat{a}$
11: **For** $v \in e$ **do**:
12: $\quad$ **If** $v \in v \in e_{\geq}(\hat{a}^{q-1})$ **then** $r_{e,v} \leftarrow s_{e,v} - \sigma\hat{a}^{q-1}$
13: $\quad$ **Else if** $v \in e_{\leq}(\hat{b}^{q-1})$ **then** $r_{e,v} \leftarrow s_{e,v} - \sigma\hat{b}^{q-1}$
14: $\quad$ **Else** $r_{e,v} \leftarrow 0$
15: **Return:** $r_e$

## C  Empirical set-up and results

### C.1  Experiments using synthetic data

In this subsection we provide details aboue how we generate synthetic hypergraphs using $k$-uniform stochastic block model and how we set the parameters for the algorithms used in our experiments. Additional synthetic experiments that demonstrate or explain the robustness of our method are also provided.

**Data generation.** We generate four sets of hypergraphs using the generalized $k$HSBM described in the main paper. All hypergraphs have $n = 100$ nodes. For simplicity, we require that each block in the hypergraph has constant size 50.

*1st set of hypergraphs.* We generate the first set of hypergraphs with $k = 3$, constant $p = 0.0765$ and varying $q \in [0.0041, 0.0735]$. Recall that for $k = 3$ there is only one possible inter-cluster probability $q \equiv q_1$. We pick $p = 0.0765$ so the expected number of intra-cluster hyperedges is 1500 for each block of size 50. We set a wide range for $q$ so that the interval covers both extremes, i.e., when the ground-truth target cluster is very clean or very noisy. These hypergraphs are used to evaluate the performance of algorithms for the unit cut-cost setting when the target cluster conductance varies. Figure 4 in the main paper uses the local clustering results on these hypergraphs.

*2nd set of hypergraphs.* For the second set of hypergraphs, we vary $k \in \{3, 4, 5, 6\}$. Moreover, we set $q_2 = \cdots = q_{\lfloor k/2 \rfloor} = 0$, so every inter-cluster hyperedge contains a single node on one side and the rest on the other side. In this setting, separating the two ground-truth communities will incur a small penalty using the cardinality cut-cost, but a large penalty using the unit cut-cost. Therefore, methods that exploit appropriate cardinality-based cut-cost should perform better. The hypergraphs are sampled so that the conductance of a block stays the same across different $k$'s. We compute the conductance based on the unit cut-cost when generating the hypergraphs, because the scale of conductance based on the unit cut-cost is less affected by $k$ than the scale of conductance based on the cardinality cut-cost. See details below for how the scale of conductance based on the cardinality cut-cost is affected by $k$. The second set of hypergraphs is used to evaluate the performance of algorithms for both unit and cardinality cut-costs when the hyperedge size varies. Figure 5 in the main paper (and Figure C.3 and Figure C.4 in the appendix) uses the local clustering results on these hypergraphs.

*3rd set of hypergraphs.* For the third set of hypergraphs, we set $q_2 = \cdots = q_{\lfloor k/2 \rfloor} = 0$. We consider constant $k = 4$ or $k = 5$, constant $p$ and varying $q_1$. These hypergraphs are used to evaluate the performance of algorithms for both unit and cardinality cut-costs when the target cluster conductance

varies. Figure C.1 and Figure C.2 in the appendix are based on the local clustering results on these hypergraphs.

*4th set of hypergraphs.* This set consists of two hypergraphs generated with $k = 3$, $p = 0.04$ and $q \in \{0.001, 0.011\}$. The ground-truth target cluster in the first hypergraph has conductance 0.05, while the ground-truth target cluster in the second hypergraph has conductance 0.3. These two hypergraphs are used to compare the performance of algorithms for the unit cut-cost setting, when the theoretical assumptions of LH holds (for the first hypergraph) or fails (for the second hypergraph).

**Parameters.** For HFD, for all synthetic experiments, we initialize the seed mass so that $\|\Delta\|_1$ is three times the volume of the target cluster (recall from Assumption 2 this is without loss of generality). We set $\sigma = 0.01$. We tune the parameters for LH as suggested by the authors [6]. Specifically, LH has a regularization parameter $\kappa$ and we let $\kappa = c \cdot r$ where $r$ is the ratio between the number of seed node(s) and the size of the target cluster. We perform a binary search on $c$ and find that $c = 0.35$ gives good results for the synthetic hypergraphs. An important parameter for LH is $\delta$. When $\delta = 1$ it models unit cut-cost and when $\delta \geq 1$ it models cardinality-based cut-cost with an upper bound $\delta$ [6]. We consider both cases $\delta = 1$ (U-LH) and $\delta \geq 1$ (C-LH). In principle, for $k$-uniform hypergraphs LH should produce the same result for any $\delta \geq k$, so one could simply set $\delta = k$ for C-LH. However in our experiments we find that the $\delta$ value that gives the best clustering results can be much larger than $k$. In order to get the best performance out of C-LH, we run C-LH for $\delta = 2^i$, $i = 0, 1, \ldots, 12$. Among the 13 output clusters from C-LH we pick the one with the lowest conductance. For ACL, we use the same set of parameter values used in [6] because that parameter setting also produces good results in our synthetic experiments.

**Scale of cardinality-based conductance.** To see how ground-truth conductance scales (computed using the cardinality cut-cost) with hyperedge size $k \geq 2$, let us assume that a hypergraph $H = (V, E)$, having $|V| = 100$ nodes and two blocks where each block contains 50 nodes, is generated from $p = 0$, $q_1 = 1$ and $q_2 = \ldots = q_{\lfloor k/2 \rfloor} = 0$. In this case, the hypergraph consists of all and only inter-cluster hyperedges. Let $C$ denote a target cluster, that is, $C$ is either one of the two ground-truth blocks. Since we have $|V| = 100$ nodes and each of the two blocks contains 50 nodes, the total number of hyperedges is

$$|E| = 2 \binom{50}{k-1} \binom{50}{1}.$$

Let $w_e$ denote the cardinality-based cut-cost given by $w_e(S) = \min\{|S \cap e|, |e \setminus S|\}/\lfloor |e|/2 \rfloor$. Then for each $e \in E$ we have that $w_e(C) = \frac{1}{\lfloor k/2 \rfloor}$. Moreover, the volume of $C$ is

$$\text{vol}(C) = (k-1) \binom{50}{k-1} \binom{50}{1} + \binom{50}{1} \binom{50}{k-1} = k \binom{50}{k-1} \binom{50}{1},$$

and hence we have

$$\Phi(C) = \frac{\text{vol}(\partial C)}{\text{vol}(C)} = \frac{\sum_{e \in E} w_e(C)}{\text{vol}(C)} = \frac{\frac{1}{\lfloor k/2 \rfloor} |E|}{\text{vol}(C)} = \frac{\frac{2}{\lfloor k/2 \rfloor} \binom{50}{k-1} \binom{50}{1}}{k \binom{50}{k-1} \binom{50}{1}} = \frac{2}{k \lfloor k/2 \rfloor}.$$

This means that, for any $p \geq 0$, $q_1 \leq 1$, $q_2 = \cdots = q_{\lfloor k/2 \rfloor} = 0$, let $B$ be one of the two blocks in $H$, then $\Phi(B) \leq 1$ for $k = 2, 3$, $\Phi(B) \leq 1/4$ for $k = 4$, and $\Phi(B) \leq 1/5$ for $k = 5$. This explains why the ranges of ground-truth conductance we consider in Figure C.1 and Figure C.2 are different from the range of ground-truth conductance in Figure 4 in the main paper. For each $k$ we try to make the range of conductance (i.e., $x$-axis) as wide as possible, but due to the different scales of cardinality-based conductance for different $k$, the ranges vary accordingly.

**Additional results.** Figure C.1 and Figure C.2 show how the algorithms perform on $k$-uniform hypergraphs for $k = 4, 5$, respectively, as we vary the target cluster conductances. The plots show that as the target cluster becomes more noisy, the performance of all methods degrades. However, C-HFD is better in terms of both conductance and F1 score, especially when the target cluster is noisy but not complete noise (i.e., the ground-truth conductance is high but not too high). For $k = 5$ and high-conductance regime, methods that use unit cut-cost, e.g., U-HFD, have poor performance because they find low-conductance clusters based on the unit cut-cost as opposed to the cardinality cut-cost. In general, lower unit cut-cost conductance does not necessarily translates to lower cardinality-based conductance or higher F1 score. For both Figure C.1 and Figure C.2, the ground-truth conductance is computed using cardinality-based cut-cost, therefore the ground-truth conductances (on the $x$-axes)

have different scales and ranges. Figure C.3 and Figure C.4 show the median (markers) and 25-75 percentiles (lower-upper bars) of conductance ratios and F1 scores for $k = 3, 4, 5, 6$. The target clusters have unit cut-cost conductances around 0.2 for Figure C.3 and 0.25 for Figure C.4. Notice that, when the target clusters are less noisy (cf. Figure 5 in the main paper where target clusters are more noisy, having unit conductance around 0.3), U-HFD and C-HFD are significantly better than other methods. The performance of U-HFD is slightly affected by the hyperedge size when the target clusters have unit conductance around 0.25, while the performance of C-HFD stays the same across all $k$'s.

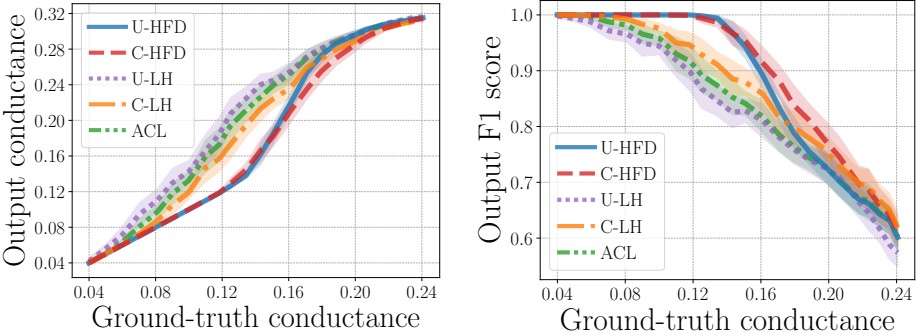

Figure C.1: Average output conductance and F1 score against ground-truth conductance, on $k$-uniform hypergraphs with $k = 4$. The error bars show variation over 50 runs using different seed nodes. Both the ground-truth and the target conductances are computed using cardinality-based cut-cost.

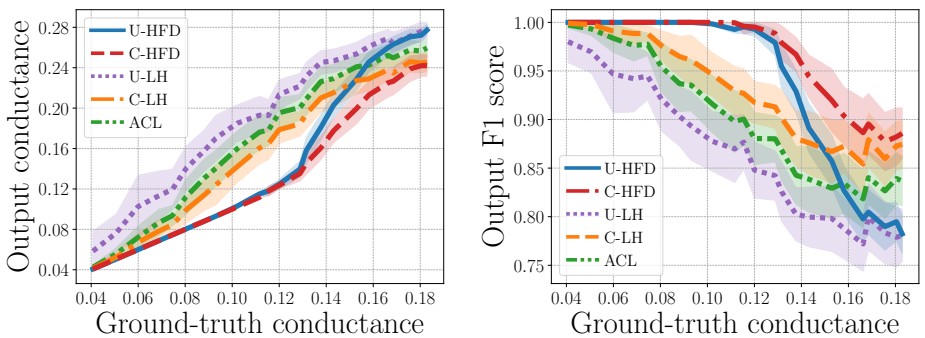

Figure C.2: Average output conductance and F1 score against ground-truth conductance, on $k$-uniform hypergraphs with $k = 5$. The error bars show variation over 50 runs using different seed nodes. Both the ground-truth and the target conductances are computed using cardinality-based cut-cost.

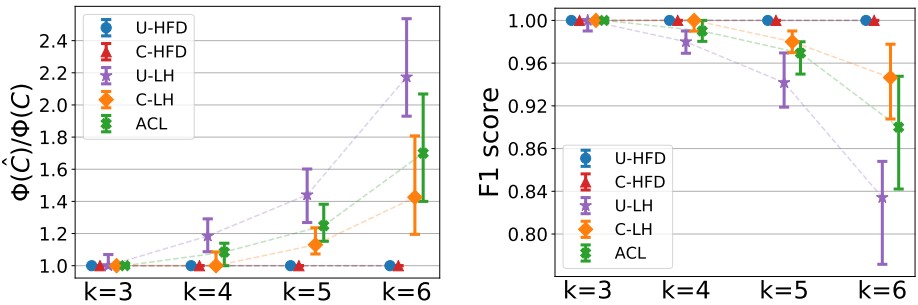

Figure C.3: Conductance ratio and F1 score on $k$-uniform hypergraphs for $k \in \{3, 4, 5, 6\}$. Target clusters have unit conductance around 0.20.

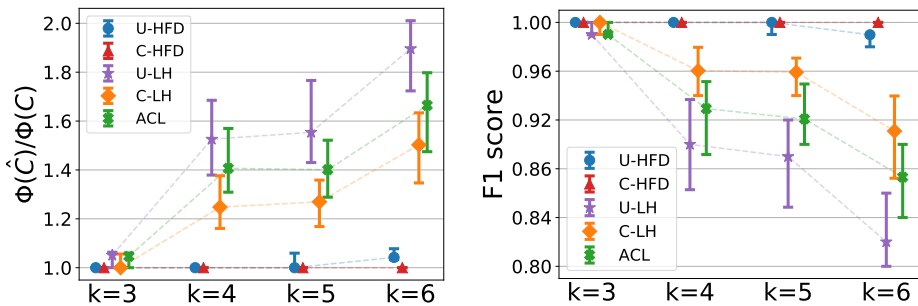

Figure C.4: Conductance ratio and F1 score on $k$-uniform hypergraphs for $k \in \{3, 4, 5, 6\}$. Target clusters have unit conductance around 0.25.

**Why is the empirical performance of U-HFD better than U-LH?** For the unit cut-cost setting, the local clustering guarantee for HFD holds under much weaker assumptions than those required for LH. The assumptions for LH could fail in many cases, and consequently we see that U-HFD has significantly better performance than U-LH in the experiments with both synthetic and real data. More specifically, the theoretical framework for LH assumes that the node embeddings are global (i.e., the solution is dense). However, in order to obtain a localized algorithm, the authors use a regularization parameter $\kappa > 0$ to impose sparsity in the solution. The localized algorithm computes a sparse approximation to the original global solution, but some clustering errors could also be introduced. In general, this does not seem to be a major issue, as localized solutions only seem to slightly affect the clustering performance as shown in Figure C.5. A more crucial assumption of LH is that its approximation guarantee relies on a strong condition that the conductance of the target cluster is upper bounded by $\frac{\gamma}{8c}$, where $\gamma \in (0, 1)$ is a tuning parameter and $c$ is a constant that depends on both $\gamma$ and a specific sampling strategy for selecting a seed node from the target cluster. In our experiments we find that this assumption often breaks. In what follows we provide a simple illustrating example using synthetic hypergraphs. First of all, we sample a sequence of hypergraphs using $k$HSBM with $n = 100$ nodes, two ground-truth communities each consisting of 50 nodes, constant $k = 3$, varying $p$ and $q$. For each hypergraph we identify one ground-truth community as the target cluster, and we select a seed node uniformly at random from the target cluster. We compute the quantity $\frac{\gamma}{8c}$ and we find that this quantity is always less than 0.12 for any $\gamma \in (0, 1)$. This means that in order for the assumption of LH to hold, the target cluster must have conductance no more than 0.12, which is a very strict requirement and cannot hold in general. In order to compare the performances of LH when its assumption holds or fails, respectively, we picked two hypergraphs (i.e., the fourth set of hypergraphs that we generate) that correspond to the two scenarios. The target clusters have conductance 0.05 and 0.3, respectively. Therefore, the assumption for LH holds for the first hypergraph but fails for the second hypergraph. Moreover, we consider both global and localized solutions for LH. The global solution demonstrates the performance of LH under the required theoretical framework, while the localized solution demonstrates what happens in practice when one uses sparse approximation for computational efficiency. For LH, we compute the global solution by simply setting the regularization parameter $\kappa$ to 0; we tune the localized solution and set $\kappa = 0.25r$ where $r$ is the ratio between the number of seed node(s) and the size of the target cluster. The way we pick $\kappa$ is similar to the authors' choice for LH. For HFD, we set $\sigma = 0.01$ and initial mass 3 times the volume of the target cluster. We run both methods multiple times, each time we use a different node from the target cluster as the single seed node. The median, lower and upper quantiles of F1 scores are shown in Figure C.5. For LH, observe that (i) for both hypergraphs where the assumption either holds or fails, localizing the solution slightly reduces the F1 score, and (ii) for both global and localized solutions, LH has much worse performance on the hypergraph where its assumption does not hold. On the other hand, HFD perfectly recovers the target clusters in both settings.

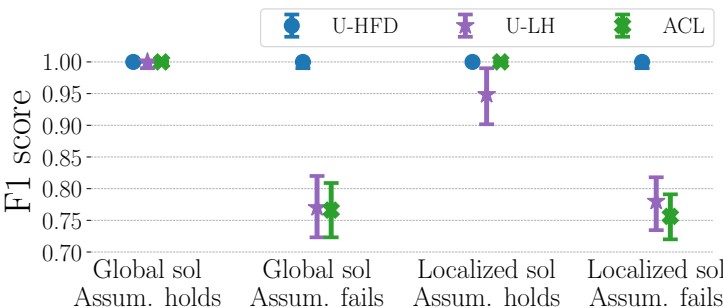

Figure C.5: Local clustering results under various settings for LH. The markers show the median, the error bars show the 25th and 75th percentiles, respectively. The left-most case aligns with the required theoretical framework for LH, moreover, the srong assumption on the target cluster conductance is satisfied; the right-most case is what typically happens in practice when one applies localized algorithm for LH, moreover, the assumption on the target cluster conductance does not hold. ACL is a heuristic method that applies to the star expansion of hypergraphs. ACL has no performance guarantee. In practice, we observe that ACL and LH have similar performances.

## C.2  Experiments using real-world data

### C.2.1  Datasets and ground-truth clusters

We provide complete details on the real hypergraphs we used in the experiments. The last three datasets are used for additional experiments in the appendix only.

*Amazon-reviews* [7, 8]. This is a hypergraph constructed from Amazon product review data, where each node represents a product. A set of products are connected by a hyperedge if they are reviewed by the same person. We use product category labels as ground truth cluster identities. In total there are 29 product categories. Because we are mostly interested in local clustering, we consider all clusters consisting of less than 10,000 nodes.

*Trivago-clicks* [9]. The nodes in this hypergraph are accommodations/hotels. A set of nodes are connected by a hyperedge if a user performed "click-out" action during the same browsing session, which means the user was forwarded to a partner site. We use geographical locations as ground truth cluster identities. There are 160 such clusters. We consider all clusters in this dataset that consists of less than 1,000 nodes and has conductance less than 0.25.

*Florida Bay food network* [10]. Nodes in this hypergraph correspond to different species or organisms that live in the Bay, and hyperedges correspond to transformed network motifs of the original dataset. Each species is labelled according its role in the food chain.

*High-school-contact* [11, 9]. Nodes in this hypergraph represent high school students. A group of people are connected by a hyperedge if they were all in proximity of one another at a given time, based on data from sensors worn by students. We use the classroom to which a student belongs to as ground truth. In total there are 9 classrooms.

*Microsoft-academic* [12, 13]. The original co-authorship network is a subset of the Microsoft Academic Graph where nodes are authors and hyperedges correspond to a publication from those authors. We take the dual of the original hypergraph by converting hyperedges to nodes and nodes to hyperedges. After constructing the dual hypergraph, we removed all hyperedges having just one node and we kept the largest connected component. In the resulting hypergraph, each node represents a paper and is labelled by its publication venue. A set of papers are connected by a hyperedge if they share a common coauthor. We combine similar computer science conferences into four broader categories: Data (KDD, WWW, VLDB, SIGMOD), ML (ICML, NeurIPS), TCS (STOC, FOCS), CV (ICCV, CVPR).

*Oil-trade network*. This hypergraph is constructed using the 2017 international oil trade records from UN Comtrade Dataset. We adopt a similar modelling approach to Figure 1 in the main paper. Each node represents a country, $\{v_1, v_2, v_3, v_4\}$ form a hyperedge if the trade surplus from each of $v_1, v_2$ to each of $v_3, v_4$ exceeds 10 million USD (this is roughly 80% percentile country-wise oil export

value). Therefore, two countries belong to the same hyperedge if they share $\geq 2$ important trading partners in common. We use this network to for the node ranking problem.

Table C.1 provides summary statistics about the hypergraphs. Table C.2 includes the statistics of all ground truth clusters that we used in the experiments.

Table C.1: Summary of real-world hypergraphs

| Dataset | Number of nodes | Number of hyperedges | Maximum hyperedge size | Maximum node degree | Median / Mean hyperedge size | Median / Mean node degree |
|---|---|---|---|---|---|---|
| Amazon-reviews | 2,268,231 | 4,285,363 | 9,350 | 28,973 | 8.0 / 17.1 | 11.0 / 32.2 |
| Trivago-clicks | 172,738 | 233,202 | 86 | 588 | 3.0 / 4.1 | 2.0 / 5.6 |
| Florida-Bay | 126 | 141,233 | 4 | 19,843 | 4.0 / 4.0 | 3,770.5 / 4,483.6 |
| Microsoft-academic | 44,216 | 22,464 | 187 | 21 | 3.0 / 5.4 | 2.0 / 2.7 |
| High-school-contact | 327 | 7,818 | 5 | 148 | 2.0 / 2.3 | 53.0 / 55.6 |
| Oil-trade | 229 | 100,639 | 4 | 16,394 | 4.0 / 4.0 | 175.0 / 1,757.9 |

Table C.2: Summary of ground-truth clusters used in the experiments

| Dataset | Cluster | Size | Volume | Conductance |
|---|---|---|---|---|
| Amazon-reviews | 1 - Amazon Fashion | 31 | 3042 | 0.06 |
| | 2 - All Beauty | 85 | 4092 | 0.12 |
| | 3 - Appliances | 48 | 183 | 0.18 |
| | 12 - Gift Cards | 148 | 2965 | 0.13 |
| | 15 - Industrial & Scientific | 5334 | 72025 | 0.14 |
| | 17 - Luxury Beauty | 1581 | 28074 | 0.11 |
| | 18 - Magazine Subs. | 157 | 2302 | 0.13 |
| | 24 - Prime Pantry | 4970 | 131114 | 0.10 |
| | 25 - Software | 802 | 11884 | 0.14 |
| Trivago-clicks | KOR - South Korea | 945 | 3696 | 0.24 |
| | ISL - Iceland | 202 | 839 | 0.21 |
| | PRI - Puerto Rico | 144 | 473 | 0.25 |
| | UA-43 - Crimea | 200 | 1091 | 0.24 |
| | VNM - Vietnam | 832 | 2322 | 0.24 |
| | HKG - Hong Kong | 536 | 4606 | 0.24 |
| | MLT - Malta | 157 | 495 | 0.24 |
| | GTM - Guatemala | 199 | 652 | 0.24 |
| | UKR - Ukraine | 264 | 648 | 0.24 |
| | SET - Estonia | 158 | 850 | 0.23 |
| Florida-Bay | Producers | 17 | 10781 | 0.70 |
| | Low-level consumers | 35 | 173311 | 0.58 |
| | High-level consumers | 70 | 375807 | 0.54 |
| Microsoft-academic | Data | 15817 | 45060 | 0.06 |
| | ML | 10265 | 26765 | 0.16 |
| | TCS | 4159 | 10065 | 0.08 |
| | CV | 13974 | 38395 | 0.08 |
| High-school-contact | Class 1 | 36 | 1773 | 0.25 |
| | Class 2 | 34 | 1947 | 0.29 |
| | Class 3 | 40 | 2987 | 0.20 |
| | Class 4 | 29 | 913 | 0.41 |
| | Class 5 | 38 | 2271 | 0.26 |
| | Class 6 | 34 | 1320 | 0.26 |
| | Class 7 | 44 | 2951 | 0.16 |
| | Class 8 | 39 | 2204 | 0.19 |
| | Class 9 | 33 | 1826 | 0.25 |

### C.2.2 Methods and parameter setting

**HFD** We use $\sigma = 0.0001$ for all the experiments. We set the total amount of initial mass $\|\Delta\|_1$ as a constant factor $t$ times the volume of the target cluster. For Amazon-reviews, on the smaller clusters

1, 2, 3, 12, 18, we used $t = 200$; on the larger clusters 15, 17, 24, 25, we used $t = 50$. For both Trivago-clicks, High-school-contact and Microsoft-academic, we used $t = 3$. For Florida Bay food network, we used $t = 20, 10, 5$ for clusters 1, 2, 3, respectively. In all experiments, the choice of $t$ is to ensure that the diffusion process will cover some part of the target and incur a high cost in the objective function. For the single seed node setting, we simply set the initial mass on the seed node as $\|\Delta\|_1$. For the multiple seed nodes setting where we are given a seed set $S$, for each $v \in S$ we set the initial mass on $v$ as $d_v\|\Delta\|_1/\mathrm{vol}(S)$.

**LH, ACL** We used the parameters as suggested by the authors [6]. For both *-LH-2.0 and *-LH-1.4, we set $\gamma = 0.1$, $\rho = 0.5$, $\kappa = c \cdot r$ where $r$ is the ratio between the number of seed nodes and the size of the target cluster, and $c$ is a tuning constant. For Amazon-reviews, we set $c = 0.025$ as suggested in [6]. For Microsoft-academic, Trivago-clicks, and Florida-Bay we also used $c = 0.025$ because it produces good results. For High-school-contact we selected $c = 0.25$ after some tuning to make sure both *-LH-2.0 and *-LH-1.4 have good results. We set the parameters for ACL in exactly the same way as in [6]. We set $\delta = 1$ for U-LH-* and $\delta = \max_{e \in E} |e|$ for C-LH-*.

### C.2.3 Additional experiments

**Multiple seed nodes.** We conduct additional experiments using multiple seed nodes for Amazon-reviews and Trivago-clicks datasets. For each target cluster, we randomly select 1% nodes from that cluster as seed nodes, and we enforce that at least 5 nodes are selected as seeds. For example, if a cluster consists of only 100 nodes, we still select 5 nodes to form a seed set. We run 30 trials for each cluster and report the median conductance and F1 score of the output clusters. The results are shown in Table C.3 and Table C.4. For the multiple seed nodes setting, the results of U-LH-1.4, U-LH-2.0 and ACL on Amazon-reviews align with the ones reported in [6]: We reproduced almost identical numbers under the same setting, with only a few small differences due to randomness in seed nodes selection. In general, using more seed nodes improves the performance for all methods in terms of both conductance and F1. For Amazon-reviews, the output clusters of HFD always have the lowest conductance, even though in some cases, low conductance does not align well with the given ground-truth, and hence the lowest conductance does not lead to the highest F1 score. Similarly, for Trivago-clicks, both U-HFD and C-HFD consistently find the lowest conductance clusters among all methods, which in general (but not always) lead to a higher F1 score. Note that, if a method uses the unit cut-cost (resp. the cardinality-based cut-cost), then we compute the conductance of the output cluster using the unit cut-cost (resp. the cardinality-based cut-cost). Therefore, depending on the specific cut-cost, the conductances in Table C.4 may have different scales. We highlight the lowest conductance for both cut-costs separately.

**Additional datasets, local clustering using unit and cardinality cut-costs.** Table C.5 and Table C.6 show local clustering results on High-school-contact and Microsoft-academic networks, respectively. We use the single seed node setting, run the methods from each node in a target cluster, and report the median conductance and F1 score. We cap the maximum number of runs to 500. Similar to the results on other datasets, the output clusters of HFD always have the lowest conductance, leading to the highest F1 score in most cases. We omit cardinality-based methods for Microsoft-academic because they are very similar to the unit cut-cost setting.

**Additional dataset, node ranking using general submodular cut-cost.** We provide another compelling use case of general submodular cut-cost. We consider the node ranking problem in the Oil-trade network. Our goal is to search the most related country of a queried country based on the trade-network structure. We use the hypergraph modelling shown in Figure 1 in the main paper. We compare HFD using unit (U-HFD, $\gamma_1 = \gamma_2 = 1$), cardinality-based (C-HFD, $\gamma_1 = 1/2$ and $\gamma_2 = 1$) and submodular (S-HFD, $\gamma_1 = 1/2$ and $\gamma_2 = 0$) cut-costs. Table C.7 shows the top-2 ranking results. In this example, we use Iran as the seed node and we rank other countries according to the ordering of dual variables returned by HFD. In 2017, US imposed strict sanctions on Iran. However, Bangladesh (generally accepted as an American ally) is among the top two ranked countries based on unit or cardinality-based cut-cost, which does not make any sense. On the other hand, S-HFD ranks Iraq and Turkmenistan as the top two. Interested readers can easily verify that these counties share strong economic or historical ties with Iran.

**Additional method: $p$-norm HFD.** We tried HFD with unit cut-cost and $p = 4$ (U-HFD-4.0). However, in practice we did not observe that a larger $p > 2$ necessarily lead to better clustering

Table C.3: Complete local clustering results for Amazon-reviews network

| Metric | Seed | Method | Cluster | | | | | | | | |
|--------|------|--------|---|---|---|----|----|----|----|----|----|
| | | | 1 | 2 | 3 | 12 | 15 | 17 | 18 | 24 | 25 |
| Conductance | Single | U-HFD | **0.17** | **0.11** | **0.12** | **0.16** | **0.36** | **0.25** | **0.17** | **0.14** | **0.28** |
| | | U-LH-2.0 | 0.42 | 0.50 | 0.25 | 0.44 | 0.74 | 0.44 | 0.57 | 0.58 | 0.61 |
| | | U-LH-1.4 | 0.33 | 0.44 | 0.25 | 0.36 | 0.81 | 0.40 | 0.51 | 0.54 | 0.59 |
| | | ACL | 0.42 | 0.50 | 0.25 | 0.54 | 0.77 | 0.52 | 0.63 | 0.68 | 0.65 |
| | Multiple | U-HFD | **0.05** | **0.10** | **0.12** | **0.13** | **0.20** | **0.16** | **0.14** | **0.11** | **0.32** |
| | | U-LH-2.0 | **0.05** | 0.15 | 0.15 | 0.21 | 0.45 | 0.45 | 0.26 | 0.18 | 0.53 |
| | | U-LH-1.4 | **0.05** | 0.13 | 0.15 | 0.15 | 0.35 | 0.33 | 0.19 | 0.14 | 0.47 |
| | | ACL | **0.05** | 0.27 | 0.16 | 0.27 | 0.56 | 0.53 | 0.33 | 0.30 | 0.59 |
| F1 score | Single | U-HFD | **0.45** | **0.09** | **0.65** | **0.92** | 0.04 | **0.10** | **0.80** | **0.81** | **0.09** |
| | | U-LH-2.0 | 0.23 | 0.07 | 0.23 | 0.29 | **0.05** | 0.06 | 0.21 | 0.28 | 0.05 |
| | | U-LH-1.4 | 0.23 | **0.09** | 0.35 | 0.40 | 0.00 | 0.07 | 0.31 | 0.35 | 0.06 |
| | | ACL | 0.23 | 0.07 | 0.22 | 0.25 | 0.04 | 0.05 | 0.17 | 0.20 | 0.04 |
| | Multiple | U-HFD | 0.49 | **0.50** | 0.69 | **0.98** | 0.19 | **0.36** | **0.91** | **0.89** | **0.33** |
| | | U-LH-2.0 | **0.59** | 0.42 | **0.73** | 0.77 | 0.22 | 0.25 | 0.65 | 0.62 | 0.17 |
| | | U-LH-1.4 | 0.52 | 0.45 | **0.73** | 0.90 | **0.27** | 0.29 | 0.79 | 0.77 | 0.20 |
| | | ACL | **0.59** | 0.25 | 0.70 | 0.64 | 0.20 | 0.19 | 0.51 | 0.49 | 0.14 |

results. We show a sample result of U-HFD-4.0 for Amazon-reviews in Table C.8. Notice that the performances of U-HFD-2.0 ($p = 2$) and U-HFD-4.0 are very similar.

**Additional method: LH + flow improve.** We tried a flow-improve method for hypergraphs [8]. We apply the flow-improve method to the output of U-LH-2.0. The method is slow in our experiments, so we only tried it on a few small instances. The results for the Florida Bay food network is shown in Table C.9. In general, we find that applying the flow-improve method does not lead to consistent performance improvements.

### C.3 Computing platform and implementation detail

We implemented the AM algorithm [4] given in Algorithm B.1 in Julia. The code is run on a personal laptop with 32GB RAM and 2.9 GHz 6-Core Intel Core i9. GPU is not used for computation. For the rest of this section, we discuss the implementation details on how we actually solve the nontrivial sub-problem in Algorithm B.1 to obtain the update $(\phi^{(k+1)}, r^{(k+1)})$.

For the unit cut-cost case, we use an exact projection algorithm [5] to obtain the update $(\phi^{(k+1)}, r^{(k+1)})$. Algorithmic details for exact projection is provided in Algorithm B.2. For cardinality-based or general submodular cut-costs, a conic Fujishige-Wolfe minimum norm algorithm [5] can be adopted to efficiently compute $(\phi^{(k+1)}, r^{(k+1)})$. Our implementation uses alternative methods that are simpler. For the cardinality cut-cost, we use a projected subgradient method that works on a related dual problem to obtain the primal update in $(\phi^{(k+1)}, r^{(k+1)})$. The subgradient method is easy to implement, requires less computation overhead, and works well in practice for the sub-problem. For the specialized submodular cut-cost shown in Figure 1, since the hyperedge consists of only 4 nodes and has a special structure, we simply perform an exhaustive search that allows us to exactly compute $(\phi^{(k+1)}, r^{(k+1)})$ using constant number of vector-vector additions and multiplications. We provide details below.

Recall that the sub-problem to compute $(\phi^{(k+1)}, r^{(k+1)})$ decomposes into a sequence of separate problems indexed by $e \in E$ (cf. (B.7)), in the following we assume $p = 2$ for simplicity):

$$\min_{\phi_e \geq 0, r_e \in \phi_e B_e} \frac{1}{2}\phi_e^2 + \frac{1}{2\sigma}\|s_e - r_e\|_2^2. \tag{C.1}$$

The dual problem of (C.1) is written as (cf. (B.8), here we have $p = q = 2$)

$$\min_{y_e} \frac{1}{2}f_e(y_e)^2 + \frac{\sigma}{2}\|y_e\|_2^2 - s_e^T y_e. \tag{C.2}$$

Table C.4: Complete local clustering results for Trivago-clicks network

| Metric | Seed | Method | KOR | ISL | PRI | UA-43 | VNM | HKG | MLT | GTM | UKR | EST |
|---|---|---|---|---|---|---|---|---|---|---|---|---|
| Conductance | Single | U-HFD | **0.010** | **0.023** | **0.014** | **0.011** | **0.018** | **0.017** | **0.010** | **0.007** | **0.016** | **0.012** |
| | | U-LH-2.0 | 0.020 | 0.042 | 0.027 | 0.027 | 0.037 | 0.035 | 0.031 | 0.035 | 0.032 | 0.019 |
| | | U-LH-1.4 | 0.036 | 0.069 | 0.047 | 0.039 | 0.060 | 0.052 | 0.040 | 0.045 | 0.065 | 0.036 |
| | | ACL | 0.027 | 0.050 | 0.034 | 0.031 | 0.042 | 0.043 | 0.047 | 0.039 | 0.043 | 0.026 |
| | | C-HFD | **0.007** | **0.016** | **0.007** | **0.005** | **0.009** | **0.011** | **0.007** | **0.003** | **0.010** | **0.009** |
| | | C-LH-2.0 | 0.022 | 0.066 | 0.030 | 0.030 | 0.035 | 0.035 | 0.029 | 0.028 | 0.029 | 0.029 |
| | | C-LH-1.4 | 0.043 | 0.095 | 0.042 | 0.048 | 0.071 | 0.059 | 0.053 | 0.047 | 0.075 | 0.046 |
| | Multiple | U-HFD | **0.009** | **0.023** | **0.011** | **0.010** | **0.014** | **0.017** | **0.010** | **0.008** | **0.017** | **0.012** |
| | | U-LH-2.0 | 0.023 | 0.034 | 0.018 | 0.021 | 0.054 | 0.030 | 0.021 | 0.022 | 0.041 | 0.018 |
| | | U-LH-1.4 | 0.048 | 0.045 | 0.038 | 0.032 | 0.084 | 0.051 | 0.049 | 0.049 | 0.085 | 0.024 |
| | | ACL | 0.030 | 0.037 | 0.018 | 0.024 | 0.064 | 0.033 | 0.021 | 0.024 | 0.045 | 0.020 |
| | | C-HFD | **0.006** | **0.016** | **0.006** | **0.005** | **0.006** | **0.011** | **0.007** | **0.003** | **0.011** | **0.009** |
| | | C-LH-2.0 | 0.024 | 0.062 | 0.021 | 0.021 | 0.047 | 0.034 | 0.023 | 0.017 | 0.036 | 0.029 |
| | | C-LH-1.4 | 0.054 | 0.067 | 0.033 | 0.037 | 0.094 | 0.057 | 0.053 | 0.044 | 0.094 | 0.032 |
| F1 score | Single | U-HFD | 0.75 | **0.99** | 0.89 | 0.85 | 0.28 | 0.82 | **0.98** | 0.94 | 0.60 | **0.94** |
| | | U-LH-2.0 | 0.70 | 0.86 | 0.79 | 0.70 | 0.24 | 0.92 | 0.88 | 0.82 | 0.50 | 0.90 |
| | | U-LH-1.4 | 0.69 | 0.84 | 0.80 | 0.75 | 0.28 | 0.87 | 0.92 | 0.83 | 0.47 | 0.90 |
| | | ACL | 0.65 | 0.84 | 0.75 | 0.68 | 0.23 | 0.90 | 0.83 | 0.69 | 0.50 | 0.88 |
| | | C-HFD | **0.76** | **0.99** | **0.95** | **0.94** | **0.32** | 0.80 | **0.98** | **0.97** | **0.68** | **0.94** |
| | | C-LH-2.0 | 0.73 | 0.90 | 0.84 | 0.78 | 0.27 | **0.94** | 0.96 | 0.88 | 0.51 | 0.83 |
| | | C-LH-1.4 | 0.71 | 0.88 | 0.84 | 0.78 | 0.27 | 0.88 | 0.93 | 0.85 | 0.50 | 0.85 |
| | Multiple | U-HFD | **0.87** | **0.99** | **0.97** | 0.92 | 0.55 | 0.82 | **0.98** | **0.97** | 0.87 | **0.94** |
| | | U-LH-2.0 | 0.83 | 0.91 | 0.92 | 0.84 | 0.71 | 0.93 | 0.95 | 0.93 | 0.86 | 0.92 |
| | | U-LH-1.4 | 0.78 | 0.84 | 0.83 | 0.79 | 0.74 | 0.85 | 0.85 | 0.84 | 0.75 | 0.87 |
| | | ACL | 0.81 | 0.89 | 0.91 | 0.85 | 0.68 | 0.93 | 0.96 | 0.91 | 0.83 | 0.90 |
| | | C-HFD | 0.86 | **0.99** | **0.97** | **0.96** | 0.32 | 0.80 | **0.98** | **0.97** | 0.69 | **0.94** |
| | | C-LH-2.0 | 0.86 | 0.94 | 0.94 | 0.87 | **0.76** | **0.94** | 0.97 | 0.94 | **0.88** | 0.91 |
| | | C-LH-1.4 | 0.83 | 0.89 | 0.90 | 0.83 | 0.67 | 0.89 | 0.92 | 0.85 | 0.77 | 0.89 |

Table C.5: Local clustering results for High-school-contact network

| Metric | Method | Class 1 | Class 2 | Class 3 | Class 4 | Class 5 | Class 6 | Class 7 | Class 8 | Class 9 |
|---|---|---|---|---|---|---|---|---|---|---|
| Conductance | U-HFD | **0.25** | **0.29** | **0.13** | **0.42** | **0.21** | **0.26** | **0.16** | **0.19** | **0.25** |
| | U-LH-2.0 | 0.31 | 0.36 | 0.23 | 0.63 | 0.33 | 0.36 | 0.18 | 0.21 | 0.30 |
| | U-LH-1.4 | 0.29 | 0.32 | 0.21 | 0.54 | 0.29 | 0.37 | **0.16** | 0.22 | 0.29 |
| | ACL | 0.62 | 0.64 | 0.61 | 0.98 | 0.61 | 0.60 | 0.59 | 0.55 | 0.59 |
| | C-HFD | **0.25** | **0.28** | **0.20** | **0.41** | **0.24** | **0.26** | **0.16** | **0.19** | **0.25** |
| | C-LH-2.0 | 0.27 | 0.33 | **0.20** | 0.57 | 0.29 | 0.32 | **0.16** | 0.20 | 0.27 |
| | C-LH-1.4 | 0.28 | 0.32 | **0.20** | 0.52 | 0.28 | 0.33 | **0.16** | 0.21 | 0.28 |
| F1 score | U-HFD | **0.99** | **1.00** | 0.59 | **0.96** | 0.73 | **1.00** | 0.88 | **1.00** | **0.99** |
| | U-LH-2.0 | 0.91 | 0.83 | 0.93 | 0.66 | **0.84** | 0.88 | 0.96 | 0.96 | 0.90 |
| | U-LH-1.4 | 0.93 | 0.78 | 0.90 | 0.78 | 0.70 | 0.90 | 0.97 | 0.95 | 0.88 |
| | ACL | 0.72 | 0.73 | 0.73 | 0.06 | 0.70 | 0.76 | 0.77 | 0.78 | 0.76 |
| | C-HFD | **0.99** | **1.00** | **1.00** | **0.96** | 0.80 | **1.00** | **1.00** | **1.00** | **0.99** |
| | C-LH-2.0 | 0.93 | 0.82 | 0.92 | 0.74 | **0.84** | 0.93 | 0.97 | 0.97 | 0.91 |
| | C-LH-1.4 | 0.94 | 0.74 | 0.69 | 0.84 | 0.76 | 0.94 | 0.96 | 0.96 | 0.85 |

Table C.6: Local clustering results for Microsoft-academic network

| Metric | Method | Cluster | | | |
|---|---|---|---|---|---|
| | | Data | ML | TCS | CV |
| Cond | U-HFD | **0.03** | **0.06** | **0.06** | **0.03** |
| | U-LH-2.0 | 0.07 | 0.09 | 0.10 | 0.07 |
| | U-LH-1.4 | 0.07 | 0.08 | 0.09 | 0.07 |
| | ACL | 0.08 | 0.11 | 0.11 | 0.09 |
| F1 score | U-HFD | **0.78** | **0.54** | **0.86** | **0.73** |
| | U-LH-2.0 | 0.67 | 0.46 | 0.71 | 0.61 |
| | U-LH-1.4 | 0.65 | 0.46 | 0.59 | 0.59 |
| | ACL | 0.64 | 0.43 | 0.70 | 0.57 |

Table C.7: Top-2 node-ranking results for Oil-trade network

| Method | Query: Iran |
|---|---|
| U-HFD | Kenya, Bangladesh |
| C-HFD | Bangladesh, United Rep. of Tanzania |
| S-HFD | Turkmenistan, Iraq |

Table C.8: Local clustering results for Amazon-reviews network using $p$-norm HFD

| Metric | Seed | Method | Cluster | | | | | | | | |
|---|---|---|---|---|---|---|---|---|---|---|---|
| | | | 1 | 2 | 3 | 12 | 15 | 17 | 18 | 24 | 25 |
| Cond | Single | U-HFD-2.0 | 0.17 | 0.11 | 0.12 | 0.16 | 0.36 | 0.25 | 0.17 | 0.14 | 0.28 |
| | | U-HFD-4.0 | 0.17 | 0.10 | 0.12 | 0.16 | 0.35 | 0.26 | 0.17 | 0.14 | 0.38 |
| | Multiple | U-HFD-2.0 | 0.05 | 0.10 | 0.12 | 0.13 | 0.20 | 0.16 | 0.14 | 0.11 | 0.32 |
| | | U-HFD-4.0 | 0.05 | 0.10 | 0.12 | 0.14 | 0.20 | 0.16 | 0.14 | 0.12 | 0.32 |
| F1 score | Single | U-HFD-2.0 | 0.45 | 0.09 | 0.65 | 0.92 | 0.04 | 0.10 | 0.80 | 0.81 | 0.09 |
| | | U-HFD-4.0 | 0.48 | 0.07 | 0.65 | 0.92 | 0.04 | 0.09 | 0.80 | 0.82 | 0.10 |
| | Multiple | U-HFD-2.0 | 0.49 | 0.50 | 0.69 | 0.98 | 0.19 | 0.36 | 0.91 | 0.89 | 0.33 |
| | | U-HFD-4.0 | 0.49 | 0.50 | 0.69 | 0.98 | 0.19 | 0.36 | 0.91 | 0.88 | 0.35 |

Table C.9: Local clustering results for the food network using unit cut-costs

| Metric | Method | Cluster | | |
|---|---|---|---|---|
| | | Producers | Low-level consumers | High-level consumers |
| Conductance | U-HFD | **0.49** | **0.36** | **0.35** |
| | U-LH-2.0 | 0.51 | 0.39 | 0.39 |
| | U-LH-2.0 + flow | 0.52 | 0.39 | 0.40 |
| | U-LH-1.4 | **0.49** | 0.39 | 0.41 |
| | ACL | 0.52 | 0.39 | 0.40 |
| F1 score | U-HFD | **0.69** | **0.47** | **0.64** |
| | U-LH-2.0 | **0.69** | 0.45 | 0.57 |
| | U-LH-2.0 + flow | **0.69** | 0.45 | 0.57 |
| | U-LH-1.4 | **0.69** | 0.45 | 0.58 |
| | ACL | **0.69** | 0.44 | 0.57 |

Let $(\phi_e^*, r_e^*)$ and $y_e^*$ denote primal and dual optimal solutions for (C.1) and (C.2), respectively. Then we have that

$$r_e^* + \sigma y_e^* = s_e \quad \text{and} \quad \phi_e^{*2} = r_e^{*T} y_e^*.$$

The dual problem (C.2) can be derived following the same way that we derive the primal-dual HFD formulations, moreover, the above relations between $\phi_e^*$, $r_e^*$ and $y_e^*$ follow immediately from the primal-dual derivation, dual optimality condition and simple algebraic work. Therefore, in order to find an optimal solution $(\phi_e^*, r_e^*)$ for the primal problem (C.2), it suffices to find an optimal solution $y_e^*$ for the dual problem (C.2) and then recover $(\phi_e^*, r_e^*)$. Now, since $1^T r_e^* = 0$, we know that $\sigma 1^T y_e^* = 1^T s_e$, i.e., $y_e^*$ lies in the hyperplane $\mathcal{H} := \{y_e | \sigma 1^T y_e = 1^T s_e\}$. Let $h$ denote the objective function of the dual problem (C.2), we compute $y_e^*$ using projected subgradient method:

$$y_e^{(k+1)} := P_{\mathcal{H}} \left( y_e^{(k)} - \frac{1}{k} \frac{g^{(k)}}{\|g^{(k)}\|_2} \right),$$

where $g^{(k)} \in \partial h(y_e^{(k)})$ is a subgradient at $y_e^{(k)}$, and $P_{\mathcal{H}}(\cdot)$ denotes the projection onto the hyperplane $\mathcal{H}$. We add the additional projection step so that, when we stop the subgradient method after $K$ iterations to get $y_e^* \approx \tilde{y}_e := y_e^{(K)}$, and approximately recover $r_e^*$ as $r_e^* \approx \tilde{r}_e := s_e - \sigma \tilde{y}_e$, the resulting $\tilde{r}_e$ would still be a proper flow routing, i.e., $1^T \tilde{r}_e = 0$ and hence it is possible to have $\tilde{r}_e \in \tilde{\phi}_e B_e$ for some $\tilde{\phi}_e$. In other words, the projection step is crucial because it permits the use of sub-optimal dual solution $\tilde{y}_e$ to obtain sub-optimal but feasible primal solution $\tilde{r}_e$.

For the cardinality cut-cost, our implementation uses the projected subgradient method we describe above to solve the sub-problem in Algorithm B.1 for $\phi_e$ and $r_e$. In what follows we talk about how we deal with the specialized submodular cut-cost.

Given $e = \{v_1, v_2, v_3, v_4\}$ and associated submodular cut-cost $w_e$ such that $w_e(\{v_i\}) = 1/2$ for $i = 1, 2, 3, 4$, $w_e(\{v_1, v_2\}) = 0$, $w_e(\{v_1, v_3\}) = w_e(\{v_1, v_4\}) = 1$, and $w_e(S) = w_e(e \setminus S)$ for any $S \subseteq e$. Let $B_e$ be the base polytope of $w_e$. The sub-problem for this hyperedge is given in (C.1). Suppose $(\phi_e^*, r_e^*)$ is optimal for (C.1), and $r_e^* = \phi_e^* \rho_e^*$ for some $\rho_e^* \in B_e$. If $\phi_e^* > 0$, then we know that $\phi_e^* = \frac{s_e^T \rho_e^*}{\sigma + \|\rho_e^*\|_2^2}$. To see this, substitute $r_e^* = \phi_e \rho_e^*$ into (C.1) and optimize for $\phi_e$ only. The relation $\phi_e^* = \frac{s_e^T \rho_e^*}{\sigma + \|\rho_e^*\|_2^2}$ follows from first-order optimality condition and the assumption that $\phi_e^* > 0$. On the other hand, if $\phi_e^* = 0$, then we simply have that $r_e^* = 0$. Therefore, in order to compute $(\phi_e^*, r_e^*)$ when $\phi_e^* > 0$, it suffices to find $\rho_e^*$. In order to find $\rho_e^*$, we look at the dual problem C.2. Let $y_e^*$ be an optimal dual solution, then we have that $\rho_e^* \in \text{argmax}_{\rho_e \in B_e} \rho_e^T y_e^*$. The subsequent claims are case analyses in order to determine all possible nontrivial candidates for $\rho_e^*$.

**Claim C.1.** *If $s_{e,v_1} = s_{e,v_2}$, then $\rho_{e,v_1}^* = \rho_{e,v_2}^* = 0$; if $s_{e,v_3} = s_{e,v_4}$, then $\rho_{e,v_3}^* = \rho_{e,v_4}^* = 0$.*

*Proof.* The optimality condition of the dual problem (C.2) is for some $\hat{\rho}_e \in \text{argmax}_{\rho_e \in B_e} \rho_e^T y_e^*$,

$$(\hat{\rho}_e^T y_e^*) \hat{\rho}_e + \sigma y_e^* = s_e. \tag{C.3}$$

Suppose $s_{e,v_1} = s_{e,v_2}$, then we must have $y_{e,v_1}^* = y_{e,v_2}^*$. Otherwise, say $y_{e,v_1}^* > y_{e,v_2}^*$, then we know that $\hat{\rho}_{e,v_1} = 1/2 > -1/2 = \hat{\rho}_{e,v_2}$, which follows from applying the greedy algorithm [2] to find $\hat{\rho}_e$ using the order of indices in $y_e^*$. But then according to the optimality condition (C.3), we have

$$s_{e,v_1} = (\hat{\rho}_e^T y_e^*) \hat{\rho}_{e,v_1} + \sigma y_{e,v_1}^* > (\hat{\rho}_e^T y_e^*) \hat{\rho}_{e,v_2} + \sigma y_{e,v_2}^* = s_{e,v_2},$$

which contradicts our assumption that $s_{e,v_1} = s_{e,v_2}$. Similarly, $y_{e,v_1}^* < y_{e,v_2}^*$ is not possible, either. Now, because $y_{e,v_1}^* = y_{e,v_2}^*$, by the optimality condition (C.3), we must also have $\hat{\rho}_{e,v_1} = \hat{\rho}_{e,v_2}$. Finally, because $\hat{\rho}_e \in B_e$, we know that $\hat{\rho}_{e,v_1} + \hat{\rho}_{e,v_2} \leq 0$ and $\hat{\rho}_{e,v_1} + \hat{\rho}_{e,v_2} = -(\hat{\rho}_{e,v_3} + \hat{\rho}_{e,v_4}) \geq -w_e(\{v_3, v_4\}) = 0$, so $\hat{\rho}_{e,v_1} + \hat{\rho}_{e,v_2} = 0$. Therefore, $\hat{\rho}_{e,v_1} = \hat{\rho}_{e,v_2} = 0$. Since $\hat{\rho}$ was chosen arbitrarily from the set $\text{argmax}_{\rho_e \in B_e} \rho_e^T y_e^*$, and $\rho_e^* \in \text{argmax}_{\rho_e \in B_e} \rho_e^T y_e^*$, we have that $\rho_{e,v_1}^* = \rho_{e,v_2}^* = 0$ as required. The other claim on nodes $v_3$ and $v_4$ follows the same way. $\square$

**Claim C.2.** *If $s_{e,v_1} \neq s_{e,v_2}$ and $s_{e,v_3} = s_{e,v_4}$, then $\rho_{e,v_1}^*, \rho_{e,v_2}^* \in \{1/2, -1/2\}$ and $\rho_{e,v_3}^* = \rho_{e,v_4}^* = 0$; if $s_{e,v_1} = s_{e,v_2}$ and $s_{e,v_3} \neq s_{e,v_4}$, then $\rho_{e,v_1}^* = \rho_{e,v_2}^* = 0$ and $\rho_{e,v_3}^*, \rho_{e,v_4}^* \in \{1/2, -1/2\}$.*

*Proof.* We will show the first case, the second case follows by symmetry. Let $\hat{\rho}_e \in \text{argmax}_{\rho_e \in B_e} \rho_e^T y_e^*$. Suppose $s_{e,v_1} \neq s_{e,v_2}$ and $s_{e,v_3} = s_{e,v_4}$. Then by Claim C.1 we have

$\hat{\rho}_{e,v_3} = \hat{\rho}_{e,v_4} = 0$. Let us assume without loss of generality that $s_{e,v_1} > s_{e,v_2}$. If $y^*_{e,v_1} < y^*_{e,v_2}$, then apply the greedy algorithm we know that $\hat{\rho}_{e,v_1} = -1/2 < 1/2 = \hat{\rho}_{e,v_2}$. But this contradicts the optimality condition (C.3). Therefore we must have $y^*_{e,v_1} \geq y^*_{e,v_2}$. There are two cases. If $y^*_{e,v_1} > y^*_{e,v_2}$, then apply the greedy algorithm we get $\hat{\rho}_{e,v_1} = 1/2$ and $\hat{\rho}_{e,v_2} = -1/2$. If $y^*_{e,v_1} = y^*_{e,v_2}$, then because $\hat{\rho}_{e,v_1} + \hat{\rho}_{e,v_2} = 0$ (see the proof of Claim C.1 for an argument for this) and $\hat{\rho}_{e,v_3} = \hat{\rho}_{e,v_4} = 0$, we have that $\hat{\rho}_e^T y^*_e = 0$. But then this contradicts the optimality condition (C.3), because $s_{e,v_1} > s_{e,v_2}$ and $y^*_{e,v_1} = y^*_{e,v_2}$. Therefore we cannot have $y^*_{e,v_1} = y^*_{e,v_2}$. Since our choice of $\hat{\rho}_e \in \mathrm{argmax}_{\rho_e \in B_e} \rho_e^T y^*_e$ was arbitrary, and $\rho^*_{e,v_1} \in \mathrm{argmax}_{\rho_e \in B_e} \rho_e^T y^*_e$, so we know that $\rho^*_e$ must satisfy the properties satisfied by $\hat{\rho}_e$. $\qquad\square$

**Claim C.3.** *If $s_{e,v_1} \neq s_{e,v_2}$ and $s_{e,v_3} \neq s_{e,v_4}$, then $\rho^*_{e,v_1}, \rho^*_{e,v_2} \in \{\pm 1/2, \pm a\}$ and $\rho^*_{e,v_3}, \rho^*_{e,v_4} \in \{\pm 1/2, \pm b\}$, where $a = (\frac{1}{2} + \sigma)(s_{e,v_1} - s_{e,v_2})/(s_{e,v_3} - s_{e,v_4})$ and $b = (\frac{1}{2} + \sigma)(s_{e,v_3} - s_{e,v_4})/(s_{e,v_1} - s_{e,v_2})$.*

*Proof.* Let us assume without loss of generality that $s_{e,v_1} > s_{e,v_2}$ and $s_{e,v_3} > s_{e,v_4}$. Let $\hat{\rho}_e \in \mathrm{argmax}_{\rho_e \in B_e} \rho_e^T y^*_e$. We have that $y^*_{e,v_1} \geq y^*_{e,v_2}$ and $y^*_{e,v_3} \geq y^*_{e,v_4}$ (see the proof of Claim C.2 for an argument for this). There are four cases and we analyze them one by one in the following.

*Case 1.* If $y^*_{e,v_1} > y^*_{e,v_2}$ and $y^*_{e,v_3} > y^*_{e,v_4}$, then we have $\hat{\rho}_{e,v_1} = \hat{\rho}_{e,v_3} = 1/2$ and $\hat{\rho}_{e,v_2} = \hat{\rho}_{e,v_4} = -1/2$.

*Case 2.* If $y^*_{e,v_1} = y^*_{e,v_2}$ and $y^*_{e,v_3} = y^*_{e,v_4}$, then $\hat{\rho}_e^T y^*_e = 0$ and hence the optimality condition (C.3) cannot be satisfied. This leads to a contradiction.

*Case 3.* Suppose that $y^*_{e,v_1} = y^*_{e,v_2}$ and $y^*_{e,v_3} > y^*_{e,v_4}$. Then according to the optimality condition (C.3), because $s_{e,v_1} > s_{e,v_2}$ and $y^*_{e,v_1} = y^*_{e,v_2}$, we must have that $\hat{\rho}_{e,v_1} > \hat{\rho}_{e,v_2}$. Moreover, because $\hat{\rho}_{e,v_1} + \hat{\rho}_{e,v_2} = 0$, we know that $\hat{\rho}_{e,v_1} = a = -\hat{\rho}_{e,v_2}$ for some $a > 0$. We also know that $\hat{\rho}_{e,v_3} = 1/2$ and $\hat{\rho}_{e,v_4} = -1/2$ since $y^*_{e,v_3} > y^*_{e,v_4}$. Substitute the primal-dual relation $\phi^*_e = \hat{\rho}_e^T y^*_e$ into (C.3) we have

$$\phi^*_e \hat{\rho}_{e,v_1} + \sigma y^*_{e,v_1} = s_{e,v_1} \text{ and } \phi^*_e \hat{\rho}_{e,v_2} + \sigma y^*_{e,v_2} = s_{e,v_2}.$$

Because $y^*_{e,v_1} = y^*_{e,v_2}$, we get that

$$\phi^*_e(\hat{\rho}_{e,v_1} - \hat{\rho}_{e,v_2}) = s_{e,v_1} - s_{e,v_2},$$

and hence

$$\phi^*_e = \frac{s_{e,v_1} - s_{e,v_2}}{\hat{\rho}_{e,v_1} - \hat{\rho}_{e,v_2}} = \frac{s_{e,v_1} - s_{e,v_2}}{2a}. \tag{C.4}$$

Because $\hat{\rho} \in \mathrm{argmax}_{\rho_e \in B_e} \rho_e^T y^*_e$ was arbitrary, and $\rho^*_e \in \mathrm{argmax}_{\rho_e \in B_e} \rho_e^T y^*_e$, we know that $\rho^*_{e,v_1} = a = -\rho^*_{e,v_2}$ and $\rho^*_{e,v_3} = 1/2 = -\rho^*_{e,v_4}$. On the other hand, since $s_{e,v_1} > s_{e,v_2}$ we know that $\phi^*_e > 0$, therefore

$$\phi^*_e = \frac{s_e^T \rho^*_e}{\sigma + \|\rho^*_e\|_2^2} = \frac{a(s_{e,v_1} - s_{e,v_2}) + \frac{1}{2}(s_{e,v_3} - s_{e,v_4})}{\sigma + 2a^2 + \frac{1}{2}}. \tag{C.5}$$

Combining equations (C.4) and (C.5) we get that $a = (\frac{1}{2} + \sigma)(s_{e,v_1} - s_{e,v_2})/(s_{e,v_3} - s_{e,v_4})$.

*Case 4.* Suppose that $y^*_{e,v_1} > y^*_{e,v_2}$ and $y^*_{e,v_3} = y^*_{e,v_4}$. The following a similar argument for Case 3, we get that $\rho^*_{e,v_1} = 1/2 = -\rho_{e,v_2}$ and $\rho^*_{e,v_3} = b = -\rho^*_{e,v_4}$ where $b = (\frac{1}{2} + \sigma)(s_{e,v_3} - s_{e,v_4})/(s_{e,v_1} - s_{e,v_2})$. $\qquad\square$

Finally, combining Claims C.1, C.2, C.3 and the constraint that $\rho^*_{e,v_1} + \rho^*_{e,v_2} = \rho^*_{e,v_3} + \rho^*_{e,v_4} = 0$, there are at most 12 possible choices for $\rho^*_e$. Therefore, an exhaustive search among these candidate vectors for $\rho^*_e$ (and hence $\phi^*_e = \frac{s_e^T \rho^*_e}{\sigma + \|\rho^*_e\|_2^2}$ and $r^*_e = \phi^*_e \rho^*_e$) that minimizes (C.1) can be done using constant number of vector-vector additions and multiplications.