# OpenReview forum: "Local Hyper-Flow Diffusion"
_NeurIPS.cc/2021/Conference — NeurIPS 2021 Poster_

### Official Review · Reviewer_xXJk · 2021-06-25

**Rating:** 6
**Confidence:** 4

**Summary:**

Submodular hypergraph is a generalization of hypergraph, where each hyperedge e is associated with a submodular function w_e such that w_e(0) = w_e(e) = 0. We can naturally define the notion of conductance for submodular hypergraphs by generalizing that for graphs. The main contribution of this work is a local clustering algorithm for submodular hypergraphs, where the goal is to find a vertex set with a small conductance that exists near a seed set just by looking around it.

The algorithm is based on a flow problem that transfers some amount of mass from the seed vertices to other vertices, with some regularizer that makes the problem amenable to continuous optimization. Let C be the target cluster. Then, the algorithm outputs a set of conductance O(sqrt(C)) if vol(S \cap C) / vol(C) = \Omega(1) and vol(S \cap C) / vol(S) = \Omega(1) (vol(S) is the volume of S) and some other conditions are satisfied. It is also shown that the support of the optimal solution has a small l1 norm. The authors proposed an alternating-minimization algorithm to solve the flow problem so that the number of vertices considered (= vertices with non-zero potentials) in the intermediate steps are small.

Empirical results are encouraging. The number of vertices with non-zero potentials is just a few thousands for a hypergraph with >2 million vertices. Also, the proposed algorithm outputs a set of a smaller conductance than previous methods on synthetic graphs and real-world hypergraphs when the submodular functions associated with the hyperedges are chosen properly.

**Main Review:**

Although the results are interesting, there are several reservations.
- A local clustering algorithm based on a flow problem for graphs has been already studied in [26]. The proposed algorithm is its natural (if not simple) generalization and not quite novel (maybe this is not crucial).
- The assumption vol(S \cap C) / vol(C) = \Omega(1) and vol(S \cap C) / vol(S) = \Omega(1) look quite restrictive. Can't we start with a seed vertex (rather than a seed set that non-trivially overlaps C) to output a set of a reasonably small conductance?
- The proposed algorithm must be empirically compared with recent works [17] and [27]. I guess the proposed algorithm runs much faster but I wonder which performs better in terms of the quality (conductance) of the output.


**Time Spent Reviewing:**

3

---

> ### Author Response · Authors · 2021-08-10
> **Response to Reviewer xXJk: Theoretical guarantee for seed vertex and additional empirical comparisons**
>
> Many thanks for your time and helpful review of our work! We are pleased that you find the results in this paper interesting. In what follows we address each of your specific points. Please let us know if you have any further questions and we will be happy to discuss them.
>
> ### Extension of flow diffusion to hypergraphs
>
> We view extending a flow diffusion model from standard graphs to general submodular hypergraphs as a plus, since now we can use a familiar and effective local clustering paradigm to probe complex hypergraphs. The same spirit is also seen in both [17], where personalized PageRank vectors are generalized to hypergraphs, and [27], where capacity releasing diffusion is extended to hypergraphs. Moreover, we wish to highlight the following two points.
>
> - First, both [17] and [27] can only work with the simplest class of hypergraphs, i.e., hypergraphs associated with unit cut-cost. In this paper, the need to deal with general submodular cut-cost functions imposes several new challenges in the analyses of both clustering quality and optimization algorithm. In particular, a distinct difference between our work and [26] is that we allow for an additional regularization parameter $\sigma\ge0$ which is crucial for numerical optimization: It is a key addition that ensures that our method is highly practical while not sacrificing clustering quality. In fact, the algorithm proposed in [26] can be improved using our theoretical result in this paper, which we will briefly explain as the next point.
>
> - Second, we note that, even for standard graphs, our method reduces to a regularized version of the method introduced in [26]. The regularization parameter $\sigma$ strongly-convexifies the dual objective function and hence improves the run time of the original algorithm in [26]. We proved in this paper that for a nonempty closed interval of $\sigma$ values (cf. Assumption 3) the clustering quality is not compromised. This theoretical result has important practical implications for flow diffusion on graphs [26]: Now that one can apply a faster variant of the local algorithm in [26] and not worry about sacrificing local clustering quality. One can show that, for standard graphs, the faster variant of [26] inspired by our theoretical result in this paper would still be strongly local.
>
> ### Can we start with a seed vertex to output a set of a reasonably small conductance?
>
> This is a very good question! A short answer to your question is: Yes, we can definitely start with a single seed node to output a set of a reasonably small conductance, both in theory and in practice.
>
> To address this question in detail, we highlight the following three points.
>
> -  Empirically, our method already works very well with single seed node. We have demonstrated this point in Section 6 since all empirical evaluations use a single node as the seed. The additional empirical comparisons with [17]  and [27] (shown below) also indicate that, in practice, the assumption that $\mbox{vol}(S \cap C) / \mbox{vol}(C) = \Omega(1)$ and $\mbox{vol}(S \cap C) / \mbox{vol}(S) = \Omega(1)$ are not strictly required. In fact, we have a rigorous justification for why our method works so well even when it starts with a single seed node, please see our third point below where we state and explain a Cheeger-type guarantee for single seed node.
>
> - Theoretically, we believe that the overlap assumption (i.e., Assumption 1) may be removed if we adopt an iterative expansion procedure (e.g., [25,27]) to ensure that a single seed node expands to a seed set that has constant overlap (in volume) with the target cluster. This will require additional assumptions that the internal conductance is larger than the conductance of the target cluster, and that any node in the target cluster is better connected internally than externally. It could remove the overlap assumption but at the same time make the method less practical, due to (i) the need to repeatedly solve an optimization problem, and (ii) the need to control an increased number of parameters. It could be an interesting future study. We did not dive into this direction in this work since the current method already works very well in practice, and we try to maintain a good balance between theory and practicality.
>
> - There is actually a trivial but perhaps less elegant way to remove the overlap assumption (i.e., Assumption 1), if we instead assume that the output cluster $S_h$ has volume proportional to the volume of the target cluster $C$. More specifically, let us say $\mbox{vol}(S_h) = \hat{\alpha}\mbox{vol}(C)$ for some $\hat{\alpha}>0$. In fact, our setting guarantees that $\hat{\alpha} \in (0,3)$, which follows directly from the proof of Lemma 2. Note that the relation $\mbox{vol}(S_h) = \hat{\alpha}\mbox{vol}(C)$ tells nothing about the conductance of $S_h$. But following the same analysis and simply combining the equations (A.11) and (A.13) in the supplementary material would yield that $\Phi(S_h) \le 4(3-\hat{\alpha})\sqrt{\Phi(C)}/\hat{\alpha}$. Here we work with singe seed node so $\beta=1$. This gives a guarantee for the single seed node setting. It explains why we obtain such good empirical results using a single seed node for local clustering: In all our experiments the algorithm returns a reasonably large cluster (e.g., $\hat{\alpha} \approx 1$). We did not mention this perspective in the paper because we would like to keep the presentation succinct. We will add additional remarks to provide the reader with a more complete picture.
>
> Finally, we note that in local graph clustering applications, we are usually interested in small clusters of possibly constant size (even though the size of the graph may be arbitrarily large), and in that case having a seed set say 1/100th of the target cluster is not unreasonable.
>
> ### Empirical comparisons with [17] and [27]
>
> Thank you for the suggestion! Even though we did not think that [17] or [27] is directly comparable to ours because they can only work with unit-cut-cost hypergraphs and they run very slow in practice, we carried out extensive additional experiments to compare with [17] and [27]. The results show that our method is consistently better even in the unit cut-cost case. Due to limited space, for detailed experimental results please kindly refer to our response to Reviewer FB6z (link: https://openreview.net/forum?id=HCkJQJyoHN&noteId=j-JQLe_4ZvY).
>
> Please let us know if you have questions about details in the experiments, we are happy to discuss them.

---

> > ### Author Response · Authors · 2021-09-01
> > **Thank you for your review, we hope that our response addressed your questions**
> >
> > Dear Reviewer xXJk, we hope that our response addressed your questions. Please feel free to let us know if you have any further suggestions or reservations. We wish to thank you again for your time and insightful comments.

---

### Official Review · Reviewer_o3fQ · 2021-06-30

**Rating:** 8
**Confidence:** 4

**Summary:**

The paper introduces a novel local hypergraph partitioning framework. This utilizes a new notion of diffusion defined through an optimization problem (Section 3), and allows to partition hypergraphs using several different types of cut-cost functions on the hyperedges. Of particular interest is the case of general submodular cut-cost functions, which did not appear in prior work.

A numerical algorithm to solve the optimization problem and partition the hypergraph is provided (Section 5). The algorithm is not proved to be local, however experiments are given, illustrating the performance of the algorithm and how it compares to existing approaches to the same problem (Section 6). These show that the algorithm recovers high quality solutions on both synthetic and real-world data, and does so in a seemingly local fashion, which allows for amenable runtimes. The author(s) also motivate(s) their interest in general submodular cut-cost functions by giving concrete experimental evidence that, in specific situations (with real-world data), these lead to better quality outputs.

The author(s) also give(s) theoretical guarantees on the quality of the solution, based on clearly stated assumptions (Section 4).

**Limitations And Societal Impact:**

Not much is said about the limitations of the work, besides the lack of theoretical guarantees on the convergence time of the numerical algorithm given. In particular the author(s) do(es) not mathematically prove that the algorithm is local, but rather just provide(s) experimental evidence for this fact.

The author(s) do(es) not mention nor address any specific concern about possible negative effects of their work. In their defense, hypergraph partitioning methods have existed prior to this paper, and it seems unlikely that this specific line of inquiry will engender any kind of predictable negative societal impact.

**Main Review:**

The paper is a solid blend of sound theory and convincing experiments.

The author(s) take(s) a principled approach to local hypergraph partitioning. Their framework is clearly explained and motivated, and they show how theoretical considerations lead to a specific choice of optimization problem. The theoretical guarantees are compelling, and their proofs are clear and sound.

The experiments are also clearly outlined, and the results therein are promising and indeed suggest that the algorithm works well in practice against relatively natural benchmarks.

While the general approach of utilizing diffusion to locally partition graph and hypergraphs certainly predates this work, the author's specific formulation and choice of algorithm is, to the best of my knowledge, new.

The author(s) do(es) not mention limitations of their work, besides the lack of theoretical guarantees on the convergence time of the given numerical algorithm.  The absence of this kind of guarantee could be of concern, given that locality is a key feature of the work as it is presented. However, in light of the promising experiments, this does not seem to completely undermine the value of the results in the paper.

**Time Spent Reviewing:**

10

---

> ### Author Response · Authors · 2021-08-10
> **Response to Reviewer o3fQ: Potential limitations and societal impact**
>
> Many thanks for your time and supportive review of our work! We will add the following additional discussions on limitations and potential societal impact in the final version. Please let us know if you have any further questions and we will be happy to discuss them.
>
> **More limitations.** Another potential limitation of this work is that there are infinitely many choices of different submodular cut-cost functions. While our work opens up new opportunities to more accurately model complex higher-order relations, it brings challenges. In practice, one may be required to spend additional effort to pick the "best" submodular cut-cost. In order to design a good cut-cost function, people who work with hypergraphs may require a priori knowledge about the construction of the datasets, and/or they may need to work with domain experts to understand which specific higher-order relations are the most appropriate. The current paper does not provide the practitioners with an automatic way to select a good submodular cut-cost given a specific dataset. We believe that it would be a fruitful future work.
>
> **Societal impact.** While it is difficult to foresee any direct negative societal impact based on this work, clustering and ranking in general are methods that may raise fairness or diversity concerns. In particular, depending on the specific dataset and the choice of cut-cost, it is possible that adversarial use of our method may lead to polarized findings. For example, in the ranking example using the international oil trade network data, where we try to rank countries that are close to Iran based on a specific multiway trading pattern (Table C.7 in the supplementary material), we find that countries that share similar cultural or political values are grouped together with Iran (under a specific submodular cut-cost). This may encourage political or economical polarization.

---

### Official Review · Reviewer_Qb7P · 2021-07-15

**Rating:** 5
**Confidence:** 4

**Summary:**

The paper studies the local hyper-flow diffusion problem, and presents a local hyper graph clustering algorithm that achieves a Cheeger-type approximation guarantee.

**Limitations And Societal Impact:**

Not applicable.

**Main Review:**

The paper studies local algorithms for finding a sparse cut in a hypergraph. Both of the studied problem and the presented algorithm are interesting. The paper further presents extensive experimental studies, and the overall presentation of the paper is nice.

On the downsides, my first concern is the significance of submodular hypergraphs, and the motivation of studying such graphs isn't very clear to me. My second and major concern on this paper is on its theoretical result, i.e., Theorem 1. It is known that, in the non-hypergraph setting, a single vertex as seed is sufficient for a local algorithm to achieve a Cheeger-type guarantee, and this is also achievable for hypergraphs [17]. Unfortunately, the main result of the current submission would assume that there is a large overlap between the seed set S and the target cluster, which assumption looks very unrealistic for me.  Specifically, in order to achieve a Cheeger-type guarantee, one needs to assume that the seed set already gives a constant-factor approximation of the target set! On the other side, when the size of target cluster is large and one uses a single vertex to form the seed set S, the approximation guarantee of Theorem 1 becomes almost trivial. The paper should also clearly state the runtime of the presented algorithm.

Also, could the authors clarify their statement on Lines 204-205 that "it is the first result that is independent of hypergraph size for general hyper graphs", given the result of [17]?

The authors should also experimentally compare their algorithm with the one presented in [17].

**Time Spent Reviewing:**

4 hours

---

> ### Author Response · Authors · 2021-08-10
> **Response to Reviewer Qb7P: Significance of submodular hypergraphs, Cheeger-type guarantee for single seed vertex, and comparisons to relevant work**
>
> Many thanks for your time and helpful review of our work. In what follows we address each of your specific points. Please let us know if you have any further questions and we will be happy to discuss them.
>
> ### Significance of submodular hypergraphs
>
> Properly modelling higher-order relations in networks and graphs has recently been found crucial in modern network analysis ([7,8], Benson et al. 2021a, 2021b). The notion of submodular hypergraph constructs a hierarchy that allows modelling different levels of complexity of higher-order relations in networks and graphs. Submodular hypergraphs define cut cost directly on the set of elements in the higher-order relations.  Unit-cut-cost hyperedges correspond to the traditional hypergraphs. Cardinality-based-cut-cost hyperedges introduce the first-level of generality under the constraint that the cost is invariant to the node orders, which have been widely used in modelling higher-order potentials in images (Kim et al. 2011) and networks [13,14]. Clustering on more general submodular hypergraphs has further been adopted in ranking analysis, subspace clustering [8]. One crucial result of this work is to provide the first theoretical guarantee of the local clustering algorithm on the entire hierarchy. Our result allows practitioners safely using the proposed approach without any limitation on the modelling complexity of their hypergraphs. Note that previous works only provide the guarantee for the unit-cut-cost, very simplistic hypergraphs, which limits the applicability of their algorithms to handle complex higher-order relations.
>
> Besides the wide range of applications, recently, the theory community also have had a separate interest on studying submodular hypergraphs by investigating their Laplacian systems (Fuji et al. 2018) and sparsification (Soma and Yoshida 2019, Benson et al. 2020). Our result also fills in a big gap of this line of research.
>
> ### Theorem 1
>
> Thank you for your detailed comment about Theorem 1 in our work! We acknowledge that our theoretical framework is built upon a different assumption than what is assumed in local graph clustering using a single seed node, e.g., in [17,42]. We wish to point out that, without additional assumptions on the seed node(s), Theorem 1 may be the strongest result one may obtain. For example, [17] assumes that the single seed node lies in the "interior" of a target cluster, i.e., it is not adjacent to any other node not belonging to the target cluster. Even in the non-hypergraph setting, single node guarantee for diffusion-based method is usually established by assuming that a good seed node is selected from a (proper) subset in the target cluster, e.g., [42]. In other words, the singe node guarantee may not hold for an arbitrary seed node inside a target cluster. In practice, it is difficult (if possible at all) to verify if a seed node is "good" or not. Moreover, PageRank-based clustering methods require the teleportation parameter $\alpha$ to be proportional to the conductance of the target cluster, e.g., [17]. In practice, it is difficult to set a good $\alpha$ value a priori without knowing the conductance of the target cluster. On the contrary, our model and theoretical framework do not require the seed node(s) to be "good" nor a priori knowledge of the target conductance. Theorem 1 holds for any arbitrary set of seed node(s) as long as it has nonempty intersection with the target, and we view the $\alpha$ in the denominator of the bound as the price we pay for being certain that the guarantee would apply to any arbitrary seed set.
>
> Note that, even if a seed set $S$ and the target cluster $C$ have constant overlap in terms of volume, there is no guarantee on the conductance of $S$ with respect to the conductance of $C$. For example, consider the following setting in a standard graph: The target cluster $C$ is a clique of size $n$, there is only one edge that connects $C$ to the rest of the graph, and suppose that the seed set $S$ is a clique of size $n/2$ and that $S$ lies entirely inside $C$. In this case, we have $\mbox{vol}(S) \approx \mbox{vol}(C)/2$, but $\Phi(C) = O(1/n^2)$ and $\Phi(S) = \Omega(1)$! Therefore, the result stated in Theorem 1 is actually highly nontrivial, since it guarantees that the output set $S_h$ will satisfy $\Phi(S_h) \le O(\sqrt{\Phi(C)})$. While we admit that Theorem 1 does not provide a tight Cheeger-type guarantee for the single seed node setting as the volume of the target cluster tends to infinity, we wish to highlight the following three points. (Please also note that the setting where the volume of a target cluster tends to infinity is not typical in local graph clustering applications, since we are usually interested in small clusters of possibly constant size, and in that case having say 1/100th of the cluster is not unreasonable.)
>
> - Empirically, our method works very well for the single seed node setting. We have demonstrated this point in Section 6 since all empirical evaluations use a single node as the seed. The additional empirical comparisons with [17] (details below) also indicate that, in practice, the assumption for constant overlap is not strictly required. Actually, we have a rigorous justification for why our method works so well with single seed node, please see our third point below where we state and explain a Cheeger-type guarantee for single seed node.
>
> - Theoretically, we believe that the overlap assumption (i.e., Assumption 1) may be removed if we adopt an iterative expansion procedure (e.g., [25,27]) to ensure that a seed node expands to a seed set that has constant overlap (in volume) with the target cluster. This will require additional assumptions that the internal conductance is larger than the conductance of the target cluster, and that any node in the target cluster is better connected internally than externally. It could remove the overlap assumption but at the same time make the method less practical, due to (i) the need to repeatedly solve an optimization problem, and (ii) the need to control an increased number of parameters. It will be an interesting future study. We did not dive into this direction in this work since the current method already works very well in practice, and we try to maintain a good balance between theory and practicality.
>
> - In addition, there is actually a trivial but perhaps less elegant way to remove the overlap assumption, if we instead assume that the output cluster $S_h$ has volume proportional to the volume of the target cluster $C$. More specifically, let us say $\mbox{vol}(S_h) = \hat{\alpha}\mbox{vol}(C)$ for some $\hat{\alpha}>0$. In fact, our setting guarantees that $\hat{\alpha} \in (0,3)$, which follows directly from the proof of Lemma 2. Note that the relation $\mbox{vol}(S_h) = \hat{\alpha}\mbox{vol}(C)$ tells nothing about the conductance of $S_h$. But following the same analysis and simply combining the equations (A.11) and (A.13) in the supplementary material would yield that $\Phi(S_h) \le 4(3-\hat{\alpha})\sqrt{\Phi(C)}/\hat{\alpha}$. Here we take $\beta = 1$ because we work with single seed node as you required. This gives a guarantee for the single seed node setting! It explains why we obtain such good empirical results using a single seed node for local clustering: In all our experiments the algorithm returns a reasonably large cluster (e.g., $\hat{\alpha} \approx 1$).
>
> We will add additional remarks to provide the reader with a more complete picture.
>
> ### Comparison with [17]
>
> We would like to highlight that [17] only works with the simplest hypergraphs, i.e., hypergraphs associated with the unit cut-cost. Therefore, the sentence "it is the first result that is independent of hypergraph size for general hypergraphs" does not apply to [17] because [17] cannot work with general hypergraphs. Previous work that can deal with slightly nontrivial hypergraphs (e.g., [9] which works with cardinality-based hypergraphs) has a Cheeger-type bound that has a dependence on the maximum hyperedge size. Our model is *the first that removes this dependence* on hyperedge size, not only for cardinality-based but also for the more general submodular hypergraphs. Submodular hypergraphs are much more complex than simple unit-cut-cost hypergraphs, therefore, we consider generality as the most significant theoretical contribution of this paper: An analysis that yields an edge-size-indepedent guarantee for complex submodular hypergraphs. For importance of general submodular hypergraphs, please refer to our response above.
>
> Even though we did not think that [17] is directly comparable to ours because it only deals with unit-cut-cost hypergraphs and it runs very slow in practice (it is a locally-biased algorithm that touches all nodes in the hypergraph), we carried out extensive additional experiments using the algorithm in [17]. The results show that our method is consistently better even in the unit cut-cost case. For detailed experimental results please kindly refer to our response to Reviewer FB6z (link: https://openreview.net/forum?id=HCkJQJyoHN&noteId=j-JQLe_4ZvY).
>
> ### Run time
>
> We provided an upper bound on iteration complexity in the supplementary material (cf. Theorem B.3). Multiplying iteration complexity by cost per iteration yields the total running time of the proposed AM algorithm. We will add the total running time to Section 5 in the main text.
>
> ### References
>
> Polynomial-time algorithms for submodular Laplacian systems. Fujii et al., 2018 \
> Spectral sparsification of hypergraphs. Soma and Yoshida, 2019 \
> Augmented sparsifiers for generalized hypergraph cuts with applications to decomposable submodular function minimization. Benson et al. 2020 \
> Higher-order network analysis takes off, fueled by old ideas and new data. Benson et al. 2021a \
> fauci-email: a json digest of Anthony Fauci's released emails. Benson et al. 2021b \
> Higher-order correlation clustering for image segmentation. Kim et al. 2011

---

> > ### Author Response · Authors · 2021-09-01
> > **Thank you for your review, we hope that our response addressed your questions**
> >
> > Dear Reviewer Qb7P, we hope that our response addressed your questions. Please feel free to let us know if you still have reservations. We wish to thank you again for your time and insightful comments.

---

### Official Review · Reviewer_FB6z · 2021-07-22

**Rating:** 7
**Confidence:** 3

**Summary:**

In this work the authors provide a local diffusion framework to a general class of hypergraphs, whose cut cost functions covers numerous important special cases. The extension of the flow to this set of hypergraphs is well-thought out, and enables an optimization framework that can scale to large graphs in practice. The authors provide experiments, both on synthetic and real-world datasets.

**Ethical Concerns:**

No ethical concerns with this paper.

**Ethics Review Area:**

["I don’t know"]

**Limitations And Societal Impact:**

There exists no obvious negative societal impact.

**Main Review:**

In recent years, due to the importance of motif-based analysis of real-world networks, community detection in hyper-graphs has received a lot of attention. Furthermore, extensions of standard cut functions have been considered, including general sub-modular cut function costs. The local community detection problem has not received a lot of attention, compared to the more general hyper-graph partitioning problem. The authors propose an interesting local diffusion framework on hypergraphs, The seed nodes receive excess mass that is spread in an optimized way to produce low cost routings. This results in an interpretable primal-dual formulation that is solved using an alternating minimization method. Under mild conditions, the authors prove a Cheeger-type result for the resulting partition. The interesting part is the dependence of the parameters alpha,beta as the dependence on the conductance is as expected.  The authors perform a set of convincing experiments to illustrate the power of their framework both on synthetic and real data. Furthermore, they show on a motif-aware partitioning problem that the submodular cost function can yield significant practical improvements, e.g., in node ranking. While there could exist more baselines in the experimental section, [9] is a competitive state-of-the-art method that lies close to the proposed method.

**Time Spent Reviewing:**

7

---

> ### Author Response · Authors · 2021-08-10
> **Response to Reviewer FB6z: Empirical comparison with more baselines**
>
> Many thanks for your time and helpful review of our work. We are pleased that you find this paper interesting. Your review seems to indicate that you may be interested in seeing comparisons with more baselines. We compared with two additional methods that are also asked by other reviewers. please see details below. Please also let us know if you have questions about any other aspects (e.g., potential improvements) of the paper, we will be happy to discuss them.
>
> &nbsp;
>
> ### Empirical comparisons with more baselines [17] and [27]
>
> Since [9] is the only local diffusion method that works with slightly nontrivial cut-cost functions, and moreover, [9] has state-of-the-art performance as you mentioned, our empirical evaluations in this paper focused on comparing with [9]. We agree that other baselines exist for local clustering in simple unit-cut-cost hypergraphs. We carried out additional experiments and compared with [17] and [27] as asked by Reviewer Qb7P and Reviewer xXJk. [17] and [27] are two recent local hypergraph clustering methods that work with unit-cut-cost hypergraphs. The results (shown below under header "Results") show that our method outperforms both [17] and [27] even in the unit cut-cost case.
>
> In what follows we provide details on the new experiments. We could not find any publicly available code for [17], so we implemented the algorithm from [17] in Julia. Since [17] uses personalized PageRank vectors to perform local clustering, we denote the method by HPPR. HPPR comes with a few tuning parameters. In order to make sure that HPPR delivers the best possible performance, we allow it to "cheat" a bit by setting the teleportation parameter $\alpha$ to be the conductance of the target cluster, i.e., $\alpha$ is set to the theoretically optimal choice. Other parameters control the algorithmic dynamics of HPPR and [17] provides some suggestions based on empirical results. In our experiments we found that the parameter values suggested by the authors of [17] did not have good performance, so we fine-tuned the parameters for each dataset and for each target cluster within each dataset. We show both results: (i) **HPPR-default**, which is the suggested setting based on [17], and (ii) **HPPR-fine**, which is the setting fine-tuned by us. We denote the hypergraph capacity realsing diffusion in [27] by **HCRD**. HCRD has 5 tuning parameters, for each dataset we tune the parameters so HCRD finds low conductance clusters.  All experiments use singe seed node. We use the experimental setting as described in Section 6, unless otherwise stated below. In particular, we used a fixed parameter setting for our method and did not fine-tune it.
>
> **Synthetic data from HSBM (Table 1).** The results on synthetic data shows that when the target cluster is clean (i.e., having a low conductance 0.15), all methods work pretty well, with our method U-HFD perfectly recovers the target cluster; however, when the target cluster is noisy (i.e., having high conductances 0.25 or 0.35), U-HFD is significantly better.
>
> **Trivago-clicks (Table 2), Florida-Bay (Table 3).**  The results on Trivago-clicks and Florida-Bay datasets show that U-HFD is still better than fine-tuned HPPR. However, C-HFD and S-HFD, which use cardinality-based cut-costs and specialized submodular cut-costs, respectively, significantly outperform U-HFD. This shows the importance for working with more advanced cut-costs than the simple unit cut-cost. Neither HPPR nor HCRD can work with nontrivial cut-costs.
>
> **Amazon-clicks (Table 4).** Since both HPPR and HCRD are extremely slow for the Amazon-clicks dataset (we report running times below), we only run 10 trials for each method and then report the median results. Since the clusters in this hypergraph are very noisy, we see that U-HFD performs much better than both HPPR and HCRD.
>
> These results again show that our method is very robust: We simply picked the same parameter setting for our method, but we need to fine-tune HPPR and HCRD for each dataset. Moreover, our method is very fast, for example, in our experiments using Cluster 3 in Amazon-clicks dataset, the average running time of HFD is **2.6 seconds**, the average running time for HPPR is **175.6 seconds**, the average running time for HCRD is **119.9 seconds**. We will add the new experiments in the supplementary material to provide the reader with a better context.
>
> Please let us know if you have questions about details in the new experiments, we are happy to discuss them.
>
> &nbsp;
>
> ### Results
>
> **Table 1a.** Synthetic data from HSBM, conductance results (lower is better)
>
> | Target cluster conductance | 0.150 | 0.250 | 0.350 |
> | ----------- | ----------- | ----------- | ----------- |
> | HCRD | 0.230 | 0.381 | 0.489 |
> | HPPR-default  | 0.339 | 0.473 | 0.510 |
> | HPPR-fine | 0.165 | 0.341 | 0.460 |
> | U-HFD | **0.150** | **0.264** | **0.437** |
> | | | | |
>
> &nbsp;
>
> **Table 1b.** Synthetic data from HSBM, F1 scores (higher is better)
>
> | Target cluster conductance | 0.150 | 0.250 | 0.350 |
> | ----------- | ----------- | ----------- | ----------- |
> | HCRD | 0.95 | 0.88 | 0.76 |
> | HPPR-default  | 0.86 | 0.70 | 0.58 |
> | HPPR-fine | 0.99 | 0.91 | 0.74 |
> | U-HFD | **1.00** | **0.99** | **0.82** |
> | | | | |
>
> &nbsp;
>
> **Table 2a.** Trivago-clicks, conductance results (lower is better)
>
> | Cluster | KOR | ISL | PRI | UA-43 | VNM | HKG | MLT | GTM | UKR | EST |
> | --- | --- | --- | --- | --- | --- | --- | --- | --- | --- | --- |
> | HCRD | 0.026 | 0.041 | 0.033 | 0.027 | 0.045 | 0.039 | 0.030 | 0.029 | 0.039 | 0.028 |
> | HPPR-default | 0.018 | 0.037 | 0.026 | 0.017 | 0.042 | 0.027 | 0.019 | 0.017 | 0.031 | 0.016 |
> | HPPR-fine | 0.011 | 0.028 | 0.020 | 0.017 | 0.026 | 0.025 | 0.012 | 0.013 | 0.022 | 0.013 |
> | U-HFD | **0.010** | **0.023** | **0.014** | **0.011** | **0.018** | **0.017** | **0.010** | **0.007** | **0.016** | **0.012** |
> | C-HFD* | 0.007 | 0.016 | 0.007 | 0.005 | 0.009 | 0.011 | 0.007 | 0.003 | 0.010 | 0.009 |
> | | | | | | | | | | | |
>
> **The conductance of C-HFD is computed based on cardinality-based cut-cost, so it is not comparable with other methods that use unit cut-cost.*
>
> &nbsp;
>
> **Table 2b.** Trivago-clicks, F1 scores (higher is better)
>
> | Cluster | KOR | ISL | PRI | UA-43 | VNM | HKG | MLT | GTM | UKR | EST |
> | --- | --- | --- | --- | --- | --- | --- | --- | --- | --- | --- |
> | HCRD | 0.67 | 0.91 | 0.81 | 0.75 | 0.25 | 0.87 | 0.93 | 0.82 | 0.48 | 0.84 |
> | HPPR-default | 0.69 | 0.87 | 0.84 | 0.81 | 0.20 | 0.94 | 0.95 | 0.93 | 0.48 | 0.90 |
> | HPPR-fine | 0.73 | 0.91 | 0.86 | 0.78 | 0.24 | **0.95** | 0.96 | 0.94 | 0.57 | **0.94** |
> | U-HFD | 0.75 | **0.99** | 0.89 | 0.85 | 0.28 | 0.82 | **0.98** | 0.94 | 0.60 | **0.94** |
> | C-HFD | **0.76** | **0.99** | **0.95** | **0.94** | **0.32** | 0.80 | **0.98** | **0.97** | **0.68** | **0.94** |
> | | | | | | | | | | | |
>
> &nbsp;
>
> **Table 3a.** Florida-Bay, conductance results (lower is better)
>
> | Cluster | Producers | Low-level predators | High-level predators |
> | --- | --- | --- | --- |
> | HCRD | 0.55 | 0.54 | 0.55 |
> | HPPR-default | 0.69 | 0.46 | 0.47 |
> | HPPR-fine | 0.60 | 0.40 | 0.40 |
> | U-HFD | **0.49** | **0.36** | **0.35** |
> | S-HFD* | 0.10 | 0.08 | 0.08 |
> | | | |
>
> **The conductance of S-HFD is computed based on a specific submodular cut-cost, so it is not comparable with other methods that use unit cut-costs.*
>
> &nbsp;
>
> **Table 3b.** Florida-Bay, F1 scores (higher is better)
>
> | Cluster | Producers | Low-level predators | High-level predators |
> | --- | --- | --- | --- |
> | HCRD | 0.62 | 0.37 | 0.53 |
> | HPPR-default | 0.44 | 0.28 | 0.67 |
> | HPPR-fine | 0.49 | 0.51 | 0.60 |
> | U-HFD | **0.69** | 0.47 | 0.64 |
> | S-HFD | **0.69** | **0.62** | **0.84** |
> | | | |
>
> &nbsp;
>
> **Table 4a.** Amazon-clicks, conductance results (lower is better)
>
> | Cluster | 1 | 2 | 3 | 12 | 15 | 17 | 18 | 24 | 25 |
> | --- | --- | --- | --- | --- | --- | --- | --- | --- | --- |
> | HCRD | 0.24 | 0.81 | 0.53 | 0.79 | 0.86 | 0.86 | 0.84 | 0.88 | 0.85 |
> | HPPR-default | 0.54 | 0.60 | 0.59 | 0.86 | 0.65 | 0.55 | 0.78 | 0.65 | 0.63 |
> | HPPR-fine | 0.32 | 0.40 | 0.37 | 0.66 | 0.63 | 0.50 | 0.48 | 0.61 | 0.58 |
> | U-HFD | **0.11** | **0.13** | **0.12** | **0.16** | **0.51** | **0.23** | **0.16** | **0.14** | **0.40** |
> | | | | | | | | | |
>
> &nbsp;
>
> **Table 4b.** Amazon-clicks, F1 scores (higher is better)
>
> | Cluster | 1 | 2 | 3 | 12 | 15 | 17 | 18 | 24 | 25 |
> | --- | --- | --- | --- | --- | --- | --- | --- | --- | --- |
> | HCRD | 0.14 | 0.04 | 0.17 | 0.16 | 0.01 | 0.01 | 0.03 | 0.01 | 0.01 |
> | HPPR-default | 0.09 | 0.03 | 0.08 | 0.06 | 0.01 | 0.01 | 0.05 | 0.01 | 0.01 |
> | HPPR-fine | 0.20 | **0.08** | 0.39 | 0.29 | **0.07** | **0.03** | 0.08 | 0.09 | 0.02 |
> | U-HFD | **0.38** | 0.06 | **0.68** | **0.92** | 0.03 | **0.03** | **0.85** | **0.83** | **0.09** |
> | | | | | | | | | |

---

> > ### Comment · Reviewer_FB6z · 2021-08-27
> > **Thank you for the detailed response**
> >
> > The authors have responded adequately to my questions, and have provided details about the experimental setting. Overall, I find this paper a worthwhile contribution.

---

### Decision · Program_Chairs · 2021-09-27

**Decision:**

Accept (Poster)

**Comment:**

The paper extends the local flow diffusion framework of [26] to the submodular hypergraph setting,. The reviewers and AC agree that this is a solid theoretical contribution, even though it does not significantly depart from the graph analysis. More importantly, the reviewers and AC agree that the authors provide extensive and convincing experimental results.  These constitute the main strength of this submission.